

# The NASA Eulerian Snow on Sea Ice Model (NESOSIM): Initial model development and analysis

Alek A. Petty[1,2], Melinda Webster[1], Linette Boisvert[1,2], Thorsten Markus[1]

[1] Cryospheric Sciences Laboratory, NASA Goddard Space Flight Center, Greenbelt, MD, USA.

[2] Earth System Science Interdisciplinary Center, University of Maryland, College Park, MD, USA.

*Correspondence to*: Alek A. Petty (alek.a.petty@nasa.gov)

**Abstract.** The NASA Eulerian Snow On Sea Ice Model (NESOSIM) is a new open source model that produces daily estimates of the depth and density of snow on sea ice across the polar oceans. NESOSIM has been developed in a three-dimensional Eulerian framework and includes two (vertical) snow layers and several simple

parameterizations to represent the key sources and sinks of snow on sea ice. The model is forced with daily inputs of snowfall and near-surface winds (from reanalyses), sea ice concentration (from satellite passive microwave data) and sea ice drift (from satellite feature tracking), during the accumulation season (August through April). In this study, we present the NESOSIM formulation, initial calibration efforts, sensitivity studies and validation efforts across an Arctic Ocean domain (100 km horizontal resolution). The simulated snow depth and density are

calibrated with in-situ data collected on drifting ice stations during the 1980s. NESOSIM demonstrates very strong agreement with the in-situ seasonal cycles of snow depth and density, and shows good (moderate) agreement with the regional snow depth (density) distributions. The results exhibit strong sensitivity to the reanalysis-derived snowfall forcing data, with the MERRA/JRA-55 (ASR) derived snow depths generally higher (lower) than ERA-Interim. We derive a new 'median' daily snowfall dataset from these three reanalysis datasets

to improve reliability in our input snowfall data. NESOSIM is run for a contemporary period (2000 to 2015) and compared against snow depth estimates derived from NASA's Operation IceBridge (OIB) snow radar data from 2009-2015, showing moderate/strong agreement, especially in the 2012-2015 comparisons.

## 1 Introduction

Snow on sea ice is a crucial component of the polar climate system. Its low thermal conductivity modulates sea

ice growth through the cold winter months (e.g. Maykut and Untersteiner, 1971, Sturm et al., 2002), while its high surface albedo limits solar radiation absorption and thus inhibits sea ice melt in spring and summer (e.g. Warren, 1982; Grenfell and Perovich, 1984; Perovich, 2002). Conversely, freshwater production from snow melt facilitates melt pond formation in spring/summer which lowers the surface albedo and promotes sea ice melt



(Eicken et al., 2002; 2004). The accumulation of snow on sea ice also modulates the freshwater flux into the ocean, a key component of the freshwater budget of the Arctic (e.g. Serreze et al., 2006).

Estimates of snow depth on sea ice are also a required input for deriving sea ice thickness from satellite altimetry, e.g. from ESA's CryoSat-2 (e.g. Laxon et al., 2013) and NASA's upcoming ICESat-2 mission (Markus et al., 2017). The altimetry technique involves measurements of sea ice freeboard, the extension of sea ice above a local sea level, and estimates of snow depth to derive sea ice thickness, with snow depth being one of the primary sources of uncertainty for both laser and radar altimetry (e.g. Giles et al., 2007). Poor knowledge of snow density provides a further source of uncertainty through its influence on the ice freeboard and radar penetration into the snow pack (e.g. Giles et al., 2007, Kern et al., 2015).

Unfortunately, observations of snow depth and density across the polar oceans are lacking, due to difficulties in remotely sensing this relatively thin ($O(10$ cm)) and heterogeneous medium, and logistical challenges associated with in-situ data collection. Passive microwave data have been used to estimate snow depth over first-year ice on a basin-scale across both poles (e.g., Markus and Cavalieri 1998, Comiso et al., 2003, Maass et al., 2015), although these data are arguably more relevant for the first-year dominated Antarctic sea ice pack and tend to underestimate snow depth in deformed sea ice regimes (e.g. Worby et al., 2008; Brucker and Markus, 2013). Combinations of satellite and/or airborne sensors with variable snow penetration depths are also being explored as a means of producing basin-scale snow depth estimates (e.g. Armitage and Ridout, 2015, Guerreiro et al., 2016; Kwok and Markus, 2017), although this approach is still in its infancy and has limited temporal coverage. NASA's Operation IceBridge has provided airborne measurements of snow depth on sea ice since its launch in 2009 (Kurtz et al., 2013). However, the Arctic snow depth data collected are primarily limited to the western Arctic sea ice cover in spring (the spring 2017 campaign also included a flight over the eastern Arctic Ocean), while the Southern Ocean data have only been briefly explored to-date (e.g. Kwok and Maksym, 2014). As such, the sea ice community often relies on simple models of snow depth forced by reanalyses (primarily snowfall data) (e.g., Maksym and Markus 2008; Kwok and Cunningham, 2008; Blanchard-Wigglesworth et al., 2018) or, for the Arctic, a climatology of snow depth produced from Soviet drifting station data collected prior to 1991 (Warren et al., 1999). The Soviet drifting station data also provide the only basin-scale assessment of snow density currently available. This snow climatology is also expected to be outdated due to the rapid changes experienced in the Arctic climate system over the last few decades (Webster et al., 2014).



In this study we present a new model to derive snow depth (and density) across the polar oceans. Our objective is to produce reliable basin-scale daily snow depth and density estimates needed for satellite altimetry calculations of sea ice thickness for both historical analyses and near real-time operations. A secondary utility of the model will be the production of daily/monthly/seasonal snow depths that can help guide climate modelling research efforts addressing the representation and importance of snow on sea ice in the global climate system. In the following sections, we present and describe the model configuration/physics, the sensitivity of the model to the input forcing data (e.g. reanalyses snowfall, satellite-derived ice drifts), and initial model calibration/validation efforts. We focus this initial study solely on the Arctic, however our plan is for the model to be applied and tested in a Southern Ocean framework in the near future. We conclude by looking ahead to potential improvements in the model physics and planned future activities related to our efforts to improve our understanding of snow on sea ice.

## 2 Model description

The NASA Eulerian Snow On Sea Ice Model (NESOSIM) is a three-dimensional, two-layer (vertical), Eulerian snow budget model developed with the primary aim of producing daily estimates of snow depth and density across the polar oceans. NESOSIM includes several parameterizations that represent key mechanisms of snow variability through the snow accumulation/growth season, and two snow layers to broadly represent the evolution of both old/compacted snow and new/fresh snow. The model schematic is shown in Figure 1. Our aim was to produce a model of physical and computational simplicity to allow for a detailed assessment of the sensitivity of the modeled snow depths to the various input data used. We expect the model to increase in complexity with future model developments, e.g. new parameterizations or improvements to existing parameterizations as needed. We decided on a Eulerian snow budget approach (as opposed to a Lagrangian approach, e.g. Kwok and Cunningham, 2008) for a number of reasons: (i) it provides a framework flexible to the presence (or lack of) ice drift data, increasing the utility of the model in regions/time periods where ice drift data might be lacking, (ii) it provides a simple assessment of the spatial significance of the parameterized budget terms included in the model, including ice dynamics, and (iii) the parameterizations developed in this framework can be easily transferred to other Eulerian sea ice models (e.g. the Los Alamos sea ice model CICE) included in General Circulation Models (GCMs). The following subsections detail the model setup and various parameterizations currently included in NESOSIM.





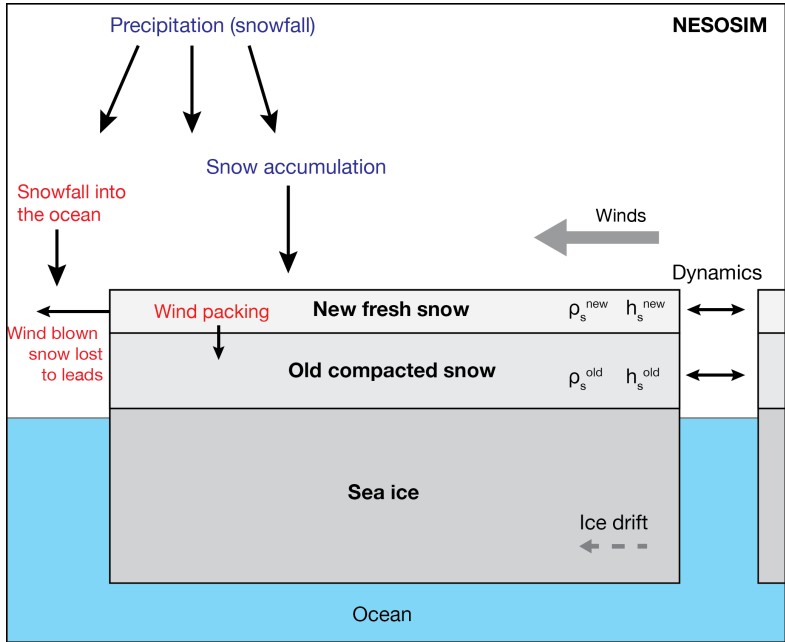

**Figure 1**: Schematic of the NASA Eulerian Snow On Sea Ice Model (NESOSIM) presented in this study. The red (blue) text indicates processes that result in a loss (gain) of snow. 'Dynamics' indicates the combination of ice/snow advection and convergence/divergence which can cause either loss or gain of snow.

## 2.1 Model configuration

NESOSIM includes two vertical layers on an x/y horizontal grid, with each horizontal grid-cell and snow layer featuring a prognostic snow volume and fixed snow density. This two-layer approach was taken to represent the strong differences in properties between dense snow, associated with wind slab, and fresh, cold snow from recent snowfall, while keeping the model computationally efficient and the model physics easily trackable. As stated previously, our plan is that NESOSIM will be used for studying snow on sea ice across the Arctic and Southern Oceans, however, for this initial analysis we run the model on a 100 km x 100 km polar stereographic grid covering the Arctic Ocean and peripheral seas (shown later). This grid resolution was chosen due to considerations of computational efficiency and the horizontal resolutions of the various input data.

The model is forced with daily data of snowfall, near-surface winds, ice concentration, and ice drift. We discuss the forcing datasets used in this study in Section 3. Note that in the model we track the evolution of snow volume (the volume of snow per unit grid cell, in units of meters) for simplicity, instead of snow depth. An effective



snow depth within a given grid cell is then produced by dividing the snow volume by the grid-cell ice
concentration. The model run is initiated each year from the end of summer (default of August 15[th], rationale
discussed in Section 2.5) to the middle of spring (May 1[st]), to avoid the complexity of snow melt processes
expected outside of these dates. We hope to extend the model runs into the melt season in future model

development efforts. The default model configuration (forcings/parameter settings) are given in Table 1.

| Model parameter | Default setting |
|---|---|
| New snow density (kg m$^{-3}$) | 200 |
| Old snow density (kg m$^{-3}$) | 350 |
| Wind packing coefficient, $\alpha$ | 0.05 |
| Blowing snow coefficient, $\beta$ | 0.0025 |
| Wind packing threshold, $\omega$ (m s$^{-1}$) | 5 |
| **Forcing data** | |
| Snowfall | ERA-I/MEDIAN-SF (as specified) |
| Near-surface winds | ERA-I |
| Sea ice concentration | Bootstrap |
| Sea ice drift | NSIDCv3 (Polar Pathfinder) |

**Table 1**: Default model forcings and parameter settings used by NESOSIM.

### 2.2 Snow accumulation

To accumulate snow in our model, the snowfall water equivalent from our reanalysis data is converted to snow
volume using a representative snow density. Snow pit and density data from the Surface Heat Budget of the

Arctic Ocean (SHEBA) experiment and the Soviet drifting ice station data helped guide the parameterization of
our seasonal snow density evolution. Our modelled snow is partitioned into two density layers: a "new" layer,
which represents recent snowfall, and an "old" layer, which represents snow that has undergone wind compaction
and snow grain metamorphism (Colbeck, 1982; Sturm and Massom, 2017). Initially, snow accumulates into the
new/fresh snow layer within a given grid cell as

Eq. (1):

$$\Delta h_s^{acc}(t,x,y) = S_f(t,x,y)\, A(t,x,y)/\rho_s^n ,$$





where $S_f$ (in units of kg m$^{-2}$) is the gridded daily snowfall across the model domain, $\rho_s{}^n$ is the density of the new snow layer, $A$ is the gridded daily ice concentration, $t$ is the daily time index, and $x$ and $y$ are the horizontal grid indices. The density of the new snow layer is fixed at $\rho_s{}^n = 200$ kg m$^{-3}$. This value implicitly represents a combination of cold, dry snowfall (~150 kg m$^{-3}$) and wet snowfall (~230 kg m$^{-3}$) based on direct observations

over Arctic sea ice (Radionov et al., 1997; Sturm et al., 2002).

Snow can be transferred from the new snow layer to the old snow layer depending on the strength of the near-surface wind forcing. The old snow layer is an implicit combination of two layers that, on average, comprise the majority of the snowpack bulk density: wind slab and depth hoar (Sturm et al., 2002; Sturm, 2009). The density of wind slab ranges between ~300 kg m$^{-3}$ and ~400 kg m$^{-3}$ on average (Colbeck, 1982; Radionov et al., 1997;

Warren et al., 1999; Sturm et al., 2002), while depth hoar has an average density of ~150 - 250 kg m$^{-3}$ (Colbeck, 1982; Sturm et al., 2002). Based on SHEBA data, both layers contribute roughly equally to the bulk thickness of the snow cover, comprising ~80% of it collectively (Sturm et al., 2002). For this reason, we use the average of the higher-end values of wind slab and depth hoar as the density value for the old snow layer. However, we note that the ratio of wind slab and depth hoar layers depends on several factors including the atmospheric conditions

during precipitation events, sea ice surface roughness, snow depth, and the internal snowpack temperature gradient (Sturm et al., 2002). Our simple parameterization scheme is thus expected to be generally representative of basin-wide conditions, but will contribute to uncertainty in our modeled snow depths.

When wind speeds are greater than 5 m s$^{-1}$, the change in snow depth from wind packing between the two snow layers respectively is given as:

Eq. (2):

$$\Delta h_s{}^{wp}(t,0,x,y) = -\alpha\, h_s(t,0,x,y) \text{ for } U(t,x,y) > \omega$$

$$\Delta h_s{}^{wp}(t,1,x,y) = (\rho_s{}^n/\rho_s{}^o)\alpha h_s(t,0,x,y) \text{ for } U(t,x,y) > \omega$$

where $U(x,y)$ is the 10 m wind speed, $\omega$ is a wind speed threshold for wind packing to occur (default of 5 m/s), $\alpha$ is a wind packing coefficient which determines the fraction of the new snow layer that is transferred into the old

snow layer (default value of 0.05), and $\rho_s{}^o$ is the density of the old snow layer. The second grid index in Eq. 2 (values of 0 and 1) represents the vertical snow layers. The wind threshold of 5 m s$^{-1}$ was determined based on


observational and modeling studies of blowing snow in the terrestrial Arctic and sea ice environments (Pomeroy et al., 1997; Radionov et al., 1997; Sturm and Stuefer, 2013).

**2.3 Ice/snow dynamics**

Snow within a given grid cell can also evolve due to ice drift. Here we adapt the ice concentration budget approach used in e.g. Holland and Kimura (2016) (and more recently in Petty et al., 2018) to snow volume as

Eq. (3):

$$\Delta h_s{}^{dyn}(t,x,y) = \nabla(h_s(t,x,y).u_i(t,x,y)),$$

where $u_i$ is the daily gridded ice drift. As in the ice concentration budget studies discussed above, we can expand this into a divergence/convergence term and an advection term as

Eq. (4):

$$\Delta h_s{}^{div}(t,x,y) = h_s(t,x,y).\nabla(u_i(t,x,y)) \text{ and}$$

$$\Delta h_s{}^{adv}(t,x,y) = \nabla(h_s(t,x,y)).u_i(t,x,y),$$

where $\Delta h_s{}^{div}$ is the change in snow volume from divergence/convergence, i.e. changes due to spatial gradients in ice drift, and $\Delta h_s{}^{adv}$ is the change in snow volume from advection, i.e. changes due to spatial gradients in snow volume (assuming constant drift). Note that this parameter is applied to both 'old' and 'new' snow layers concurrently.

**2.4 Blowing snow lost to leads**

Snow within a grid cell can also be lost to leads/open water in the ice pack due to the impact of wind forcing, i.e. blowing snow lost to leads. This parameter is expected to be most significant in regions where high lead fractions, wind speeds and snowfall (e.g. the marginal ice zone in the North Atlantic sector of the Arctic) are expected to result in significant wind blown snow lost to leads/open water (e.g. Leonard and Maksym, 2011). Note that we only apply this wind loss term to the new snow layer as we assume the 'old' wind packed snow layer is immune to the impact of wind forcing (e.g. Petrich et al., 2012; Trujillo et al., 2016). The blowing snow to leads is calculated as



Eq. (5):

$$\Delta\, h_s{}^{bs}(t,x,y) = \beta\, U(t,x,y)\, h_s(t,0,x,y)(1-A(t,x,y)),$$

where $\beta$ is a blowing snow coefficient (default value of 0.025).

We also keep track of snow that enters the ocean through snowfall into the open water fraction and blowing snow

lost to leads, a quantity of relevance to those interested in the freshwater budgets of the polar oceans. This is

given as

Eq. (6):

$$S_f{}^{oce}(t,x,y) = S_f(t,x,y)\,(1-A(t,x,y))/\rho_s{}^n \;+\; \Delta\, h_s{}^{bs}(t,x,y\,.$$

**2.5 Model evolution**

At each daily time step, the snow volume and density within each grid cell is updated based on the budget terms

described above using a forward Euler method as

Eq. (7):

$$h_s(t+1,0,x,y) = \; h_s(t,0,x,y) + \; \Delta\, h_s{}^{acc}(t,x,y) + \Delta\, h_s{}^{dyn}(t,0,x,y) + \Delta\, h_s{}^{wp}(0,x,y) + \Delta\, h_s{}^{bs}(x,y)$$

and

$$h_s(t+1,1,x,y) = \; h_s(t,1,x,y) + \Delta\, h_s{}^{dyn}(t,1,x,y) + \Delta\, h_s{}^{wp}(1,x,y).$$

Note that we also calculate a bulk snow density, which is the weighted average density across the two snow

layers, as

Eq. (8):

$$\rho_s{}^b\,(t,x,y) = \; ((h_s(t,0,x,y)\rho_s{}^n + \; h_s(t,1,x,y)\rho_s{}^o)/(h_s(t,0,x,y) + h_s(t,1,x,y)).$$

As discussed earlier, each model run is initialized at the end of summer (default of August 15[th]) and run until the

following spring (May 1[st]). This early summer start time was chosen to include the significant snowfall expected



across the Central Arctic through August (Radionov et al., 1997; Warren et al., 1999; Boisvert et al., submitted). We acknowledge that this end of August time period also likely includes surface melt events that are not captured/included in this model but are hoped to be addressed in future model developments. We also apply a variable initial snow volume (at $t = 0$) across our model domain, as discussed in the following section. The model

is easily adaptable, such that it can be reinitialized from a later date and run forward in time, making it suitable for near real-time operations, an expected use of this model in the near future.

For initial model calibration we also ran NESOSIM with different combinations of the model parameterizations discussed above. When we turn off the wind-packing parameterization, snow remains fixed in the 'new' snow layer, despite the strength of the wind forcing, so the model effectively becomes a 1-layer model. To account for

the low bias in snow density expected by constraining the snow density to the density of fresh/new snow, we forced this snow layer with the daily climatological snow density based on Warren et al., (1999), which we refer to as ρ-W99.

### 3 Model forcing and calibration/validation data

In the following subsections we describe the forcing data and calibration/validation data used in this study,
including atmospheric forcing data (snowfall and winds), satellite-derived ice drifts, satellite-derived ice concentration, soviet drifting station snow depths/densities (for model calibration) and Operation IceBridge snow depths (for model validation).

### 3.1 Atmospheric forcing

We use snowfall data provided by the European Center for Medium Range Weather Forecast (ECMWF) ERA-
Interim (ERA-I) reanalysis. ERA-I is a global reanalysis that utilizes a 4D variational data assimilation scheme (Dee et al., 2011). We use the 12-hourly ERA-I snowfall data from August 15[th] 1980 to May 1[st] 1991 and August 15[th] 2000 to May 1[st] 2015. We use the 0.75° x 0.75 ° horizontal resolution data, which are summed to produce daily snowfall estimates across the Arctic. ERA-I snowfall data have been used in previous studies exploring snow accumulation over Arctic sea ice (e.g. Kwok and Cunningham, 2008; Blanchard-Wigglesworth et al.,
2018), while comparisons of reanalysis-derived precipitation data with coastal weather stations suggests ERA-I is one the better products available for Arctic studies (Serreze and Hurst, 2000; Lindsay et al., 2014). A more detailed comparison of snowfall/precipitation estimates over the Arctic Ocean has recently been carried out alongside this study (Boisvert et al., submitted), which we expect to build on in the future.



| Reanalysis | Producer | Resolution* | Coverage |
|---|---|---|---|
| ERA-Interim | European Centre for Medium-Range Weather Forecasts (ECMWF) | 0.75° x 0.75° | 1979 - present (NRT, few months data latency) |
| ASRv1 | Various contributors, see Bromwich et al., (2016) | 30 km x 30 km | 2000 - 2012 |
| JRA-55 | Japanese Meteorological Agency (JMA) | 0.56° x 0.56° | 1958 - present (NRT, few months data latency) |
| MERRA | NASA's Global Modeling and Assimilation Office (GMAO) | 0.5° x 0.66° | 1979 - June 2016 |

**Table 2:** Summary of the four different reanalysis datasets used in this study (data availability often subject to change/updates, information given at the date of submission). NRT: Near real-time. *different resolutions available in some cases.

We explore the sensitivity of our results to the input snowfall data by forcing the model with snowfall estimates provided by three additional reanalysis-derived snowfall products. Unfortunately, not all reanalyses provide direct estimates of snowfall (and rainfall), and instead provide just total precipitation, e.g. the data from the widely used National Centers for Environmental Prediction (NCEP)-National Center for Atmospheric Research (NCAR) Reanalysis 1 and 2, so we focus our analysis on three other commonly used reanalyses that provide direct estimates of snowfall: the Japanese Meteorological Agency 55-year Japanese reanalysis (JRA-55); NASA's Modern-Era Retrospective Analysis for Research and Application (MERRA); and the Arctic System Reanalysis, version 1 (ASRv1), as described below and summarized in Table 2.

*JRA-55*: The Japanese Meteorological Agency (JRA) 55-year Japanese reanalysis (JRA-55) is a global atmospheric reanalysis that utilizes a 4-D variational assimilation system covering the period 1958 to present (Kobayashi et al, 2015). JRA-55 was developed as an improvement to their previous 25-year reanalysis (JRA-25), which we do not include in this study. We use the daily JRA-55 snowfall data from August 15[th] 1980 to May 1[st] 1991 and August 15[th] 2000 to May 1[st] 2015. The data were obtained from the National Center for Atmospheric Research's Research Data Archive at a horizontal resolution of 0.56° x 0.56° (~ 60 km), downscaled from the original 1.25° x 1.25° Gaussian grid. The data are being produced on a near real-time basis (2-6 month data latency).

*MERRA:* NASA's Modern-Era Retrospective Analysis for Research and Application (MERRA) is a global reanalysis that utilizes a 3D variational data assimilation scheme within the Goddard Earth Observing System Data Assimilation System (GEOS-5) (Rienecker et al, 2011). We use the daily MERRA snowfall data from August 15[th] 1980 to May 1[st] 1991 and August 15[th] 2000 to May 1[st] 2015. The data are provided at a horizontal resolution of 0.5° (latitude) by 0.66° (longitude). Note that an updated version of MERRA (MERRA-2) is also

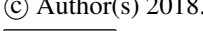



available, but is known to have a high precipitation bias compared to the other reanalyses (Boisvert et al., submitted) so we exclude this from our study.

*ASRv1*: The Arctic System Reanalysis, version 1 (ASRv1) is a regional reanalysis based on the Weather Research and Forecasting model (Polar WRF) that utilizes a 3D variational data assimilation scheme and is adapted for the

polar regions (Hines and Bromwich, 2008). The ASRv1 data are only available from 2000 to 2012, so we use the daily snowfall data from August 15th 2000 to May 1st 2012, which is provided at a horizontal resolution of 30 km x 30 km.

Considering the expected importance and uncertainty of the reanalysis-derived snowfall for deriving snow depth, we also produce a synthesized snowfall dataset by taking the median snowfall across the gridded snowfall

products, for each daily grid-cell (data referred to as MEDIAN-SF). We use the gridded ERA-I, JRA-55 and MERRA snowfall data, as these products all cover the longer-term (1980-2015) time period.

NESOSIM also requires daily estimates of near-surface winds to drive the wind packing and wind loss terms, which we take from the ERA-I reanalysis for all reanalysis model runs. Jakobsen et al., (2012, Figure 2) show that ERA-I winds had the lowest biases of several reanalysis-derived near-surface wind estimates compared to

TARA drifting station data. We compute the magnitude of the winds from the six-hourly *u/v* vectors before averaging to produce a daily (gridded) wind magnitude dataset.

We linearly interpolate all the daily snowfall (and ERA-I wind magnitude) estimates onto our 100 km x 100 km polar stereographic model domain.

### 3.2 Satellite derived ice drift data

We primarily make use of the daily Polar Pathfinder ice drift data, version 3 (Tschudi et al., 2016) made available through the National Snow and Ice Data Center (the product is referred to herein as NSIDCv3). A daily ice drift is calculated using a cross-correlation technique applied to sequential daily satellite images acquired by passive microwave satellite sensors (i.e. a one day lag in parcel tracking) which are blended via optimal interpolation with estimates from the International Arctic Buoy Programme (IABP) and wind data from the National Centers

for Environmental Prediction (NCEP)-National Center for Atmospheric Research (NCAR) Reanalysis. The data are available daily from October 1978 through February 2017 (at the time of writing) at a horizontal resolution of 25 km x 25 km. In this study we use the daily data from August 15th 1980 to May 1st 1991 and August 15th 2000





to May 1st 2015. We grid the daily ice drift data onto our 100 km model domain (using linear interpolation) and smooth the data using a simple Gaussian filter (as in Holland and Kimura, 2016 and Petty et al., 2018).

Recent studies have explored the uncertainty in satellite-derived ice drift data (Sumata et al., 2014) and errors introduced by the NSIDC interpolation methodology (Szanyi et al, 2016). We thus also explore the sensitivity of
the model results to the input ice drift data by forcing the model with ice drift estimates provided by three additional satellite-derived ice drift products, as described below and summarized in Table 3.

| Product | Resolution | Daily lag | Data source | Coverage | Availability |
|---|---|---|---|---|---|
| NSIDCv3 | 25 km | 1 day | AVHRR, SMMR, SSM/I , AMSR-E,IAPBs, NCEP-R1 | October 1978 - Feb 2017 | Public |
| OSI-SAF | 62.5 km | 2 days | ASCAT* | October 2010 - present | Public/NRT |
| KIMURA | 60 km | 1 day | AMSR-E, AMSR-2 | Jan 2003 - Sep 2011 / July 2012 - Dec 2016 | On request |
| CERSAT | 62.5 km | 3 days | ASCAT* | January 2007 - present? | Public/NRT |

**Table 3**: Summary of the different ice drift datasets used in this study based on information obtained at the time of submission. *These agencies produce drift datasets using different individual/combinations of satellite sensors not utilized in this study. NRT: Near real-time.

*OSISAF*: The European Organization for the Exploitation of Meteorological Satellites (EUMETSAT) produce a number of low-resolution sea ice drift products from satellite passive microwave sensors and scatterometry (Lavergne, 2010). Here we use the merged ice drift product, which increases coverage and reliability over their single sensor drift products (Lavergne, 2010). The merged drift product uses a 2 day lag in ice parcel tracking and a Continuous Maximum Cross Correlation (CMMC) method to optimize the drift product, and is available daily
(October through April) since 2010 at a horizontal resolution of 62.5 km x 62.5 km.

*CERSAT:* The Centre ERS d'Archivage et de Traitement (CERSAT), part of the Institut Français de Recherché pour l'Exploitation de la Mer (IFREMER) produce a number of ice drift datasets by merging various combinations of satellite passive microwave and scatterometry data (Girard-Ardhiun and Ezraty 2012). Here we use data produced from the merging of Advanced Scatterometer (ASCAT) and the Special Sensor Microwave
Imager (SSMI) data, which are available daily (September to May) since 2007 at a horizontal resolution of 62.5 km x 62.5 km. Note that CERSAT provide data using both a 3 and 6 day lag in the tracking of ice displacement, but we use the 3 day lag data as this is closest to the 1 day lag used by the NSIDCv3 product.



*KIMURA:* The KIMURA drift data are produced using brightness temperatures obtained by the Advanced Microwave Scanning Radiometer for EOS (AMSR-E) from January 2003 to September 2011 and the Advanced Microwave Scanning Radiometer 2 (AMSR-2) from July 2012 to December 2016 using a cross-correlation approach (see Kimura et al., 2013 for more details). Wintertime (November-December, January-March) ice drifts

are derived using the 36-GHz channel, while the summertime drifts used in this study (August-October, April) are derived using the 18-GHz channel, to maximize the reliability and coverage of the data. The data are provided at a 60 km x 60 km horizontal resolution.

We use data from these three additional products from August 15th 2010, 2012, 2013 and 2014 to May 1st of the subsequent years, a period of coincident data coverage across the four drift products (including NSIDCv3). We

linearly interpolate all the daily drift datasets onto the 100 km x 100 km polar stereographic model domain used in this study. As highlighted above, not all the products produce drift estimates in August, or even September, so for those products we assume no ice drifts through this period. To investigate the importance of ice drift, we also run the model assuming no ice drift for the entire model simulation (NODRIFT), as discussed in more detail later.

### 3.3 Sea ice concentration

We use the daily Bootstrap sea ice concentration (SIC) data, version 3 (Comiso, 2000 updated 2017), which are produced from passive microwave brightness temperature estimates and made available through the NSIDC. We choose to primarily use the Bootstrap over, for example NASA Team data (Cavalieri et al., 1996 updated 2017), another commonly used SIC dataset, as Bootstrap SIC data are less sensitive to surface melt, producing higher concentrations in general. We use the NASA Team data in a sensitivity study to explore the sensitivity of the

model to this choice of sea ice concentration data. Due to differences in satellite orbit and sensor characteristics, the SIC data feature a time-varying pole hole depending on the passive microwave sensor used. As we require consistent SIC data across the pole hole, we follow the approach of Petty et al., (2018) and apply a mean SIC calculated in a $0.5^o$ halo around the variable pole hole to all grid cells within the pole hole. The data are provided at a 25 km x 25 km resolution polar stereographic grid from 1978 through 2016, and we use the daily data from

August 15[th] 1980 to May 1[st] 1991 and August 15[th] 2000 to May 1[st] 2015. We linearly interpolate the daily SIC data onto our 100 km x 100 km model domain. Note that a gap in the passive microwave record exists from December 3rd 1987 to January 13th 1988, so we do not run the model through the 1987-1988 winter period.



### 3.4 Soviet station data and initial conditions

We use *in-situ* snow data collected on the former Soviet Union's drifting ice stations for initial model calibration and to help guide our choice of initial conditions (Radionov et al., 1997; Warren et al., 1999; Fetterer and Radionov, 2000). The drifting ice stations were in operation in 1937 and 1954-1991, although in this study we

use the field observations collected from 1980-1991 due to the temporal overlap with the model forcing data. During the drifting ice stations, snow depth data were collected every 10 days in 10 m intervals along a 500 m or 1000 m survey line. Snow density measurements were made every ~100 m along the same survey lines, and atmospheric conditions were recorded at near-daily frequencies. Despite their limited spatial coverage, these data provide the most complete record of snow and atmospheric conditions to date over the Arctic sea ice pack.

*Initial conditions*: We initialize the model on August 15$^{th}$ of each year with a snow depth representing the fraction of snow assumed to have survived the summer melt season and/or accumulated during summer. The August snow depth climatology compiled by Warren et al., (1999, referred to herein as W99) from the Soviet station data suggests significant amounts of snow (up to 10 cm) are present in late summer, especially over the Central Arctic sea ice north of Greenland. This inclusion of an initial snow depth was also guided by our preliminary model

calibration studies that showed that including these initial conditions provided a better match with the seasonal snow depths observations (calibrations presented later). To produce initial mid-August snow depths, we use a near-surface air temperature-based scaling of the August W99 snow depth climatology to account for changes in the duration of the summer melt season (e.g. Markus et al., 2009). Briefly, we calculate the annual number of days with continuous, above-freezing, air temperatures (taken from the ERA-I reanalysis), which we refer to here

as the summer melt duration. To create an initial (August) snow depth estimate for a given year, we linearly scale the W99 August snow depth climatology based on the summer melt duration of the chosen year and the climatological summer melt duration given in Radionov et al., (1997). If the melt duration is longer than the climatological mean in a specific region, the scaled August climatology reflects a reduction in snow depth in August due to the longer melt season. The snow depth is then distributed evenly over the 'old' and 'new' snow

layers based on the climatological observations that some snow persists through summer (Radionov et al., 1997), and the occurrence of summer snowfall events (Radionov et al., 1997; Perovich et al., 2017). While admittedly this is a crude approach for parameterizing an initial snow depth, our sensitivity studies demonstrated that initial conditions were necessary to improve the comparison with the drifting station observations (as presented and discussed in the following section), and indicate that late summer snowfall events might play a significant role in

establishing the snow cover on Arctic sea ice prior to the fall/winter season (Warren et al., 1999). The August



W99 snow depth climatology and temperature scaled initial snow depth estimates (for 2012 and 2013) are shown in Figure 2.

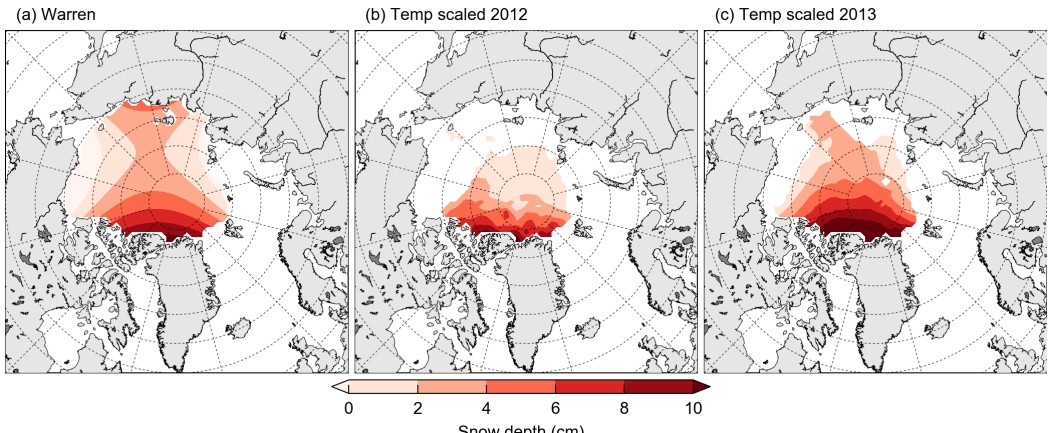

**Figure 2**: (left) Warren climatology of August snow depth, (middle and left) the initial conditions used in this study (broadly representing the snow volume as of August 15th) for 2012 and 2013 respectively, calculated using near-surface air temperature scaling.

*Model calibration:* For our model calibration we use the raw snow depth and density data from the Soviet drifting stations 25, 26, 30 and 31. The data represent the average of a given survey line. The majority of survey lines remained constant each time they were sampled, so the dataset is a near-continuous time-series with a 10-day temporal resolution. Most survey lines were 1000 m in length, although in the earlier part of the historical record (e.g., before the 1980s), some ice stations had survey lines that were 500 m in length. Maps of the drifting stations are given in the supplementary information (Figure S3). Briefly, Station 25 drifted from the Central Arctic to the East Siberian Sea providing data from autumn 1981 to spring 1984, Station 26 drifted around the north of the East Siberian Sea providing data from autumn 1983 to spring 1984, Station 30 drifted around the north of the East Siberian Sea providing data from autumn 1988 to winter 1991, and Station 30 drifted around the Beaufort Sea providing data from winter 1989 to winter 1991. We use a simple nearest neighbor algorithm to match the data to the nearest model grid-cell for the relevant day the drifting station data were collected.

### 3.5 NASA's Operation IceBridge data

We compare our NESOSIM snow depth estimates with spring snow depths collected by NASA's Operation IceBridge (OIB) airborne mission. NASA's OIB mission began collecting airborne observations of the polar



regions in 2009, bridging the gap between NASA's Ice, Cloud, and land Elevation Satellite (ICESat) mission which retired in 2009, and the future ICESat-2 mission scheduled for launch in the summer of 2018 (Markus et al., 2017). The OIB aircraft carry a suite of instruments designed to measure both land and sea ice, including their overlying snow cover. Here we primarily make use of snow depth estimates derived from the ultra-wideband

Snow Radar (Panzer et al., 2013), which are available at a 40 m along-track resolution. Various algorithms have been developed to produce snow depth estimates from the OIB Snow Radar data (Kwok et al., 2017), with the products showing broad agreement in the regional snow depth distributions, but significant intraregional and interannual differences, due primarily to changes in the radar configuration and algorithm tuning. To account for these differences we use the snow depth data from the (i) Snow Radar Layer Detection (SRLD) (Koenig et al.,

2016), (ii) NASA Goddard Space Flight Center (GSFC) (Kurtz et al., 2013) and (iii) Jet Propulsion Laboratory (JPL) (Kwok and Maksym, 2014; Kwok et al., 2017) snow depth products, that have produced, and made available, snow depth estimates at a 40 m along-track resolution from 2009 to 2015. We bin the 40 m OIB snow depth data onto our 100 km model grid and keep only the grid cells that included a significant quantity (> 1000 points) of the raw snow depth data. The OIB data are provided for various days through spring of the relevant

campaign (data from mid-March to early-May, depending on the campaign year), so we grid the OIB data daily, and compare this with coincident (daily) NESOSIM snow depth estimates. The OIB data are collected mainly over the western Arctic sea ice, limiting our validation effort to this region of the Arctic.

**4 Model calibration and analysis**

We carried out initial model calibration over the period Aug 15[th] 1980 - May 1[st] 1991 due to the coincident Soviet

station data available during this period. As noted previously, this excludes the 1987-1988 winter season due to the lack of complete sea ice concentration data available during this period. As stated earlier, our initial calibration efforts involved manually tuning NESOSIM to improve the general fit with the mean seasonal snow depth/density cycles shown in the Soviet station data. Specifically, we included the temperature-scaled initial August snow depths and tuned both the wind packing coefficient, $\alpha$ (Eq. 5), and blowing snow coefficient, $\beta$ (Eq.

6). We decided against a more optimized calibration effort due to limitations in the calibration data, i.e. its sparse availability in space/time and differences in spatial scales. We instead used the drifting station data to guide our model choices to achieve a more realistic seasonal cycle in snow depth and density.



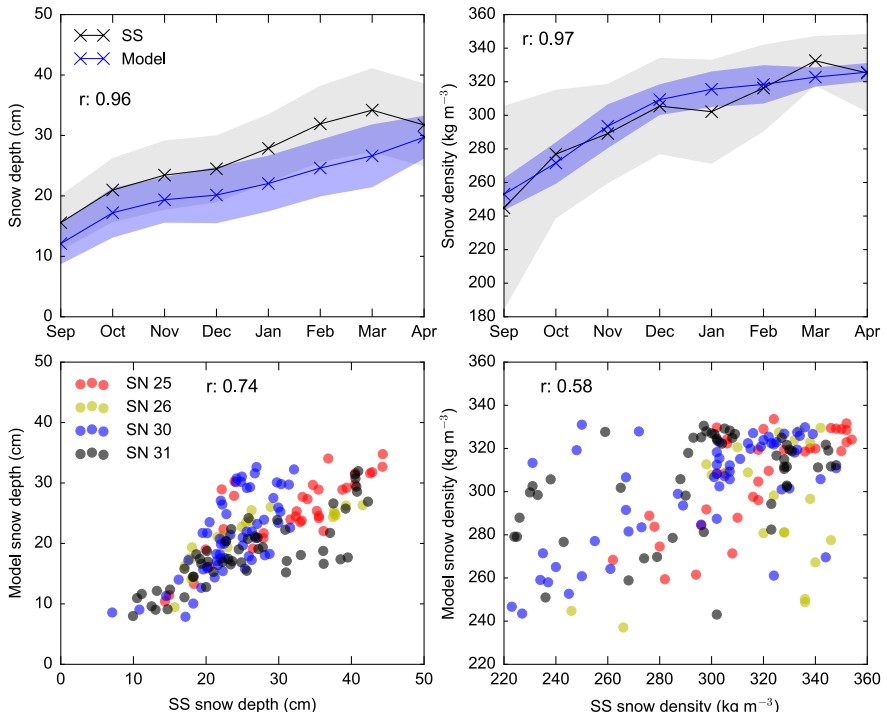

**Figure 3**: Comparison of NESOSIM snow depth (left) and snow density (right) data with drifting station data collected between 1981 and 1991. The top panels show the mean seasonal evolution of the snow depth and density for the model (blue) and Soviet station data (black), with the data binned into the different months the data were collected. The shaded area represents one standard deviation from the annual monthly mean. The bottom panels show scatter plots of all points for which there were temporal crossovers. The r-values indicate the correlation coefficient, while the colors indicate the different stations that collected the data. The NESOSIM data are from the default/ERA-I model configuration.

In Figure 3 we show comparisons of our NESOSIM results using the default model configuration (summarized in Table 1) and ERA-I snowfall forcing with the drifting station snow depth and density data. Figure 3 shows both the mean seasonal cycle based on all drifting station data points and coincident model grid cell values over this time period binned monthly, and the correlations of snow depth and snow density for all coincident data (described in Section 3.4). Our calibrated NESOSIM results capture extremely well the mean seasonal drifting station snow depth ($r = 0.96$ with a low bias of ~3 to 7 cm) and snow density ($r = 0.97$, no significant seasonal bias). The large spread in the in-situ snow density in September-October is due to the survival of snow through the summer melt season (high density) and recent autumn snowfall (low density). The correlations between the



raw drifting station data and NESOSIM snow depths are lower, but still strong ($r = 0.74$), while the snow density

correlation strength is more moderate ($r = 0.58$), suggesting the model may be better capturing regional

variability in snow depth over snow density. It should be noted, however, that snow density is highly variable in

space and subject to large measurement uncertainties when collected in situ (Sturm, 2009). In general, the

moderate/high correlations and seasonal comparisons provide confidence in the utility of NESOSIM for

estimating snow depths across the Arctic.

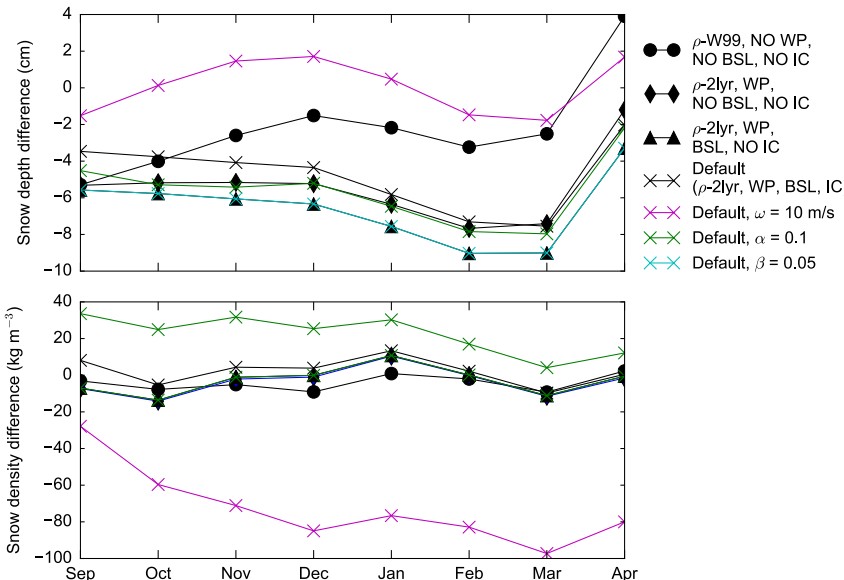

**Figure 4**: Differences between the mean (1980-1991) seasonal cycles in the drifting station data against various
configurations of NESOSIM. The different symbols represent different levels of model sophistication, ρ-W99:
climatological Warren snow density, ρ-2lyr: default prognostic two-layer snow density, WP: wind packing
parameterization, BSL: blowing snow loss parameterization, IC: initial conditions. The different colours then
represent a doubling of individual model parameters, with all other settings fixed to the default settings (see Table
1). The black crosses/line represents the default/ERA-I results (as shown in Figure 3).

In Figure 4, we highlight the sensitivity of NESOSIM to the chosen model configuration/sophistication, broadly

representing the heuristic model tuning that was undertaken. First we tested the results of NESOSIM with

different combinations of the various model parameterizations included.  Note that as discussed at the end of

Section 2, when we turn off the wind-packing parameterization the model essentially becomes a one layer model

so we use a fixed Warren et al., (1999) seasonal snow density climatology (constant density value across the



Arctic). As this is based on the same drifting station data we compare our results to, it is perhaps unsurprising that this configuration provides a better match with the seasonal drifting station snow depth cycle, including deeper snow depths (and reduced low snow depth bias) from November to April. We chose to develop NESOSIM to allow for the production of snow depths that agree well with the old drifting station snow climatology, but able to also respond to the expected interannual variability and trends in Arctic climate over recent decades, hence the decision to develop and include more advanced density parameterizations.

Including the blowing snow loss parameterization resulted in slightly lower snow depths (~2 cm) but no significant change in snow density. As the drifting station data are collected primarily within the Central Arctic, it was expected that including blowing snow loss would not result in significant differences, as this parameterization is expected to provide more of an impact in the marginal ice zone, where unfortunately in-situ snow depth data are lacking. Including the initial snow depths resulted in a small increase in snow depth and density, especially earlier in the seasonal cycle, as expected, reducing the low bias compared to the drifting station data. The seasonal correlations were similarly high across these model configurations, highlighting the primary role of the model configuration choices in determining the general bias of the seasonal snow depth/density cycle.

As a simple demonstration of the sensitivity of the model to the relatively unconstrained model parameters introduced in NESOSIM (the wind packing threshold, $\omega$, the wind packing coefficient, $\alpha$, the blowing snow loss coefficient, $\beta$), Figure 4 also shows results from NESOSIM with these three model parameters individually doubled (based on the default/ERA-I configuration). Doubling the wind packing threshold, $\omega$, (from 5 to 10 m/s) has a large impact on both the snow depth and density. By essentially reducing the likelihood for wind packing to occur, the snow accumulates and remains in the fresher 'new' snow layer for longer, significantly reducing the bulk snow density and increasing the seasonal snow depths. While this does produce snow depths that appear to agree well with the drifting station data, the low bias in the seasonal snow density suggest this is unphysical. Doubling the wind packing coefficient, $\alpha$, (from 0.05 to 0.1) has broadly the opposite effect, as expected, reducing the snow depths by increasing the transfer of snow from the fresher 'new' snow layer to the denser 'old' snow layer. Doubling the blowing snow loss coefficient, $\beta$, (from 0.025 to 0.05) has a negligible impact, again likely due to the location of the in-situ data away from the lower concentration ice regimes where this process is more significant.



As stated earlier, the differences in spatial scales and data coverage (time and space) make interpreting these comparisons/calibrations challenging. Specific model configurations may be required based on user demands, and our expectations is for these calibrations to evolve as new calibration data are made available and physical parameterizations introduced/updated. Note that we also compared the simulations of NESOSIM forced by the
MERRA and JRA-55 snowfall data (Figures S2 and S3 provided in the Supplementary Information). In general the seasonal correlations with the drifting station data were similar to the ERA-I results, but the correlations of the raw data were slightly lower for JRA-55 ($r = 0.69$ for snow depth and $r = 0.58$ for snow density) and significantly lower for MERRA ($r = 0.44$ for snow depth and $r = 0.57$ for snow density).

As discussed in Section 3.1, we also produced a synthesis snowfall dataset (MEDIAN-SF) using the median
snowfall across the gridded ERA-I, JRA-55 and MERRA datasets. The MEDIAN-SF forced results are similar to the ERA-I results, in general, and show correlations similar to ERA-I and JRA-55 ($r = 0.68$ for snow depth and $r = 0.58$ for snow density). The MEDIAN-SF seasonal snow depths have a reduced low bias compared to the ERA-I results, although this difference is small. For the rest of this analysis we choose to mainly focus on the MEDIAN-SF forced results using the default configuration (Table 1) as we expect these results to be less prone
to errors in the individual reanalyses and more reliable in regions/periods of challenging (e.g. heavy) snowfall. We provide a further assessment of the impact of the snowfall data in the following regional analysis and when we analyze the regional distributions across the more recent (2000-2015) time period.

### 4.1 Regional analysis

Here we provide a more detailed assessment of the regional NESOSIM results during this early Soviet station
period (1980-1991). We focus our analysis on the Arctic Ocean (AO, everything north of 60 $^{\circ}$N) and three specific regions that were chosen to represent different components of the Arctic sea ice/climate system: (i) the Central Arctic (CA, captures the thicker/multi-year ice over north of Greenland), (ii) the Eastern Arctic (EA, the increasingly first-year ice dominated sea ice regime), (iii) North Atlantic (NA, a region influenced by the transpolar ice drift and the North Atlantic storm track), as shown in Figure 5.





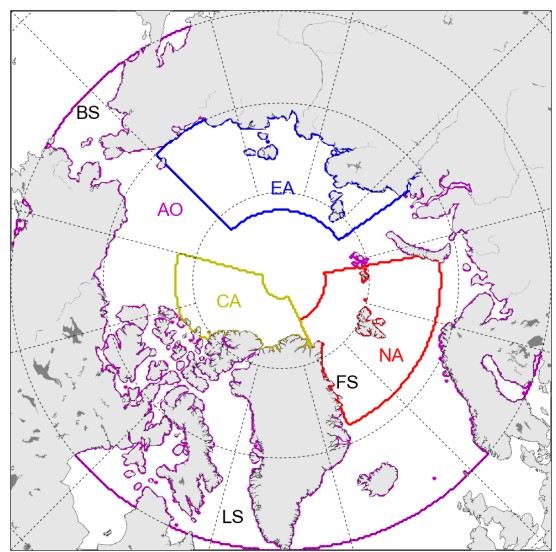

**Figure 5**: Map of the Arctic model domain and regions used in this study: AO: Arctic Ocean, CA: Central Arctic, EA: Eastern Arctic, NA: North Atlantic. BS: Bering Sea, LS: Labrador Sea are peripheral seas discussed in the manuscript.

5   Figure 6 shows the seasonal snow depth and density evolution across our four study regions for the 1980-1991 time period, using the default/MEDIAN-SF configuration (Table 1). The AO and CA region especially show strong initial increases in snow depth through fall (August to October) with the snow depth increasing at a slower rate from November to May, which is in good agreement with the W99 climatology. The EA and NA regions show a more uniform increase in snow depth from August to April. The NA region shows more daily snow depth

10   variability, which was expected due to the strong ice drifts and the location of the NA storm track where passing cyclones can deposit large quantities of snow in a short period of time. By May 1st the mean snow depths (and interannual variability, calculated as one standard deviation of the annual values) are given as: 29.8 +/- 2.2 cm (AO), 32.6 +/- 5.2 cm (CA), 27.3 +/- 2.8 cm (EA), 40.7 +/- 6.9 cm (NA).

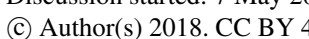



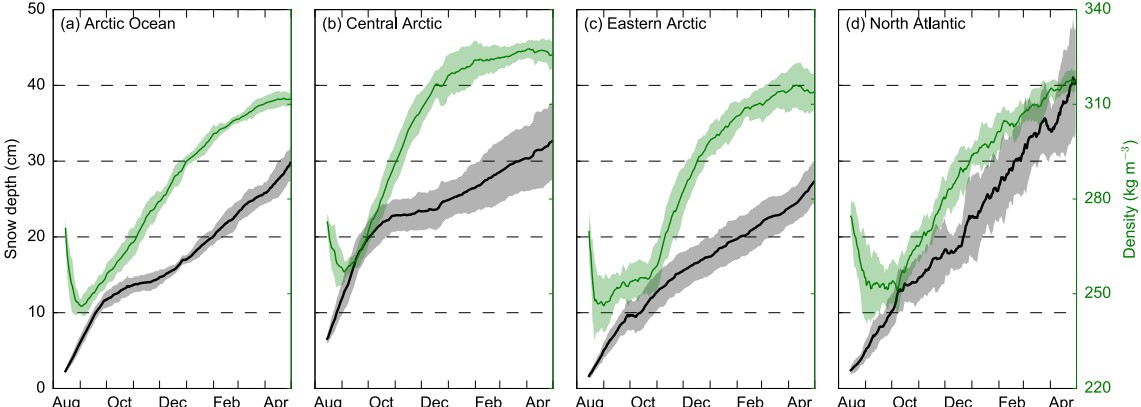

**Figure 6**: Seasonal snow depth (black) and bulk density (green) evolution across the four study regions (shown in Figure 3), initiated from August 15[th] 1980-1990 and run until May 1[st] of the following year using the MEDIAN-SF/parameter settings (Table 1). The thick lines show the mean values over this time period, while the shaded areas represent the interannual variability (one standard deviation).

We see stronger increases in the bulk snow density through fall across all regions (also shown in Figure 4), with this density increase slowing through winter/spring, especially in the CA region, after December. The AO, CA and NA regions also show an interesting initial decrease in snow density, which is driven by the accumulation of new snow (with a lower density) compared to the equal mix of old and new snow densities included in our initial conditions. The mean bulk snow densities as of May 1st are given as: $312 +/- 2$ kg/m$^3$ (AO), $326 +/- 4$ kg/m$^3$ (CA), $314 +/- 6$ kg/m$^3$ (EA), $318 +/- 3$ kg/m$^3$ (NA).

In Figure 7 we show the seasonal/regional snow depths from NESOSIM forced by the four different reanalysis-derived snowfall estimates (ERA-I, JRA-55, MERRA and MEDIAN-SF), as described in Section 3.1. In general, the results show significant differences in the seasonal snow depths across all regions (up to ~10 cm across all regions). The rankings of snow depth between the different products is broadly consistent across the four regions, with JRA-55 and MERRA producing consistently higher snow depths (except in the EA region where MERRA produces slightly higher snow depths), and ERA-I consistently lower. The MEDIAN-SF snow depths are, in general, slightly higher than the ERA-I forced snow depths. In the CA region we can see that MERRA, ERA-I and MEDIAN-SF forced results are all broadly similar, with JRA-55 significantly higher (by ~5 cm from October onwards). It is thus expected that the MEDIAN-SF snowfall data will have excluded much of the high JRA-55 snowfall data (the benefits of using a median instead of a mean snowfall). Despite the NA region having the



highest snow depths and interannual variability, the intra-reanalysis spread is similar to the other regions. The results further allude to the MEDIAN-SF dataset being a useful tool for producing estimates of snow depth considering the large uncertainty in reanalysis-derived snowfall. We further analyze the reanalysis sensitivity in the following section.

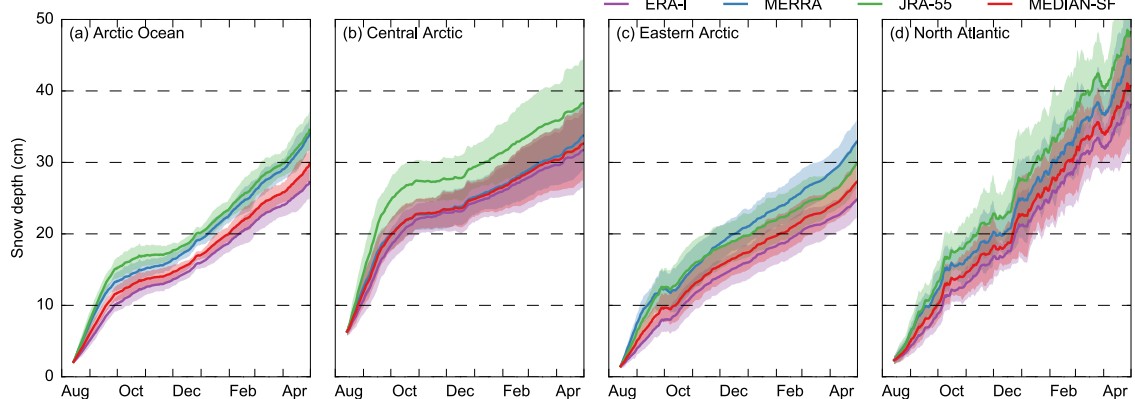

**Figure 7:** Seasonal snow depth evolution across the four study regions (shown in Figure 7), initiated from August 15th 1980-1990 and run until May 1st of the following year, forced by four different reanalysis snowfall products. The thick lines show the mean (daily) regional snow depths over this time period, while the shaded areas represent interannual variability (one standard deviation). All model runs use the default forcings/parameter settings.

## 4.2 Budget analysis

Here we discuss the relative contributions to the seasonal snow depth evolution from the various snow budget terms currently included in NESOSIM, focusing on the old time period results presented thus far. Results of the various NESOSIM budget terms and the total snow depth/volume and bulk density are shown in Figure 8 across our four study regions. The black (green) lines/shading that represent the snow depth (bulk density) are the same as the results shown in Figure 6.

Across the AO region, we see that accumulation is higher than snow depth, as expected (higher by ~25 cm by May 1st, around double the May 1st snow depth), with wind packing (~18 cm) and wind blowing snow lost to leads (~9 cm), providing significant sinks of snow. In the EA and CA region especially, the blowing snow loss term is negligible, while in the NA region it is more significant (contributes a sink of ~18 cm by May 1st). The



NA region also shows a small (~2 cm) increase in snow depth driven by snow/ice convergence and a seasonally variable change in snow depth from ice/snow advection.

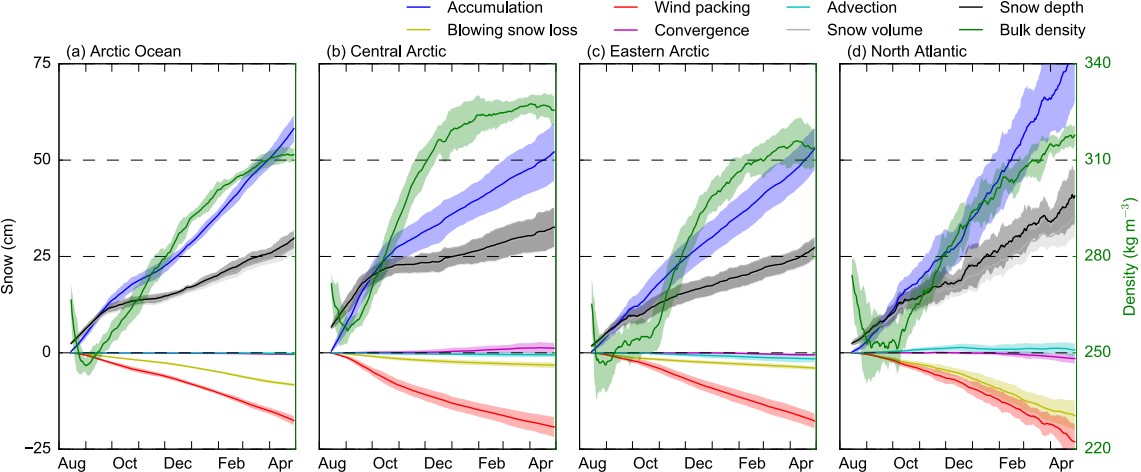

**Figure 8**: Seasonal snow budget evolution across the four study regions (shown in Figure 5), initiated from August 15[th] 1980-1990 and run until May 1st of the following year using the default/MEDIAN_SF NESOSIM simulations. The thick lines show the mean, daily, regional values over this time period, while the shaded areas represent the interannual variability (one standard deviation).

To further explore the different budget terms we also show maps of the various budget terms as of May 1st over the same 1981-1991 time period, as shown in Figure 9. The maps highlight that many of these terms, especially the ice/snow dynamics (advection and convergence), exhibit high spatial variability, which the regional means discussed previously mask. For example, the NA region shows a strong mix of positive snow advection and convergence adjacent to the coast of Svalbard (i.e. snow is drifting into the region and is constrained against the coastline), but an advection out of the region further to the north as the ice either drifts down towards Svalbard/Fram Strait or into the Central Arctic.

The bimodal ice dynamic behaviour around the pole is thought to be spurious considering interpolating issues across the pole hole in the NSIDCv3 drift product (Szanyi et al, 2016), one reason why we did not include this region in our regional analysis. In the following section we assess the sensitivity of our results to the input ice drift dataset, which will provide some further information as to the reliability of these dynamic budget terms.



**Figure 9:** Snow budget terms as of May 1$^{st}$, averaged over the 1981 to 1991 time period. The black lines show the four study regions used throughout this study. All model runs used the default forcings/parameter settings. Note the different color bar scales in panels (h) to (k).



## 4 Sensitivity studies and model validation

Here we present and analyze the NESOSIM results from 2000 to 2015, a period broadly defined as the New Arctic considering the rapid sea ice declines during this time period (e.g. Serreze and Stroeve, 2015). This period

5    also covers the temporal range of NASA's ICESat (2003 to 2008) and ESA's CryoSat-2 (2010 onwards) satellite altimetry missions, meaning the snow depth/density results presented here are planned to be of more relevance to those estimating sea ice thickness from these freeboard measurements. The period also includes temporal overlap with the ASR forcing data, and various satellite-derived ice drift products used. Results from NESOSIM forced with the MEDIAN-SF snowfall forcing and default settings from 2000 to 2015 are shown in Figure 10.

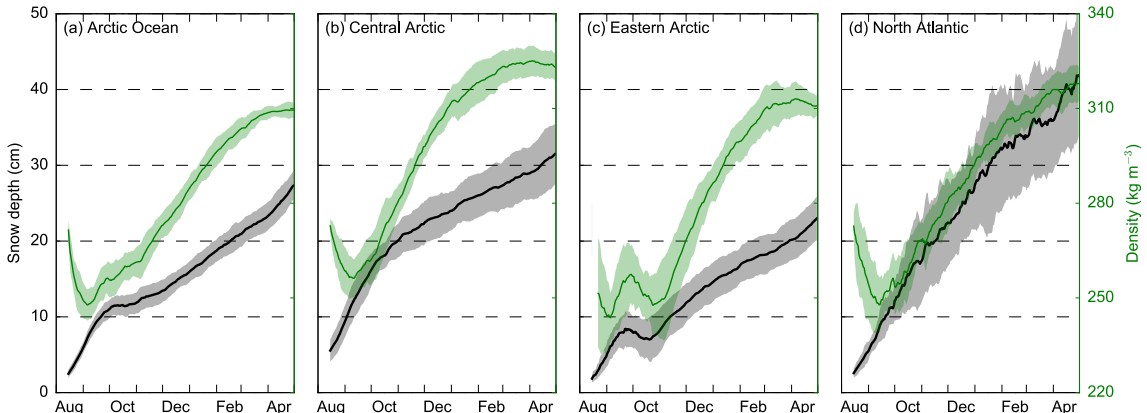

**Figure 10**: As in Figure 6 but for the simulations initiated from August 15[th] 2000-2014 and run until May 1[st] of the following year. The simulations all use the default/MEDIAN-SF configuration.

The seasonal cycles over this New Arctic period are similar to the old period, but generally feature slightly reduced snow depths. The mean (2001-2015) May 1st snow depths are: 27.3 +/- 1.9 cm (AO), 31.5 +/- 4.0 cm

15    (CA), 23.0 +/- 2.9 cm (EA), 41.8 +/- 8.1 cm (NA) while the mean May 1st bulk snow densities are: 309 +/- 2 kg/m$^3$ (AO), 323 +/- 4 kg/m$^3$ (CA), 311 +/- 3 kg/m$^3$ (EA), 318 +/- 6 kg/m$^3$ (NA). The May 1st snow depth results are summarized in Table 4, to aid comparison with the snow depths produced in the following sensitivity studies. In the CA region, we see a more gradual increase in snow depth through fall compared to the old time period, while in the EA region we see an interesting decline in snow depth through September/October, which



was not present in the old period except for the small period of constant snow depth at the start of October. While the NA region shows a similar snow depth as of May 1st between the two time periods, the new time period shows deeper snow depths in fall and winter. The results allude to strong regional variability in snow depth across the Arctic Ocean. A more detailed study accounting for differences in the input forcing data is likely

needed before any conclusions can be made regarding potential trends in seasonal Arctic snow depths, which is beyond the scope of this paper. We hope to explore trends in our simulated snow depths in future work, however.

| | May 1st snow depth (cm) | | | |
|---|---|---|---|---|
| **NESOSIM configuration** | **Arctic Ocean (AO)** | **Central Arctic (CA)** | **Eastern Arctic (EA)** | **North Atlantic (NA)** |
| 2001-2015 (ERA-I) | 25.0 (1.7) | 30.4 (4.1) | 21.7 (2.5) | 37.9 (7.3) |
| 2001-2015 (MERRA) | 30.0 (2.6) | 31.3 (3.1) | 25.5 (3.6) | 44.9 (9.0) |
| 2001-2015 (JRA-55) | 31.5 (1.9) | 37.1 (4.7) | 25.1 (3.2) | 49.5 (9.5) |
| 2001-2015 (MEDIAN-SF) | 27.3 (1.9) | 31.5 (4.0) | 23.0 (2.9) | 41.8 (8.1) |
| 2001-2012 (ASRv1) | 21.0 (1.4) | 23.3 (3.2) | 16.4 (2.7) | 36.9 (5.4) |
| 2011-2015* (MEDIAN-SF/ NSIDCv3) | 26.2 (2.2) | 32.2 (4.6) | 22.7 (3.5) | 39.2 (9.0) |
| 2011-2015* (MEDIAN-SF/OSISAF) | 25.8 (2.1) | 32.6 (3.8) | 23.2 (3.6) | 38.1 (9.5) |
| 2011-2015* (MEDIAN-SF/KIMURA) | 25.3 (2.2) | 32.7 (4.3) | 21.0 (3.5) | 38.3 (10.5) |
| 2011-2015* (MEDIAN-SF/CERSAT) | 25.9 (2.1) | 32.8 (3.9) | 23.0 (3.6) | 37.7 (9.9) |
| 2011-2015* (MEDIAN-SF/NODRIFT) | 26.8 (2.2) | 32.2 (2.9) | 24.6 (3.9) | 33.9 (10.2) |
| 2001-2015 (MEDIAN-SF, NASA Team) | 22.6 (1.7) | 27.4 (4.1) | 19.8 (2.7) | 34.1 (7.7) |

**Table 4**: Mean snow depths as of May 1st across the four study regions (rows, regions given in Figure 5) for different NESOSIM using different forcings and time periods (columns). The numbers in brackets represent interannual variability and are calculated as one standard deviation of the annual values. *Note that these 2011-
2015 ice drift sensitivity runs exclude the 2012-2013 winter season due to the lack of KIMURA drift data.

### 5.1 Reanalysis sensitivity

In Figure 11 we show the seasonal/regional snow depths from NESOSIM forced by the four different reanalysis-derived snowfall estimates over this new time period. We use the same reanalyses shown in the old time period sensitivity test (ERA-I, JRA-55, MERRA and MEDIAN-SF) but also include the ASRv1 forced results which are
available during this period, but only up to 2012. The various reanalysis forced May 1st results are summarized in Table 4.





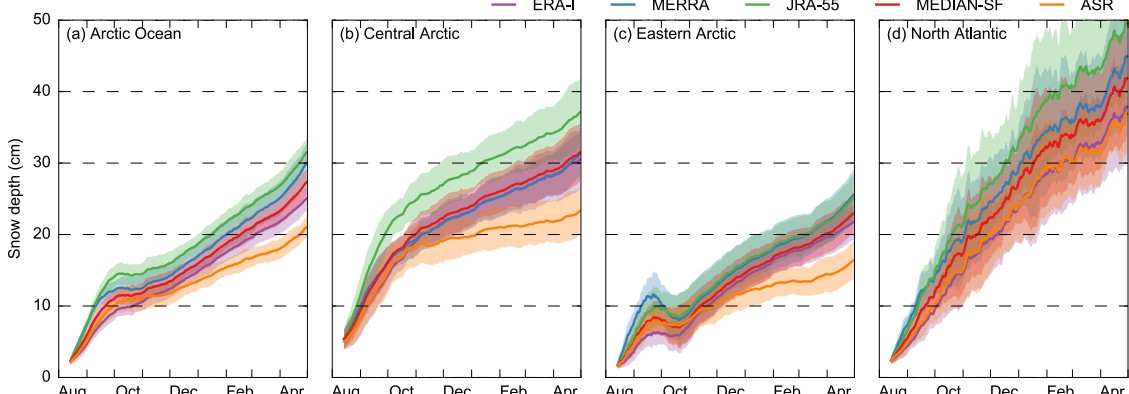

**Figure 11:** As in Figure 7 but for the simulations initiated from August 15[th] 2000-2014 and run until May 1[st] of the following year. This figure also includes results using the ASRv1 forced simulations (which are limited to Aug 15th 2000 to May 1st 2012).

5   In general, the results show similar sensitivity to the input snowfall data compared to the old time period results. The rankings of snow depth between the different products is also similar, except for the EA region, where MERRA now shows a clear high snow depth bias compared to the other forced simulations. The ASRv1 forced snow depths in the AO, CA and EA regions are significantly lower during the December-April time period, despite showing strong similarities to the other reanalysis-forced results in August to November. The ASRv1

10   results in the NA region however, are very similar to the ERA-I forced results. Note that we tested the impact of the different time periods by producing the same figure for the 2000-2012 period (not shown) which showed that the differences between ASRv1 and the other products was similar and not sensitive to this time period difference.

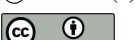



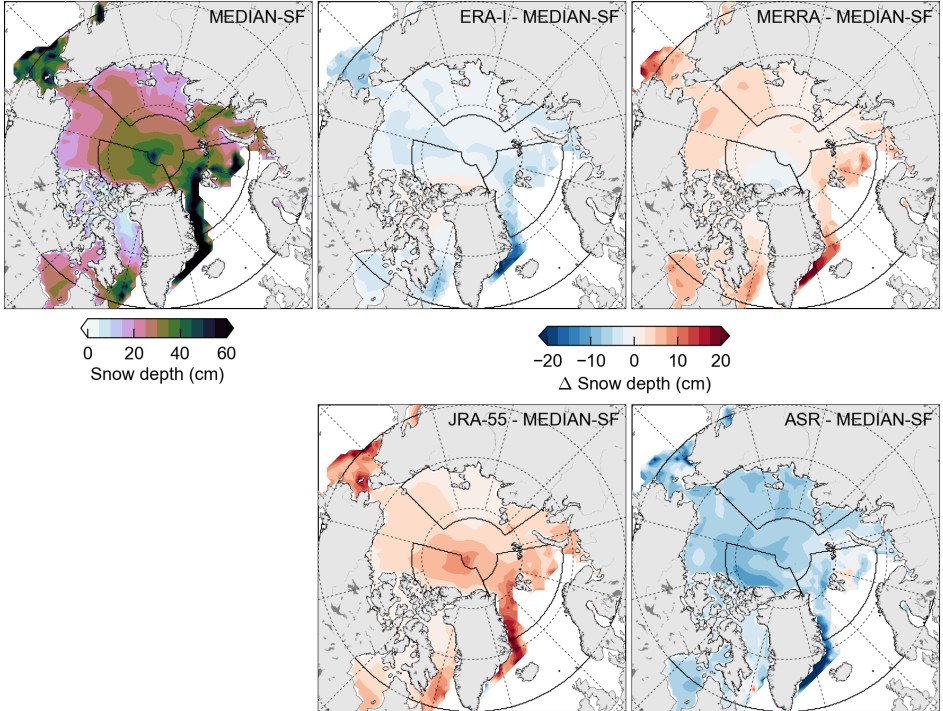

**Figure 12**: Modeled snow depth on May 1st (averaged over May 1$^{st}$ 2001 to 2015), using the MEDIAN-SF snowfall forcing (top left) and then the difference to the simulations forced by the four different snowfall products (bottom and top right). The ASRv1 forced results are limited to May 1st 2012.

5    Figure 12 shows maps of the mean snow depths on May 1$^{st}$ over the same 2001-2015 time period, for the model simulations forced by the MEDIAN-SF snowfall then the differences from this MEDIAN-SF simulation using the four individual snowfall products. The maps highlight the regional variability across the products, but consistency in the MERRA/JRA-55 (ASR) high (low) bias compared to MEDIAN-SF. The JRA-55 and MERRA forced results both show significantly higher (10-20 cm) snow depths through Bering Strait, the NA/Fram Strait region,

10  and the southern Labrador Sea. The ERA-I results show slightly lower snow depths over most of the Arctic, small increases around the Canadian Archipelago, and larger decreases in the Fram and Bering Strait region, driven by the larger differences in these regions in the MERRA/JRA-55 forcings. The magnitude of the precipitation events in Fram Strait are highly variable, due to the active storm track and the resulting difficulties of producing reliable precipitation rates during these events. As discussed in the old time period section, it is challenging to determine





from this study any particular reanalysis-derived snowfall dataset that might be more appropriate (or an obvious outlier) for producing accurate snow depth estimates across the Arctic. However the MEDIAN-SF forced results appear to provide a useful synthesis of the available snowfall data.

The regional snow budget results and May 1st budget maps using the default/MEDIAN-SF simulations are similar to the figures presented for the old time period (Figures 8 and 9) and are thus provided in the Supplementary Information (Figure S4 and S5). The noteworthy differences in the budget terms include a more significant increase in blowing snow lost to leads in the CA region and increases in convergent driven snow depth increases in the new period, although accumulation and wind packing still dominate the budget terms for both periods. The NA region also includes an interesting advection-driven reduction in snow depth in March/April that was not present in the old time period results.

### 5.2 Ice drift sensitivity

The newer time period allows us to explore the sensitivity of NESOSIM to the input satellite-derived ice drift data due to the coincident data products available during this period. Here we show results from the default/MEDIAN-SF configuration forced by four different satellite-derived ice drift products: NSIDCv3, KIMURA, CERSAT and OSISAF, as described in Section 3.2. Due to limitations in the temporal coverage of the different drift datasets, the model is only run for four years initialized from Aug 15th 2011-2015 (excluding 2012 initialized runs as KIMURA data are not available due to gaps in the AMSR-E/AMSR2 record). The regional snow depth estimates from NESOSIM forced by these four ice drift products are shown in Figure 13, with the May 1st results summarized in Table 4. In general, the ice drift sensitivity study shows a smaller spread in the mean snow depths across the different products (up to ~3.5 cm), compared with the reanalysis sensitivity study (up to ~14 cm). We also show results of NESOSIM forced with no ice drift (NODRIFT), which demonstrates that including ice drift appears not to be a crucial process for capturing the regional variability in snow depths, i.e. ice dynamics appear to be a clear second order term compared to snowfall when presented at this regional scale. The most obvious impact of ice drift is in the EA region, where the inclusion of ice drift reduces the snow depth by a few centimeters, with the magnitude depending on the ice drift product chosen (the KIMURA forced results shows the biggest decrease in this region).

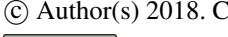



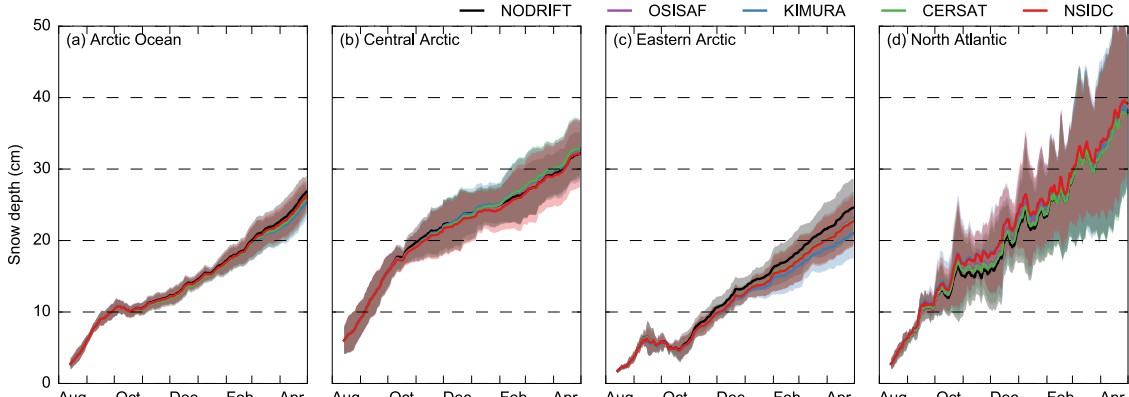

**Figure 13**: Seasonal snow depth evolution across the four study regions (shown in Figure 3), initiated in August 15th 2010, 2012, 2013, 2014 and run until May 1st of the following year, forced by four different ice drift datasets and assuming no ice drift (NODRIFT). The thick lines show the mean (daily) regional snow depths over this time period, while the shaded areas represent interannual variability (one standard deviation). All model runs use the default/MEDIAN-SF parameter settings.

Figure 14 shows maps of the snow depths averaged on May 1st over the same 2011-2015 time period, for the model simulations assuming no drift (NODRIFT) then the differences from this NODRIFT simulation using the various ice drift products. In general, the results show strong similarity in the spatial impacts of ice drift, including strong decreases in snow depth (up to ~10 cm) in the northeastern sector of the Arctic, and increases (up to ~10-20 cm) in the region directly north and west of Svalbard. There are clear differences between the different ice drift results though, with the NSIDCv3 and KIMURA forced results showing more of an impact on snow depth in the peripheral Arctic regions, e.g. strong decreases in the north and increases in the south Bering Strait, and strong increases in the Labrador Sea. This is thought to be driven primarily by the increased spatial coverage of these data compared to OSI-SAF and CERSAT, which may be masking some of the ice drift data in these regions of low ice concentration and uncertain ice drift. The maps also highlight that at more local scales, the ice dynamic contribution to snow depth variability could be significant. The data around the pole hole are also questionable in some of the products and may be related to interpolation issues across the pole hole. More specifically, the NSIDCv3 and OSISAF forced simulations show increases in snow depth at the north pole, which are not apparent in the CERSAT and KIMURA simulations, suggesting this increase is likely spurious.




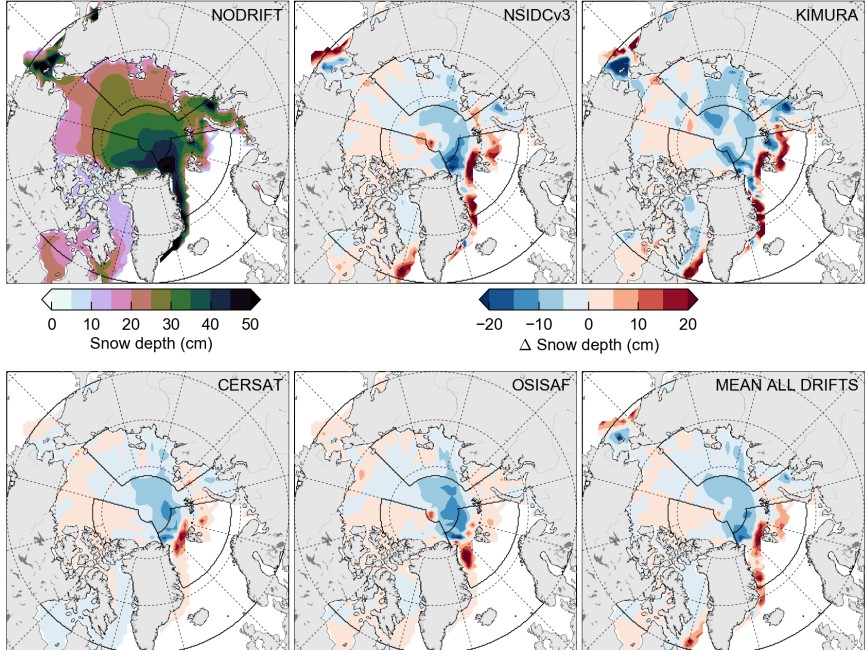

**Figure 14**: Modeled snow depth on May 1st (averaged over May $1^{st}$ 2011, 2013, 2014, 2015), assuming no ice drift (NODRIFT, top left) and then the difference to the simulations forced by the four different ice drift products and the mean snow depth from the four different forced model runs.

In general, Figures 13 and 14 suggest that the NSIDCv3 (Polar Pathfinder) forced simulations exhibit no obvious biases compared to the results using the other drift products, except for the issues of spurious snow depths within the pole hole. Note that another reason for exploring the ice drift products was to understand any potential biases if one of the near real-time products (e.g. OSI-SAF, CERSAT) were used to run NESOSIM in a near real-time framework, which does not appear to be the case.

## 5.3 Ice concentration sensitivity

Finally we present and discuss the snow depth results from NESOSIM driven by two different satellite-derived ice concentration products (Bootstrap and NASA Team) as described in Section 3.3. The regional snow depth estimates from NESOSIM forced by these two ice concentration products over the new time period are shown in Figure 15, with the May 1st results summarized in Table 4. In general, the ice concentration sensitivity study




demonstrates that the choice of ice concentration product is significant, with differences of several centimeters between the two simulations across the study regions (e.g. > 7 cm differences in the NA snow depths).

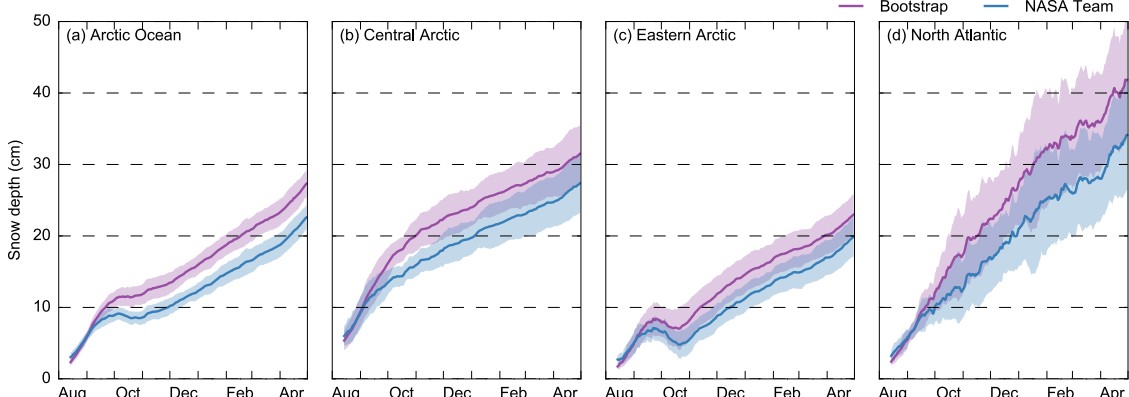

**Figure 15**: Seasonal snow depth evolution across the four study regions (shown in Figure 3), initiated from
August 15[th] 2000-2014 and run until  May 1st of the following year, forced by the Bootstrap (magenta) and NASA Team (blue) ice concentration datasets. The thick lines show the mean (daily) regional snow depths over this time period, while the shaded areas represent interannual variability (one standard deviation). All model runs use the default/MEDIAN-SF configuration.

This was somewhat expected given the known low bias in the NASA Team concentration data (e.g. Meier, 2005;
Ivanova et al., 2015), reducing the concentration of sea ice for snow to accumulate on. More specifically the Bootstrap data use daily-variable tie-points and are thus thought to improve the distinction between surface melt and open water. The lower concentrations also increase the blowing snow lost to leads term (as this is a function of the open water fraction). The snow budget terms using the NASA Team concentration data are shown in the Supplementary Information (Figure S6) to highlight this further, with all regions showing reduced snow
accumulation and blowing snow lost to leads increased, and now significant, across all regions. Again, we believe the Bootstrap data better represent the seasonal ice conditions, although we appreciate uncertainties still remain regarding the treatment of surface melt/melt ponds and their impact on snow accumulation/depth.

**5.4  Validation with Operation IceBridge data**

Here we present and discuss comparisons of our NESOSIM snow depth estimates with NASA's Operation
IceBridge spring snow depth data from 2009 to 2015, as described in Section 3.5. We first show the basin-averaged results for the various OIB snow depth products each spring (from 2009 to 2015) and the coincident



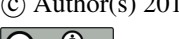

NESOSIM snow depth estimates, to assess how well NESOSIM captures the mean snow depth and expected interannual snow depth variability across this broad region of the Arctic. As discussed in Section 3.5, the OIB flights mainly cover the western Arctic sea ice pack, broadly within and to the west of the Central Arctic domain used in our earlier regional analyses, although this does vary each year. Maps of the OIB snow depth results

across the different products are given in Kwok et al., (2017).

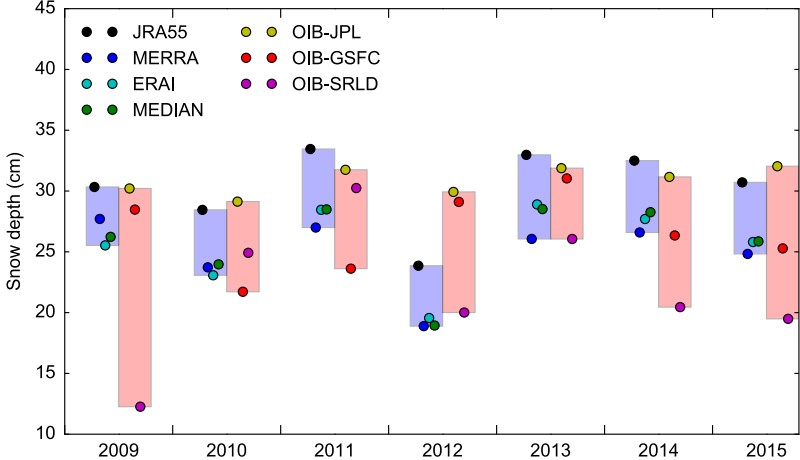

**Figure 16**: Comparisons of the annual mean snow depths from NESOSIM (default configuration) forced with different reanalyses, and the various Operation IceBridge (OIB) snow depth products. The blue (red) shading represents the annual mean spread across the different NESOSIM results (OIB products). The markers are spread
across the shaded areas to improve readability.

Figure 16 highlights the significant and variable spread in the annual mean OIB snow depth estimates (product spread of ~5 to 20 cm depending on the year), with the OIB-JPL snow depths consistently higher and less variable than the other two OIB products (SRLD and GSFC). The reanalysis-forced NESOSIM snow depths exhibit a more consistent spread of ~5 cm, with the JRA-55 forced results consistently higher than the other
reanalyses. This was expected based on our previous analyses (e.g. the Central Arctic results shown in Figure 11b). The large spread in the OIB snow depths make it challenging to assess the reliability and accuracy of our NESOSIM results. In general, however, there is broad agreement between the NESOSIM and OIB results in terms of the mean snow depths and the broad pattern of interannual variability.





**Figure 17**: Scatter plots of NASA's Operation IceBridge (OIB) snow depths from the three OIB products interpolated onto our 100 km model grid, and coincident NESOSIM/MEDIAN-SF snow depth estimates for 2009 to 2015 (a to g) and all years of data (h), including the correlation coefficient (r) and root mean squared error (RMSE).



To assess how well the model captures regional snow depth variability we show scatter plots in Figure 17 of the MEDIAN-SF-NESOSIM snow depths and the three OIB snow depth products from 2009-2015. A summary of the correlation coefficients (r) and root mean squared errors (RMSEs) across the three OIB products and NESOSIM forced by the three individual (and median) reanalysis products, are given in Table 5.

| Year | MEDIAN-SF | | | ERA-I | | | JRA-55 | | | MERRA | | |
|---|---|---|---|---|---|---|---|---|---|---|---|---|
| | SRLD | JPL | GSFC | SRLD | JPL | GSFC | SRLD | JPL | GSFC | SRLD | JPL | GSFC |
| 2009 | 0.17 | 0.16 | 0.29 | 0.26 | 0.23 | 0.36 | 0.21 | 0.18 | 0.35 | 0.08 | 0.11 | 0.20 |
| | 16 cm | 11 cm | 12 cm | 15 cm | 10 cm | 12 cm | 21 cm | 11 cm | 12 cm | 18 cm | 11 cm | 13 cm |
| 2010 | 0.36 | 0.24 | 0.37 | 0.42 | 0.36 | 0.44 | 0.40 | 0.25 | 0.40 | 0.20 | 0.05 | 0.20 |
| | 11 cm | 11 cm | 11 cm | 10 cm | 10 cm | 9 cm | 15 cm | 11 cm | 13 cm | 12 cm | 13 cm | 12 cm |
| 2011 | 0.34 | 0.25 | 0.43 | 0.53 | 0.44 | 0.59 | 0.25 | 0.17 | 0.35 | 0.09 | -0.01 | 0.23 |
| | 11 cm | 8 cm | 9 cm | 10 cm | 7 cm | 8 cm | 15 cm | 9 cm | 12 cm | 10 cm | 9 cm | 10 cm |
| 2012 | 0.72 | 0.68 | 0.72 | 0.73 | 0.70 | 0.74 | 0.70 | 0.66 | 0.71 | 0.65 | 0.61 | 0.65 |
| | 9 cm | 9 cm | 10 cm | 8 cm | 8 cm | 9 cm | 12 cm | 11 cm | 10 cm | 8 cm | 10 cm | 11 cm |
| 2013 | 0.68 | 0.66 | 0.64 | 0.72 | 0.72 | 0.68 | 0.66 | 0.63 | 0.62 | 0.66 | 0.64 | 0.63 |
| | 7 cm | 14 cm | 15 cm | 7 cm | 13 cm | 14 cm | 9 cm | 11 cm | 13 cm | 8 cm | 14 cm | 15 cm |
| 2014 | 0.64 | 0.56 | 0.64 | 0.69 | 0.63 | 0.68 | 0.61 | 0.54 | 0.62 | 0.50 | 0.42 | 0.53 |
| | 11 cm | 11 cm | 10 cm | 10 cm | 10 cm | 10 cm | 13 cm | 13 cm | 15 cm | 12 cm | 12 cm | 10 cm |
| 2015 | 0.50 | 0.42 | 0.50 | 0.58 | 0.52 | 0.55 | 0.49 | 0.41 | 0.50 | 0.30 | 0.21 | 0.35 |
| | 10 cm | 11 cm | 10 cm | 10 cm | 11 cm | 9 cm | 13 cm | 12 cm | 15 cm | 11 cm | 12 cm | 9 cm |
| All years | 0.55 | 0.51 | 0.48 | 0.61 | 0.58 | 0.54 | 0.55 | 0.50 | 0.49 | 0.42 | 0.38 | 0.39 |
| | 11 cm | 11 cm | 11 cm | 10 cm | 10 cm | 10 cm | 14 cm | 11 cm | 13 cm | 11 cm | 12 cm | 12 cm |

5 **Table 5:** Correlation coefficient (r, top rows) and root mean squared error (RMSE, bottom rows) from the correlations between the various reanalysis-forced NESOSIM results, and OIB derived snow depths. The MEDIAN-SF scatter plots are shown in Figure 17.

In general, the comparisons are highly variable and depend mainly on the chosen analysis year and the reanalysis snowfall dataset, rather than the OIB product. The correlations between the OIB snow depths and the NESOSIM

10 snow depths improve significantly in 2012 ($r$ = 0.61 to 0.74) compared to the proceeding years ($r$ = -0.01 to 0.59). The improved correlations in 2012 onwards coincide with increases in the OIB flight coverage, that include more of the Central Arctic and Beaufort/Chukchi seas, meaning the data better represent the regional variability in snow depths across the western Arctic. The strength of the correlations are highest in 2012 and 2013, while the RMSEs are lowest (< 10 cm) between 2011 and 2013, especially in the ERA-I and MEDIAN-SF



forced results. The OIB-SLRD RMSEs are generally lower than the RMSEs calculated with the other OIB products between 2010 and 2015, but significantly higher in 2009 when the signal-to-noise ratio of the earlier version of the Snow Radar used on OIB was higher (Kwok et al., 2017). The 2009 OIB snow depth results should thus be treated with caution.

The 'all years' results in Table 5 provide a summary of the correlations using all the OIB snow depths from 2009-2015. The MERRA forced results produce significantly lower correlations to the OIB snow depths ($r = 0.38$ to $0.42$) compared to the other reanalyses, while the ERA-I forced results show the highest correlations ($r = 0.58$ to $0.61$) and lowest RMSEs (10 cm). The MEDIAN-SF results show slightly lower correlations ($r = 0.48$ to $0.55$) and higher RMSEs (11 cm) compared to ERA-I.  In general, however, the moderate to strong correlations give us

confidence that NESOSIM is producing reasonable snow depth estimates across the western Arctic. The RMSEs of ~10 cm imply the expected level of accuracy in our NESOSIM snow depths, although these validations are hindered by uncertainty in the OIB snow depth observations (Kwok et al., 2017) and a lack of OIB observations in the eastern Arctic Ocean.

In the sensitivity studies presented earlier, we focused primarily on the MEDIAN-SF simulations, due to

considerations of snowfall reliability in regions of high and uncertain precipitation - e.g. the North Atlantic sector. The OIB data lack coverage in this region, however, making it hard to assess if this synthesized forcing snowfall produces more accurate snow depths in these more challenging regions of the Arctic. Data from the 2017 OIB flights into the eastern Arctic will hopefully provide some assessment of our NESOSIM snow depths in this region of the Arctic, however (the data has yet to be released). Our contemporary (New Arctic) NESOSIM

results still lack validation of the simulated snow densities, due to the lack of basin-scale density data available during this time period.

## 6 Summary

In this study we presented the newly developed NASA Eulerian Snow On Sea Ice Model (NESOSIM). The snow depth and density simulated in NESOSIM (from August 15[th] to May 1[st]) across an Arctic Ocean domain (100 km

horizontal grid) were first compared against in-situ data collected by drifting Soviet stations during the 1980s. The model produced very strong agreement with the seasonal cycles of snow depth and density and good (moderate) agreement with the regional snow depth (density) distribution. A budget analysis provided insight into the relative processes contributing to the seasonal evolution in snow depth, with snow accumulation driving



increases in snow depth, and wind packing reducing snow depth (through an increase in the bulk snow density). Blowing snow lost to leads provided a significant sink of snow, but only in the lower ice concentration, high wind/snow depth regime of the North Atlantic sector.

The model was run for a contemporary period (2000 to 2015) to produce seasonal snow depth and density estimates representative of the New Arctic climate system. The model showed strong sensitivity to the reanalysis-derived snowfall forcing data, with the MERRA/JRA-55 (ASR) derived snow depths generally higher (lower) than ERA-I. We derived a new synthesized snowfall dataset based on the median ERA-I, MERRA and JRA-55 snowfall data, to improve model reliability especially in regions of high/uncertain precipitation. The results across this newer period also allowed us to explore the sensitivity of NESOSIM to the input ice drift data, where we showed this had a second order effect compared to the choice of reanalysis snowfall forcing. The ice drift still appears to be important at smaller spatial scales, e.g. by reducing snow depths in the Eastern Arctic and driving higher snow depths north of Svalbard and within Fram Strait. We compared our NESOSIM snow depths against spring snow depths derived from data collected by NASA's Operation IceBridge (OIB) since 2009 (up to spring of 2015). Our comparisons show moderate/strong correlations for the data collected from 2012-2015, with the ERA-I and MEDIAN-SF forced results showing the best correspondence with the OIB snow depths. These encouraging comparisons provide us with some confidence in our simulated daily NESOSOM snow depth and density estimates, however we expect that further model development, testing, and validation is needed.

NESOSIM is being made available as an open source project (https://github.com/akpetty/NESOSIM), to encourage continued model development and active engagement with the snow on sea ice community. The model code is written in Python, an open source programming language (Python Software Foundation, https://www.python.org/), to better enable future community development efforts. Our hope is that the model will continue to evolve as additional snow processes are incorporated, especially as new field and remote sensing snow observations are collected and made available for calibration/validation. Obvious examples of planned future improvements include the incorporation of snow-ice formation, snow melt and rain on snow processes, which are not currently included in this initial model version, enabling the model to be run year-round.

As we look towards the launch of NASA's ICESat-2 and the production of sea ice thickness from the derived freeboard product, we must also consider potential increases in model resolution, and a better assessment of the ability of NESOSIM to capture smaller-scale (< 100 km) snow depth variability. Snow depth and density information collected during the Norwegian young sea ICE (N-ICE2015) expedition (Merkouriadi et a., 2017)



and the upcoming Multidisciplinary drifting Observatory for the Study of Arctic Climate (MOSAiC) will provide crucial insight into the importance of smaller-scale phenomena not currently included in NESOSIM, while our model results can hopefully provide useful basin-scale context to the measurements being taken.

**Model availability**

All the data processing and figure generation was carried out using the Python programming language (Python Software Foundation, https://www.python.org/). The model code, including installation details and test data, can be found on GitHub (https://github.com/akpetty/NESOSIM).

**Data availability**

A link to the model output (hosted on the NASA Cryospheric Sciences website) will be made available after

completion of peer review, along with the gridded OIB snow depths and KIMURA ice drift data.

The ERA-I snowfall and wind data were obtained through the ECWMF Meteorological Archival and Retrieval System (http://apps. ecmwf.int/datasets/data/interim_full_ daily/). The JRA-55 snowfall data were obtained through the NCEP Research Data Archive (RDA) (http://rda.ucar.edu/ datasets/ds628.0). The MERRA snowfall data were obtained through the NASA Goddard Earth Sciences Data and Information Services Center

(https://disc.sci.gsfc.nasa.gov/datasets?page=1&keywords=merra).

The sea ice concentration data were obtained through the National Snow and Ice Data Center (NSIDC), including daily NASA Team (http://nsidc.org/data/nsidc-0051) and Bootstrap (https://nsidc.org/data/nsidc-0079) data.

The NSIDCv3 Polar Pathfinder ice drift data were obtained through the NSIDC (http://nsidc.org/data/nsidc-0116). The CERSAT ice drift data were obtained from the IFREMER website (ftp://ftp.ifremer.fr/ifremer/cersat/products/gridded/psi-drift/). The OSI-SAF data were obtained through their web portal (http://osisaf.met.no/p/ice/).

**Acknowledgements**

All authors were funded for this work through the NASA-ESA Snow On Sea Ice (NESOSI) project.



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
