# Peer review of "The NASA Eulerian Snow on Sea Ice Model (NESOSIM) v1.0: Initial model development and analysis"

_Geoscientific Model Development, 2018_

## Short Comment (SC1) · 11 Jun 2018

Dear authors,

in my role as Executive editor of GMD, I would like to bring to your attention our Editorial version 1.1: http://www.geosci-model-dev.net/8/3487/2015/gmd-8-3487-2015.html This highlights some requirements of papers published in GMD, which is also available on the GMD website in the 'Manuscript Types' section: http://www.geoscientific-model-development.net/submission/manuscript_types.html In particular, please note that for your paper, the following requirement has not been met in the Discussions paper:

- "The main paper must give the model name and version number (or other unique

identifier) in the title."

Please provide in addition to the name a version number of NESOSIM in the title of your revised manuscript. Note, that a version number is important to identify the specific state of your developments.

As explained in
https://www.geoscientific-model-development.net/about/manuscript_types.html. GMD is encouraging authors to upload the program code of models (including relevant data sets) as supplement or make the code and data of the exact model version described in the paper accessible through a DOI (digital object identifier). For projects in GitHub a DOI for a released code version can easily be created using Zenodo, see https://guides.github.com/activities/citable-code/ for details.

Yours, Astrid Kerkweg

---

## Referee Comment (RC1) · Anonymous Referee #1 · 26 Jun 2018

The paper describes the development, calibration and validation, and sensitivity of a snow on sea ice model. Given that the model is likely to be used to retrieve ice thickness from CryoSat and ICESat altimetry, the model should be documented in the literature. However, the paper needs to be improved and some points clarified before it can be published. I would suggest that the authors consider reorganizing the paper to make it more readable and potentially shorter.

General Comments

While I recognize that a model cannot include everything, I am surprised that the authors do not include snow melt. I see this as a major shortcoming in the model. The top-left panel of Figure 3 shows, what might be interpreted, as a melt signal between March and April. I suspect that melt is also a factor in the North Atlantic sector. A

warming Arctic is almost certainly likely to have melt earlier. This years warming 'spike' likely caused melting in some sections of the Arctic. While this might not have resulted in a loss of snow mass, it would have increased snow density and caused a reduction in snow depth - refreezing/metamorphosis is another process. The authors should address in more detail leaving these key processes out.

Another concern is the use of the Polar Stereographic grid. The model tracks snow volume but only sea ice concentration, and not area, appears in Equation 1. Are the authors assuming that Polar Stereographic grids are equal area? This is not the case, they are conformal but not equal area. Cells at 70 N are about 10% smaller than cells at the pole. Maybe I am missing something here, and maybe this erroneous assumption might not have a big impact, but the authors should satisfy both themselves and readers that the choice of grid does not have an impact.

An obvious question, given the prevalence of Warren 1999 snow depths in sea ice thickness retreivals, is how different are the results presented here from W99? I would argue that W99 is the current benchmark for evaluation of snow depth products. The authors should include some discussion on this topic. I think it would also be useful to put the uncertainty reported in this paper ($\sim$ 10 cm) in the context of thickness retrievals. Ten centimeters is at least 30% of snow depths in the Arctic and similar to inter-annual variability in snow depth. While there is clearly room for model improvements, quality of precipitation fields and other forcing fields also come into play. It would be good to discuss these issues.

I find the model formulation as described in section and equations 1 through 8 confusing. This may be because I think of the process in a different way to that described here. Hopefully, my interpretation in the following paragraph, whether right or wrong, will help improve the model description.

I see the model as analogous to the evolution equations for ice thickness (or any other tracer), where the change in snow depth is the sum of a dynamic component (-del

*dot* (hu) [I think] not del(hu)), and a static "snow depth evolution" component that represents snow accumulation, ablation by wind, and compaction ($h^{acc}-h^{wp}+h^{bs}$) for the "new" snow layer and $h^{wp}$ for the "old" snow layer). What I find confusing is that $h^{acc}$ is the product of snow accumulation (snowfall/density) and sea ice concentration. Shouldn't what I call the sum representing the "snow depth evolution" component be multiplied by concentration (A) not just (Sf/density); e.g. A*(Sf/density $-h^{wp}+h^{bs}$) [Note my comment in the paragraph above – this only applies to a uniform grid]. Similarly, should h in –del *dot* (hu) be Ah.

Maybe this is what the authors mean by "we track snow volume" – e.g. (Ah) – and "An effective snow depth...". However, I would suggest that it is Ah that is the effective snow depth because this term represents a mean grid cell snow depth (including open water areas). Whereas, h can be thought of as a physical snow depth because it represents the process of accumulation, wind ablation and compaction at point on an ice floe. I think what I describe is a conceptual difference rather than an error in the model because $h^{wp}$ and $h^{bs}$ are tuned, so the wind packing and blowing snow coefficients can be thought of as including sea ice concentration, i.e. you can change the model description in the paper without changing the model.

Two further issues are with model calibration and the use of the ensemble mean snowfall. If I understand correctly, the parameters are tuned using ERA-Interim snowfall and these "best" parameters are applied to model runs with MERRA, JRA55 and the Ensemble Snowfall. I would suggest this is the wrong approach. The calibration process will compensate for biases in ERA-Interim snowfall. However, biases in the other reanalyses are different. A different "best" parameter set should be expected for each reanalysis snowfall product. Conceptually, model equations, parameters and forcing data are all part of the Model. Using parameters obtained for ERA-Interim, might have detrimental effects on snow depths when other reanalyses are used. I would recommend that the MERRA, JRA55 and (maybe) ensemble runs should be calibrated separately.

With regards to the Ensemble snowfall, the assumption behind an ensemble average being a better estimate is that individual ensemble members bracket "reality". Is this the case with reanalysis snowfall? If all ensemble members are biased in one direction, the ensemble average will also be biased. My understanding is that both ERA-Interim and MERRA precipitation are both biased high compared to land stations. Based on Fig 11, JRA55 is also high. So is the ensemble snowfall an improvement over the individual ensemble members?

There needs to be more detail about the ensemble snowfall was generated. For example, I can envisage ERA-Interim, MERRA and JRA55 all having snow but the location of this event being shifted by one or two grid cells. On one side of the event, while ERA-Interim might have no snow, MERRA and JRA55 do have snow. By taking the median, snowfall from MERRA or JRA55, would go into the ensemble product. On the other side of the event, MERRA might not have snow but ERA-Interim and JRA55 do, so snowfall would go into the ensemble product. This would result in a larger region receiving snow. How do you deal with that situation.

With respect to the flow and structure of the paper, sections 4 and 5 seem repetitive, especially where model sensitivity to reanalyses is discussed. Essentially, sensitivity analyses for both periods give the same results. It makes sense (to me) that if you are going to compare the two time periods then the discussion and plots are merged. This would reduce repetition, make it easier for readers to compare the two time periods, and maybe shorten the paper.

Many of the figures could be improved. In many figures, the colors are not sufficiently distinct, dark purple and dark blue. This is the case with figures where dots are used. Maybe get rid of the black borders to the symbols. Also use different symbols. For many of the line graphs, increasing the weight of lines in the legend and in plots would help. Also consider whether or not you need to show the spread. The overlapping shading obscures the lines showing the means. Using shading works for two, or possibly three, series, especially if they are separated, but it starts to detract from a plot and not convey

the information you want it to with more series. For example, the North Atlantic plot in Figure 11: I can't see the lower limit of the JRA55 spread or the upper limit of ASR. In many cases the spread is not discussed in the text. If you still need or want to show spread, you could just show May 1 snow depth spread as vertical bars off to the right hand side of each panel. The issue of including plots in figures but not describing the plots in the text occurs in several figures (e.g. Fig 9). I would suggest that if it not discussed, then don't include it.

Specific comments

L3, P5. "...to avoid complexity of snow melt processes". As I note in General Comments, Fig 3 shows what could be interpreted as melt. I would like to see more justification. Furthermore, a simple temperature index approach could have been used to account for melt.

Equation 1. See General Comments.

Equation 3. Should this be $-\nabla \cdot (hu)$

Equation 4. Should the divergence be $-h\nabla \cdot u$

Equation 5. You have a wind speed threshold for wind packing but not for blowing snow. Why is this? Studies for prairie environments indicate blowing snow initiates above $\sim$ 4 m/s, which is similar to your wind compaction threshold.

Section 2.5. Suggest this section is moved to 2.1 as an introduction to the modelling framework. This sets up the discussion of the parameterizations of the accumulation and sink terms.

L12, P11. An advantage of reanalyses is that they produce consistent outputs. Mixing and matching fields from different reanalyses breaks this consistency. How similar are the ERA-Interim winds to MERRA and JRA55.

L17, P11. "We linearly interpolate..." Do you mean bilinear interpolation? See General

Comments. This needs more detail.

L13, P14. Are significant amounts of snow in summer likely to be present in recent years? The data in Warren 1999 is 30 years old at a minimum. Are there observations from N-ICE or other field campaigns to justify non-zero initial snow depths. Furthermore, how do you initialize new sea ice? This needs to be explained.

L24, P14. "The snow depth is distributed evenly over the old and new snow layers...". Is there a reason why initial snow depth was not just assumed to be dense old snow.

L19, P16. "We carried out initial model calibration..." For this study, you only calibrated the model once, right? Suggest drop "initial" throughout this discussion unless multiple calibrations were made.

L22, P16. "...calibration involved manually tuning NESOSIM to improve the general fit..." Was this fit judged "by eye" or was some metric used? Also were all years 1980 to 1991 used, or did you leave a year out for validation during this period. While I recognize that you validated for 2000 to 2015 using OIB data, measurement accuracy and conditions might be different between the two periods.

L14, P17. It looks as if there is a larger difference in snow depth between January and March. Modelled snow depths gradually increase, while observed depths appear to increase in accumulation rate. Is this a shortcoming of the snowfall products. Also, you should mention the decrease in depths in April that could relate to melting and or compaction.

L2, P18. It is difficult to believe a correlation of 0.6 for density given the spread of points in the plot.

L7, P19. "Including the blowing snow loss... but no significant change in snow density." My first thought here, is why expect any change in density? The only mechanism by which density can be influenced by the blowing snow parameterization is a reduction in the "new sow" depth. So how deep is this "new snow" layer and how quickly does it

get redistributed to the "old snow" layer?

L7 to 15, P19. Maybe add that blowing snow loss in the central Arctic are small because sea ice concentration is close to 100%.

Section 4.2 and Figure 8. I am struggling to make sense of this section. I think part of the problem is that evolution terms are shown as cumulative, which makes comparison difficult: a big snow storm could deposit several 10's of centimeters of snow, dominating the snow depth for the rest of the season. I think you can compare the magnitudes of the terms at the end of the season (May 1) (as you do in the text) but not during the season. To compare terms during the season, I think you need to compare the timestep change in each component. The comparison is not helped by the fact that it is very difficult to distinguish lines in Figure 8. The lines in the legend need to be thicker. I would suggest leaving snow volume out of the figure.

L12, P24. Prefer "advected" to "drifting". For snow, drifting implies blowing snow.

Figure 9. I would suggest showing only the evolution terms that you discuss in section 4.2. Other plots can be put in supplementary figures. While I suggest you don't show the snow volume, note that the units are a depth. Moreover, I think Ah_s (sea ice concentration * snow depth) is better thought of as a gridcell mean thickness.

L13, P26. Prefer "Soviet Station" to "old" period.

L13, P26. Given the spread in snow depths in the "New Arctic" and "Soviet Station" periods, are they really that different?

L8, P29. Maybe use "difference" instead of "bias" as you have no "truth".

L2, P30. See General Comments. If Median-SF is biased it might not be that useful.

L21, P30. "regional variability" – suggest "regional scale". Regional scale is contrasted with Pan-Arctic Scale.

Figure 17. Why does NESOSIM have zero snow depths but OIB has non-zero snow

depths. It is difficult to interpret the panels with the OIB datasets overlayed. Maybe just show All-years but with separate panels for SRLD, JPL and GSFC OIB products. The individual years can be included as supplementary figures and/or discussed in the text. Also maybe use dots rather than x's to avoid symbols overlapping.

Technical Comments

Abstract, L9, P1. "Several simple parameterizations to represent key sources and sinks". The number of processes is not large, so you might as well list them explicitly, rather than keeping the reader guessing :).

L22, P3. Suggest "availability" rather than "presence".

L12, P4. "(Show later)" Give a figure number.

L14, P4. "Ice drift". Suggest "Ice Motion" to avoid confusion with drifting snow.

L8, P5. "...our reanalysis data..." Suggest "...reanalysis fields...".

Table 1. Add symbols for snow densities.

Figure 3. As snow depths and densities are binned, could the data in the upper panels be shown as "box and whiskers" or just boxes. That way readers can see the amount of overlap between depth and density estimates. I suggest you spell out Soviet Stations in the figures. Add 1:1 lines on the lower panels. It would be nice to have a single symbol in the legend.

Figure 4. Does No Initial Conditions (NO IC) mean the model was initialized with 0 cm snow depth?

L1, P26. Shouldn't this be Section 5?

L13-14, P29. Reference needed.

Figure 16. Why two symbols in the legend. Also the colors are difficult to distinguish. Maybe no black border on symbols. Also use different symbols.

---

## Referee Comment (RC2) · Anonymous Referee #2 · 9 Jul 2018

This clearly written manuscripts provides a detailed description and exploration of the new NESOSIM snow model. NESOSIM produces gridded, daily snow thicknesses and densities for Arctic ocean sea ice during the accumulation season (defined as mid-August through April) given daily inputs of Arctic wide snowfall, sea ice concentration, ice drift, and near surface winds. Although the Arctic melt season may extend well into September, the model does not include thermodynamic or radiative processes, and this certainly limits its utility. Rather, the emphasis here is on the impacts of wind via wind packing and blowing snow loss to leads/open water. The parametrizations are fairly simple – winds exceeding a threshold can only decrease snow thickness and increase snow density. There are no snow drifts, for example, or sub-grid regions of bare ice which are present in other models. In addition, there are no snow-aging

processes that may contribute to density changes. Still, the authors do a commendable job validating their model against observations and do a thorough evaluation of model sensitivity to the various snowfall reanalysis, ice drift and ice concentration products. This latter analysis highlights the true utility of the model – a simple framework for the inter-comparison of reanalysis-derived snow on sea ice data products.

Some specific scientific comments: The authors need to better place the work in scientific context and show how the work is unique. How is this an improvement over the simple models of snow depth forced from reanalyses? There are more complex snow on sea ice models (Lecomte-LIM, Liston-SnowModel, Hunke-CICE) which include some of the same processes (ice drift, dynamics, precipitation) yet rather than develop wind loss and compaction include some distinctly different processes (thermodynamics, radiative properties, snow ice formation, dune formation, ridge accumulation...). Are these models missing the "key sources and sinks"? There is also Dery and Tremblay (2004, JPO) that specifically looks at the effects of wind redistribution with an explicit mass flux into leads. Is your approach better? More useful? Consistent?

What is the impact of excluding thermodynamic processes on your results? Does this change your conclusions about the impact of wind processes?

Some misleading statements: First sentence of the abstract. ... "produces daily estimates of depth and density of snow across the polar oceans". Not yet because of some important missing processes. Qualify with Arctic only and during the accumulation season. Using old vs new snow in the text and figures. It's clear that there is intention to one day include snow aging, but for now there is only fresh vs compaction. The depth hoar densities of 150-250 is never used in the model even though paragraph 10 seems to suggest that it is. The old snow value is 350 kg/m3 which is not the average of the higher end of wind slab and depth hoar (325 kg/m3) but rather the average of the wind slab bounds.

Perhaps future developments could be kept to a specific section to better clarify what

the model does and doesn't do.

Snow density in NEOSIM is bounded by the two chosen snow density parameters (200 and 350 kg /m3) even though the observations referenced give values for dry snow of 150 and wind slab ∼400 kg /m3 on average. Why exclude these possibilities at the outset? Instead of using an average value, doesn't it make more sense to use the upper and lower bounds given the nature of the parameterization? How sensitive is the model to these values?

The late summer initial conditions integrate all the missing snow melt processes and for that reason, they are rather important. The paragraph on page 14 does a fairly good job motivating your approach, but it would be clearer if you showed the equations for hs(0) and hs(1) after summer melt. Also better explain how snowfall events factor into this parameterization and explain why keeping the same fraction for fresh/compacted snow is the right approach (or clarify if you do something different). It would also be informative to see the Aug 15 values in your figures 3 and 4. Are there Aug observations to help validate the IC parameterization and fig 2 in particular?

Why absorb the timestep in the model equations? In (2) the parameter alpha has a timestep dependence that isn't explicitly called out and as a result, 0.05 is less meaningful. Better to define an alpha with units of per second.

Are the differences between simulations with different snowfall estimates larger than the differences between time periods? Are the time period differences significant?

Fig 14 seems to suggest that ice drift is actually quite important but masked by basin or large regional averaging. Magnitudes of the differences are similar to the snowfall sensitivity. Impacts are near the ice edge (increase ice retreat?) and add to smaller (but still > 100 km) scale variability (potentially impacting melt-pond formation).

Technical corrections:

Table1. add the model variable in the table.

Define U in Eq. (5).

Missing ) in Eq. (6)

Why is the bs term in Eq. (7) positive?

Missing t dependence in some terms in Eq. (7) and the next equation Line 26 missing "of" in "one the better.."

Add W99 to upper panels of fig3

Add W99 to fig 6 (a)

In explanation for fig 8, NA region shows a small "decrease" due to convergence, not increase. CA shows a small increase but is not mentioned.

Fig (9). What is (b) Ocean? Change (d) wind/leads to wind loss to leads. Is there an ice area cutoff for snow depth in (k)?

Table 4 could be improved by adding the 1981-1991 time period for comparison, identifying the boxed regions as "reanalysis sensitivity", "ice drift sensitivity", "ice concentration sensitivity" and including in the description that the default configuration is MEDIAN-SF, NSIDCv3 ice drift and Bootstrap ice concentration

End of p 30. Comment on the NA region in table 4 when ice drift is included. Seems to be important here too.

Fig 13, (d) NA does not appear to be consistent with table 4. The NODRIFT value on May 1 is around 34 cm in table 4 but seems to be much higher in this figure. Explain.

Last comment before 5.3 does not seem correct. There does appear to be bias in the real-time products with respect to the peripheral seas .

P 37. What are uncertainties in the OIB observations?

Mention the results from sensitivity of ice concentration in the summary. These were interesting and significant.

---

## Referee Comment (RC3) · Anonymous Referee #3 · 11 Jul 2018

The authors present a new open source model, the NASA Eulerian Snow on Sea Ice Model, for estimating daily depth and density of snow on sea ice. The authors note at a few points in the paper that the model is being developed primarily with application to altimetry-based ice thickness determination in mind, though other applications are likely. The model is a simple representation of the snow that is largely an accounting of snowfall produced by reanalysis data, similar to prior efforts (e.g. Maksym and Markus 2008; Kwok and Cunningham, 2008), with terms for snow compaction, loss to leads, and transport on sea ice. It is Eulerian, but features pseudo transport by exchange between grid cells, features only 2 layers, and is forced with available spatially and temporally complete datasets that are known to be of limited accuracy (e.g. Reanalysis, passive microwave concentration). The model is calibrated/validated against limited

available snow on sea ice data from Operation Ice Bridge and from 1980s era Soviet drifting stations. The description of the model is complete and in this regard the model is publishable with minor revisions – but reviewer doesn't feel the model is very good or useful in its current form for its intended purpose. Reviewer focuses most of this review on highlighting its shortcomings. In fact, a possible conclusion of this data presented would be that simple treatment of snow on sea ice will not meet the accuracy levels required for altimetry applications. The reviewer encourages the early career team to put the paper aside for awhile and take the time write a model that would actually be highly used.

The reviewer feels that the key issues are that the model is excessively simplistic, not representative of known physical process (even at the level of simplicity targeted), and that its results show it is inadequate for the intended purpose. There are errors in the equations presented, many compromises appear to have been made that make accuracy and/or realism lower in favor of rapid release, and as a result the work is unlikely to have much impact as presented. The presentation in the paper is quite long, and focuses on trying to convince the reader that the model is good, rather than taking a hard look and comparing against a reasonable standard.

The development of a snow product for improving retrieval of sea ice thickness from altimetry is critical for ICESAT 2 to be useful and this team should have NASA's support to do just that. Such a snow model's accuracy goal must be based on a desired accuracy in thickness retrievals. (e.g. retrieval of ice thickness accurate to +-0.5m over a given domain demands snow depth accurate to O5cm over the same domain). The model presented here is not up to meeting these kinds of needs, and does not leverage the existing (more sophisticated) models of snow on sea ice (e.g. LIM, SnowModel, CICE).

Some major issues include: Model design relative to state of knowledge: 1. The two layers used in the two layer model (new snow and windslab) do not represent the two layers of the snowpack discussed in literature (wind slab and depth hoar). Authors

cite and discuss the literature indicating that windslab and depth hoar dominate the mass of the pack and have quite different density – then ignore these decades of observation to invent a new scheme unsupported by observations. Respecting the effort to create a simple, 2 layer model, new snow should not be one of the two layers. The references cited clearly state that new snow rarely comprises much of the Arctic snowpack, because it is very rapidly converted to windslab. The preservation of a new snow layer appeared to be designed for modeling loss of snow into leads – but little is known about the magnitude of this flux, and it was minor in this model. 2. The model is operated on a 100x100km grid, which is very coarse relative to the variability in ice – which is shown to be important in impacting the accumulation of snow. The data sets used provide much higher ice concentration, and movement information – this data should be used at full resolution and atmospheric data can be downsampled. 3. Melt is neglected despite it being important during part of the timeframe and having significant impact on results.

Quality of the Model Results and Characterization thereof 1. Validation shown indicates the model produces results that do not capture the variability in observed snow depth or density reliably. Authors focus on averages of model output over decadal timeframes, which can be made to match observations by tuning of the arbitrary, non-physical constants in the model. This focus fails to acknowledge the inability of the model to capture interannual or spatial variability. 2. Prediction intervals are not provided, but scatter plots show little relationship between individual observations of snow depth and modeled snow depth. No discussion is provided of how these errors would propagate in the intended use (altimetry retrievals of ice thickness) but it appears errors are sufficient to radically alter retrievals of depth and appear to indicate the data would not be useful for altimetry retrievals of ice thickness from ICESAT2. Authors fail to acknowledge any of these shortcomings and go to great pains to make the results appear good. 3. Modeled variability in density appears to have very little relationship to observations. 4. Comparison with the southern ocean, are pushed to a future effort, but validation statements in the paper suggest the model applies to 'polar oceans'. 5.

Results from the median of the three reanalysis products are declared 'better' repeatedly with no reasonable support. Taking the median of atmospheric reanalysis models would result in nonphysical jumps between atmospheric states and the removal of extreme events from the record, and is challenging to support physically.

DETAILED COMMENTS Page 1, line 16. "very strong agreement" Delete "very strong"

Page 1 line 22 descriptions of agreement too subjective. The use here is altimetry. Tell the reader about the error in estimates implied.

Page 2 line 5-8. Poorly worded sentence. Consider modifying. One suggestion is: The altimetry technique involves measurements of freeboard, the extension of sea ice or snow surface above a local sea level. Estimates of snow depth are required to derive sea ice thickness from either snow surface freeboard or ice freeboard, because snow depresses ice freeboard and adds to snow surface freeboard. Snow depth is one of the primary sources of uncertainty for both laser and radar altimetry (e.g. Giles et al., 2007).

Page 2 line 10. Replace 'lacking' with something more descriptive/accurate (they aren't lacking they are just not complete/good enough).

Page 2 line 22-24. The sea ice community often relies on simple models of snow depth forced by reanalyses – please clarify how this is different. To the reader, it still looks like a simple model forced by reanalyses!

P 3 Line 16 "and two snow layers to broadly represent the evolution of both old/compacted snow and new/fresh snow." The assignment of the two layers in this two layer model is not consistent with the widespread understanding of the primary two layers on sea ice as depth hoar and windslab. New snow is occasionally present but usually rapidly transformed to windslab. It may be an acceptable third layer. See many of the snow on sea ice references cited here, such as Sturm et al., 2002 – generally the snow is treated in these two layers. The author's choice here to take the

two layers to represent layers that the extensive literature reviewed does not discuss is perplexing.

P2 line 18 replace "detailed" with "iterative". The simplified scheme does not permit a 'detailed' assessment of connection between input data and snow depth given its lack of physical complexity – it permits an easier iteration of possibilities.

P4 line 13 Input data from passive microwave higher resolution than 100x100km, even if atmospheric data is not. Since ice concentration is so important, reviewer questions if 100km resolution is adequate. Further - does observed snow depth vary over 100km resolution? Since this is the motivation, what resolution is needed for useful for altimetry based determination of sea ice thickness?

Page 4 line 14 add "from reanalysis data" after the word 'drift'.

Page 4 line 16 – (volume of snow per unit grid cell in units of meters) – doesn't make sense volume is meters cubed. Throughout the treatment of snow varies between depth and volume freely, but this free transition between volume and depth is challenged for some considerations of snow – particularly convergence/divergences. Since the goal here is to understand depth for altimetry retrieval, a convergence, which moves volume into a cell, is not the same as a change in depth.

Page 5 table one – put formal references to data sources, e.g. "bootstrap" is not sufficient.

Page 5 delete "snow pit and density data. . . helped guide. . . parameterization . . . seasonal evolution." There is no prescribed seasonal evolution of density, use of snow pit data etc. in this model. Two constant snow densities are selected and declared. This sentence obfuscates the very simple, non-experimentally supported nature of the scheme.

Page 6 line 8 replace bulk density with mass.

Page 6 – here authors note that the community of snow science experts and prior

literature they have created generally group the snow into two layers (wind slab and depth hoar). They further note substantial differences observed in density of these two layers, and that these two layers comprise the majority of the snowpack. Not noted, but available in the literature is data showing that the contribution of the two layers to the overall snowpack varies from the approximately 50-50% contribution seen at SHEBA. So it seems windslab and depth hoar are the two layers to model. But. . . these two layers are different than the layers the authors have chosen (new snow/old snow). It seems a major departure from decades of snow research is being made here and it is not being well defended. Why?

Page 6 line 12 "for this reason we use the average of higher end values of ws and dh". Reviewer sees no reason provided supporting the use of the higher end of the range of values for each of the two common layers. The mean density of each layer, multiplied by the mean fraction of each layer should provide a more representative density for the combined wind slab and depth hoar. Further, the value selected is not the average of the higher end of the range of values for each of the two common layers, leaving it unclear how it was determined.

Page 6 Line 16 "Our simple parameterization is thus expected to be generally representative" No reasonable evidence provided supports this. Statements like this are found throughout this paper. Delete or support with concrete evidence that quantifies what the range of uncertainty they will work within.

Page 6, Line 23 (default of 5m/s). Default or for the purposes of this work is it simply always set to this?

Page 6, Line 24 "determines the fraction. . . transferred. . ." Over what time? (seems that the coefficient is model timestep dependent. . . and perhaps shouldn't be)

Page 6 line 26 'Wind threshold of 5m/s was determined based on. . .' studies. Please add a description of the range of wind thresholds indicated by these studies, and why 5m/s was selected from within that range.

Page 6 Line 8 Daily gridded ice drift is still required in this Eulerian scheme, eliminating it as a reason for choosing Eulerian over lagrangian, discussed above.

Page 7, line 19. Reviewer is not aware of any evidence indicating that the loss of snow to leads in the North Atlantic sector of the Arctic is significant relative to the thick snowpack in that region. No evidence seems to be coming out of the N-ICE experiment to that effect. Some quantification of loss to leads in the Antarctic has been made by Leonard and Maksym as noted, but this was in the southern ocean. Please cite appropriate literature or delete speculation.

Page 8 line 4 – This parameterization doesn't make sense and is under supported for several reasons. 1. It appears that a constant coefficient beta is multiplied by 10m windspeed NOT by the amount which the wind speed exceeds the threshold velocity! So snow is lost to leads even when winds are too slow to move snow. 2. The amount of the snow lost to leads increases linearly with windspeed, when the drifting snow volume is well known to vary more rapidly than linearly 3. The loss to leads varies linearly with open water area, again this is likely more rapid than linear, and a thought experiment with random lead spacing/size could arrive at a better approximation. 4. The parameterization removes a fraction (2.5%) of the new snow layer to leads on each windy timestep – timestep is then important due to compounding what timestep is this defined for? 5. Is this parameterization/ value supported by any field quantification of loss to leads or is it simply made up due to lack of available observation. Either is fine, but state which it is. Page 6 line 9 – missing parenthesis on equation

Page 6 Equation 7 – appears incorrect. Change due to blowing snow is added (last term), but this should be a loss term (loss into leads). It appears that the term calculated in Eq 5 is always positive, so adding here will result in addition of snow, not loss. Similarly, how signs are handled on dynamics, convergence and divergence as well as advection depends on how (+-) ui is defined in equation 3 and 4, and this is not (but should be) specified above... so the reviewer is unsure if the sign here is handled correctly.

Page 8 line 21 August is mid- late summer. Change "early" to 'late' or delete.

Page 9 line 2-3 Do these melt events invalidate the results here? Is this model useful before these 'hoped for' additions occur? It sounds like this is being hurried along.

Page 9 line 8-13 This paragraph appears to handle a specific test case, not discussed here. Seems out of place possibly a draft fragment. Unclear what tests this new density applies to, or how this test relates to the model released for community use. (update after later reading, now understand what this refers to, but still feel it was out of place and not well enough contextualized here)

Page 9 line 16 Soviet - capitalize.

P 9 line 26 one OF the

P9 – would be appropriate to acknowledge the lack of validation sites or validation data over Arctic sea ice, and uncertain accuracy of the products in that region.

P11- Taking the median of the reanalysis products is an interesting idea if one has no idea which of the different products is best, but don't authors have better information about which is doing best from the comparison studies in literature?

P14 L10 – Initial conditions the Warren climatology is quite outdated. It is good you are trying to update them somehow. Is there evidence, e.g. from current autonomous ice mass balance buoys, that snow still regularly survives summer? Can you 'calibrate' this adjustment scheme based on those observations? Would a degree-day model be better than number of melting days? Also, what category is this snow placed in? Does it have a density reflective of melting snow (i.e. 400-500 kg/m3)?

P14 L22 – explain how this is 'linearly scaled' a bit better. Provide an equation. Is the fraction by which duration of melt is different from mean simply multiplied by snow depth? Does it mean that at 2x duration no snow is left and at 0x duration 2x snow is left?

P14 L 28 – were necessary... Could this be because the model doesn't handle melt processes?

P15Fig2 – These substantial August snow depths in 2012 and 2013 should be compared against available buoy data to determine if they are reasonable. The reviewer believes they are not and that this is ultimately a nonphysical tuning mechanism that helps account for lack of melt processes and poor representation of precipitation phase at this time of year in reanlyses.

P 16 L 17 – this section is missing a clear statement of how accurate OIB data is expected to be.

P16 L 19-27 – pretty hand wavey – not rigorous.

Show plots of how snow evolves in model – what fraction of the snowpack is new snow layer vs time (it would have to be small to be realistic.)

P 17 Fig 3 – modeled data appears systemically low by about 5 cm depth. Snow density has essentially no relationship between modeled and observed. Individual year data – which is how this data would be used to derive altimetry based estimates of ice thickness – appear poor.

P17 L 13 delete "extremely" – an error of ~5cm on a snowpack of ~20cm is still a 25% error.

P18 L 1 – r of 0.74 would not generally be characterized as 'strong' P18 L2 - delete 'more' ... its just moderate. Also, what is actually suggested here is that the model is good at predicting the MEAN – because you can tune your constants to make the mean look very nice, but not very good at capturing the interannual variability that is key to getting snow depth right for altimetry. P18,L4-5 "In General, the moderate/high correlations... provide confidence..." This statement is hand waving and cheerleadery without content. Delete this statement and replace it with a statement that articulates the degree of certainty with which output of the model should be treated. Suggest

authors calculate the +-95% prediction interval for a modeled density or depth relative to this dataset. Suggest authors do this for individual locations/months on individual years, as well as mean.

P18, Figure 4 – This comparison is really just showing how well tuned the models are on average. Since the model is not presented as a mean climatology, but rather is presented as a deterministic snow product for specific locations on specific years, this comparison is inadequate.

P19 L 3-6 Here authors make an odd argument. The model does not reproduce the climatology observations as well as a single mean over the entire timeframe. They argue this is OK because the model will handle interannual variability better because of its 'more advanced' density parameterization. The density parameterization is not particularly physically realistic, however, and fails to meaningfully capture interannual variability of the climatology density data (figure 3d). Reviewer therefore finds this statement lacking.

P19 l10 – not all marginal ice zones have low concentration – clarify that low concentration areas are where greater impact is expected

P 19 L 19 – Reviewer disagrees that wind threshold velocity for blowing snow is unconstrained. Resources reviewed can establish that under all but extreme conditions (e.g. recent rain on snow) a threshold of 10 m/s is pretty high, maybe unreasonable. It would be better to range this within the values observed in the references cited earlier.

P 20 L 15-18 – Speculative. Reviewer finds no reason to believe the median should be superior in regions of heavy snowfall. Defend or delete.

P21 L 5 – again this represents the mean over the decade being presented/compared. The model performance over this timeframe is highly tunable and not the performance metric of interest to an end user taking this data as an input to altimetry – that user would want to know the prediction interval for individual or moderate size groups of

snow datapoints, and probably also whether there is any change in mean bias over time.

P 22 Fig 6 – are standard deviations in depths this low comparable to any observations? If so they suggest a single climatology would be adequate for most end uses.

P22 L 15 plurals

P23 L4 The fact that there is scatter among the reanalyses is not necessarily an argument for taking the median of them. Delete.

P23 L19 providing significant VOLUME REDUCTIONS and sinks of snow (wind packing is not a sink)

P24 L1 – Convergence really causes an increase in snow volume, not an increase in depth. The use of depth vs volume for a cell needs to be sorted out and treated consistently throughout this paper. Reviewer recalls a section way up at the top saying snow would be handled in volume throughout the paper, but has seen treatment vary.

P25 Fig 9b – unclear what "Ocean" refers to. Snowfall directly into water? Caption needs to be more descriptive and the figure subcomponents should be linked back to which equation # they represent. Fig 9f – this underscores the issue with treating convergence as a change in depth- in reality convergence/divergence of the ice at scale does not change depth in the sense that such would be used to interpret remote sensing. It changes snow volume, further, it appears the impact of convergence/divergence is noisy at best. Fig 9 I – density map warrants discussion. For example, density appears highest in central arctic (far from melt) This is likely untrue and an issue with not including melt/rain on snow processes. Very low density is indicated in marginal areas around N Greenland and in Baffin bay/Canadian islands. Can these be supported at all? Fig 9 h-k colors appear to fade toward lower value indication near land in general. Is this valid or a plotting artifact? Again volume and depth are used interchangeably and plotted on a depth scale. This needs to be resolved consistently. Page 24 L 5-7

– Authors state these measurements are explicitly for altimetry retrievals, so they must have characteristics useful for such, (more than matching seasonal evolution on average) including: 1. Capture interannual variability 2. Capture spatial variability 3. No long term bias or trends in error (that could be mistaken as trends in ice thickness). If these cannot be shown, perhaps a discussion about whether this approach is viable is needed.

Page 26 Figure 10 – "As in figure 6" This is a nice addition to convey consistency, but pls also provide full description in caption, don't make reader hunt back several pages.

Page 26 L15 Without melt processes included, what explains the loss in depth?

Page 26 L 5 Why the depths are different during the later time period IN THIS MODEL is within the scope of demonstrating a model, even if understanding why they are changing in reality is not. Please answer: Are depths less due to less ice for snow to fall on? Or due to less precip? The reader must know if the model is representing the changing Arctic – since it is calibrated on old data. This cannot reasonably be scoped out of the study.

P 27 Fig 11 – see comment on Figure 10

P 29 Fig 12 – please clarify what positive and negative deviation mean (is the product higher or lower snow than the median product)?

P 30 line 3 – it is not clear that the median provides a result any more useful than the others. One should note which product compared best to coastal stations data and any other indications from literature which might be best.

P 30 line 20 – this is not surprising and should be noted as such. Advection/convergence/divergence was much less important than snowfall in the plots above.

P30 L 23-24 – here is where the idea of snow depth vs snow volume is really important. Dynamics are perhaps not important in depth over a 100km cell average, but they are important to the DEPTH on the actual subgrid ice, since divergence creates new

ice with no snow, rather than rearranging all the snow into a gridcell average. The averaging over the 100km cell at each timestep may be particularly important in ice generating areas, where snow is continually averaged back into source regions, rather than being advected out entirely. Tracking ice classes within the cells, as is done in CICE may be critically important.

P 31 figure 13 – the drift scheme matters little over huge areas because convergence and divergence cancel. This plot is just not the right way to consider this, particularly in the context of use fore spatially distributed altimetry observations. Figure 14 suggests that the drift products don't differ that much between them in the central basin, but that having drift represented at all is very important, altering snowpack by O50% in large areas of the Arctic.

P32 L6 – There are actually substantial biases in the peripheral seas – which may not be important overall, but cannot be ignored in the statement about biases.

P33 – given the importance of concentration product, better understanding the role of changing concentration in the changing modeled snow depth above is important.

P34 – Tough to compare to observational data this noisy. Reviewer agrees they can be considered 'in agreement' within the bounds of the error of either... both of which are large. Are any of the OIB algorithms emerging as superior? Must all three be treated as equally likely?

P35 – comparison of 100km grid cells still includes substantial averaging, but already shows poor agreement. Agreement should be presented in terms of a 95 % prediction interval so user knows the capability of the method in useful terms – if the model says snow was xx OIB will say snow was xx +- yy 95% of the time.

P36 – this discussion of the comparison of the scatter plots goes to great lengths to avoid describing the obvious. The model isn't very good at reproducing variability on OIB data, and if you believe OIB snow data is in any way representative of the variability

in snow depth on ice, the modeled snow depth isn't very good at capturing spatial or interannual variability. The conclusion should then be that more sophisticated model representations are needed or that OIB data is trash. Since the model didn't agree with the Soviet drift station data scatter plot very well either, I don't think you can conclude that the model is adequate but OIB is trash.

P37 – "in general, however, the moderate to strong correlations... gives us confidence" Reviewer cussed in exasperation when reading this. This is a science paper not an opinion piece. These are not moderate to strong correlations! They clearly show NE-SOSIM cannot capture the variability observed well. Get this subjective language out of the paper and replace it with quantifications of how well the model does at both representing means (where performance is good because of tuning) and variability (where the model is not working so well). Talk about whether the model is good enough to be used in altimetry honestly and present some paths forward to getting there if it isn't.

P 37 L 19 – data is yet to be released in parenthesis? Thin its out now...

P 37 L 20 – There must be some field data available that you could at least spot check it against!

P 37, L 26 delete very strong, delete good

P 37 L 28 contributing to the MODELED seasonal evolution in snow depth

P38 L5 uncapitalize New, consider replacing with 'more recent.'

P38 L7 There is no evidence presented that this median product is better, and good reason to believe it just averages in erroneous values and non-physically jumps between atmospheric states toward limited representation of extreme events. Defend the use on scientific merit or consider deleting the median product.

P38 L10 use consistent language... it is 2nd order on mean, but first order in some regions.

P 38 L 14 "moderate/strong correlations" This statement is flatly unsupported by the results shown, and authors 'confidence' in line 16 is unfounded. The product does not represent the OIB data well in terms of the intended use – in retrieving thickness from freeboard.

Please provide a variable list

---

## Author Comment (AC1) · 24 Aug 2018

Thanks for the information. We have added v1.0 to the title! Data/code links are already provided and we will explore publishing the GitHub repo on Zenodo too.

---

## Author Comment (AC2) · 24 Aug 2018

**Author response to Anonymous Reviewer #1 on: "The NASA Eulerian Snow on Sea Ice Model (NESOSIM): Initial model development and analysis" *by* Alek A. Petty et al.**

Reviewer comments are in black, our responses are in blue.

We will also submit the revised manuscript and a word document highlighting the tracked changes we have made based on these comments.

The paper describes the development, calibration and validation, and sensitivity of a snow on sea ice model. Given that the model is likely to be used to retrieve ice thickness from CryoSat and ICESat altimetry, the model should be documented in the literature. However, the paper needs to be improved and some points clarified before it can be published. I would suggest that the authors consider reorganizing the paper to make it more readable and potentially shorter.

We sincerely thank the reviewer for providing this review. It included some very helpful suggestions on how we can improve the manuscript, and areas where we need to provide more justification etc.

General Comments

While I recognize that a model cannot include everything, I am surprised that the authors do not include snow melt. I see this as a major shortcoming in the model. The top-left panel of Figure 3 shows, what might be interpreted, as a melt signal between March and April. I suspect that melt is also a factor in the North Atlantic sector. A warming Arctic is almost certainly likely to have melt earlier. This years warming 'spike' likely caused melting in some sections of the Arctic. While this might not have resulted in a loss of snow mass, it would have increased snow density and caused a reduction in snow depth - refreezing/metamorphosis is another process. The authors should address in more detail leaving these key processes out.

We have added more details in the summary of planned future model improvements, including the lack of snow thermodynamics (highest priority). We feel this discussion combined with the comments in the introduction make clear we see the lack of melt processes as the main shortcoming of the model.

We have also added the following to the end of the abstract: ' Potential improvements to this initial NESOSIM formulation are discussed in the hopes of improving the accuracy and reliability of these simulated snow depth and density.'

Regarding the decrease in the observed snow depths in the Soviet Station data: we actually doubt this is due to a melt event. Air temperatures remained below freezing for all of April for each station in 1980-1991. None of the stations showed a decrease in snow depth except for 1 (Station 31) in 1990 – the air temperatures rose to -4C, which could potentially have been a melt event given that it's a daily average. Looking at the sea level pressure for that time, there was a big decrease between April 10 and April 30 (the time when the decrease in snow depth occurs) – which suggests there may have been a storm event, which likely redistributed the snow and may have contributed to a decrease in mean depth along the survey line. A storm event would also explain the decrease in density (e.g., fresh snowfall).

Another point to consider is that we only took data from four stations (1980-1991). Despite some of them lasting for multiple years, this is still a relatively small sample size, and the mean in snow depth will be strongly affected by variations in a single data point. We refer the reviewer to

Warren et al. (1999) which shows data from all stations (1954-1991) and thus is more representative of the seasonal cycle.

Another concern is the use of the Polar Stereographic grid. The model tracks snow volume but only sea ice concentration, and not area, appears in Equation 1. Are the authors assuming that Polar Stereographic grids are equal area? This is not the case, they are conformal but not equal area. Cells at 70 N are about 10% smaller than cells at the pole. Maybe I am missing something here, and maybe this erroneous assumption might not have a big impact, but the authors should satisfy both themselves and readers that the choice of grid does not have an impact.

This is an interesting point, and working with projections/grids always provide some complexity that should not be quickly overlooked. In this case we are using equal areas in our polar stereographic projection, so quantities are being conserved in the model. However the reviewer is correct that the polar stereographic projection does provide a distortion to the real world such that our grid cells will be larger at lower latitudes. We do not see this as a huge concern as this is similar to the issue one always has when using polar stereographic data. We did consider using equal area projection (the Lambert azimuthal projection) but this can introduce potential issues with the vector projections in this non-conformal projection that could offset the potential benefits of less distorted area grids.

An obvious question, given the prevalence of Warren 1999 snow depths in sea ice thickness retrievals, is how different are the results presented here from W99? I would argue that W99 is the current benchmark for evaluation of snow depth products. The authors should include some discussion on this topic. I think it would also be useful to put the uncertainty reported in this paper (~ 10 cm) in the context of thickness retrievals. Ten centimeters is at least 30% of snow depths in the Arctic and similar to inter-annual variability in snow depth. While there is clearly room for model improvements, quality of precipitation fields and other forcing fields also come into play. It would be good to discuss these issues.

Another good point. Instead of simply comparing the results against W99, we believe the more interesting comparison is between W99 (actually the modified W99) and OIB (ground truth) and NESOSIM and OIB. This was similar to the approach taken by Blanchard-Wigglesworth et al., (2018) in their recent model evaluation efforts (published during this discussion period).

We have carried out this analysis showing that the difference between the model and mW99 compared to OIB is similar, but also depends on the chosen OIB product. See the figure attached:

[Figure]

**Figure 1:** Comparison of modified Warren snow depths against OIB snow depths for the years 2009-2015. The data are binned to the 100 km model grid, only for grid-cells with significant (>1000 OIB data points).

While the model doesn't really offer an improvement over modified Warren in terms of these comparisons, our hope is that with continued model developments and calibrations we can improve the performance and reliability. Uncertainty in the OIB data and the possibility they have been tuned to fit the modified Warren climatology make this interpretation more challenging.

We have added more discussion of these comparisons, and the comparisons to mW99 to the revised manuscript.

I find the model formulation as described in section and equations 1 through 8 confusing. This may be because I think of the process in a different way to that described here. Hopefully, my interpretation in the following paragraph, whether right or wrong, will help improve the model description.

Yes we appreciate these efforts to make the model formulation more readable. As you say, we are really tracking an effective snow depth, and we have tried to make this clearer in the revised manuscript.

I see the model as analogous to the evolution equations for ice thickness (or any other tracer), where the change in snow depth is the sum of a dynamic component (-del*dot* (hu) [I think] not del(hu)), and a static "snow depth evolution" component that rep- resents snow accumulation, ablation by wind, and compaction ($h^{acc}$-$h^{wp}$+$h^{bs}$ for the "new" snow layer and $h^{wp}$ for the "old" snow layer). What I find confusing is that $h^{acc}$ is the product of snow accumulation (snowfall/density) and sea ice concentration. Shouldn't what I call the sum representing the "snow depth evolution" component be multiplied by concentration (A) not just (Sf/density); e.g. A*(Sf/density -$h^{wp}$+$h^{bs}$) [Note my comment in the paragraph above – this only applies to a uniform grid]. Similarly, should h in –del *dot* (hu) be Ah.

I think our use of effective snow depth was confusing things. As you say we are really tracking Ah, which is why our use of A only really comes in when we introduce the accumulation term (if

we keep using it we double count). The other terms then act on this effective snow depth, so we don't need to apply the ice concentration again in those terms.

Maybe this is what the authors mean by "we track snow volume" – e.g. (Ah) – and "An effective snow depth...". However, I would suggest that it is Ah that is the effective snow depth because this term represents a mean grid cell snow depth (including open water areas). Whereas, h can be thought of as a physical snow depth because it represents the process of accumulation, wind ablation and compaction at point on an ice floe. I think what I describe is a conceptual difference rather than an error in the model because $h^{wp}$ and $h^{bs}$ are tuned, so the wind packing and blowing snow coefficients can be thought of as including sea ice concentration, i.e. you can change the model description in the paper without changing the model.

Agreed. We have changed this in the revised manuscript to read as: 'Note that in the model we track the evolution of an effective snow depth within each grid-cell (the volume of snow per unit grid cell area) for simplicity. The actual snow depth over the ice fraction is calculated by dividing the effective grid-cell snow depth by the grid-cell ice concentration.'

Two further issues are with model calibration and the use of the ensemble mean snowfall. If I understand correctly, the parameters are tuned using ERA-Interim snowfall and these "best" parameters are applied to model runs with MERRA, JRA55 and the Ensemble Snowfall. I would suggest this is the wrong approach. The calibration process will compensate for biases in ERA-Interim snowfall. However, biases in the other reanalyses are different. A different "best" parameter set should be expected for each reanalysis snowfall product. Conceptually, model equations, parameters and forcing data are all part of the Model. Using parameters obtained for ERA-Interim, might have detrimental effects on snow depths when other reanalyses are used. I would recommend that the MERRA, JRA55 and (maybe) ensemble runs should be calibrated separately.

We did consider this, and the idea makes sense. However, in reality the tuning was not highly optimized and instead we attempted to find parameters that achieved a good balance between capturing the seasonal cycles in snow depth and density in the Soviet Station data. Again, this was mostly due to concerns regarding how representative the Soviet Station data were for calibration purposes. We have added the following to the revised manuscript to explain this:

P17: "We also decided against specific model configuration parameter tuning due to the limitations in the calibration data, however this should be considered when analyzing the model performance, especially with regard to our validation efforts (i.e. more sophisticated and/or configuration specific tuning could improve the comparisons shown)."

This was also discussed in Section 4: " Specific model configurations may be required based on user demands, and our expectations is for these calibrations to evolve as new calibration data are made available and physical parameterizations introduced/updated."

We also added the following to this paragraph: " As discussed in Section 3, it is likely that specific model configuration tuning could improve these comparisons and the later validation efforts, but we decided against a more optimized calibration approach due to the limitations in the Soviet station data."

With regards to the Ensemble snowfall, the assumption behind an ensemble average being a better estimate is that individual ensemble members bracket "reality". Is this the case with reanalysis snowfall? If all ensemble members are biased in one direction, the ensemble average

will also be biased. My understanding is that both ERA-Interim and MERRA precipitation are both biased high compared to land stations. Based on Fig 11, JRA55 is also high. So is the ensemble snowfall an improvement over the individual ensemble members?

This is a good point and one we perhaps didn't make clear enough in our discussion of the ensemble 'median' snowfall data. We have dropped the idea that the median snowfall dataset might be somehow 'better' from Section 4 ("as we expect these results to be less prone to errors in the individual reanalyses etc.), so this now reads:
"we choose to mainly focus on the MEDIAN-SF forced results using the default configuration (Table 1) for simplicity"

We don't feel that land stations are wholly representative of conditions over the sea ice (confirmed from conversations with colleagues in NASA's GMAO for example) hence our recent study currently in press looking at precipitation estimates over the Arctic Ocean from reanalyses (Boisvert et al., 2018). While again we are limited in direct observations over sea ice, comparisons of the precipitation converted to snow depth (using constant snow density approximations and lagrangian feature tracking) against drifting snow buoys in the central Arctic Ocean showed no obvious bias in the differences between the reanalysis derived snow depth estimates used in this study and the buoy snow depths (in Boisvert et al., 2018). It did appear that MERRA-2 has a clear positive bias so was dropped from our analysis. Clearly more work needs to be done to better understand the precip within the Central Arctic, however.

There needs to be more detail about the ensemble snowfall was generated. For example, I can envisage ERA-Interim, MERRA and JRA55 all having snow but the location of this event being shifted by one or two grid cells. On one side of the event, while ERA-Interim might have no snow, MERRA and JRA55 do have snow. By taking the median, snowfall from MERRA or JRA55, would go into the ensemble product. On the other side of the event, MERRA might not have snow but ERA-Interim and JRA55 do, so snowfall would go into the ensemble product. This would result in a larger region receiving snow. How do you deal with that situation.

The Boisvert et al., (2018) analysis showed that the reanalyses tended to agree well in terms of the presence of a precip event, but differed strongly in the magnitude of the precip during an event. The fact the reanalyses assimilate the same sea level pressure observations means they tend to simulate storms in the same locations, , but they differ in the magnitude and phase of the precip. There was also no obvious suggestion from this analysis that there were significant biases in the location of precipitation events, although the different model grids complicate this slightly. The fact we re-grid everything onto our coarser 100 km grid should help somewhat with this issue. In general, we see this as a crude effort to produce a 'consensus' snowfall estimate on our model grid. Ideally we would have more reanalyses (that provide a direct snowfall estimate) to increase our confidence that the reanalyses are bracketing reality.

With respect to the flow and structure of the paper, sections 4 and 5 seem repetitive, especially where model sensitivity to reanalyses is discussed. Essentially, sensitivity analyses for both periods give the same results. It makes sense (to me) that if you are going to compare the two time periods then the discussion and plots are merged. This would reduce repetition, make it easier for readers to compare the two time periods, and maybe shorten the paper.

We have followed your suggestion and merged these sections, mainly through dropping most of the figures and discussion of the 1980s results. As you say, the differences are not large in terms of the model performance evaluation, and our interpretation of the different figures was pretty

similar. We have kept the budget figures in the SI. We refer the reviewer to our revised manuscript which highlights this change and our justification.

Many of the figures could be improved. In many figures, the colors are not sufficiently distinct, dark purple and dark blue. This is the case with figures where dots are used. Maybe get rid of the black borders to the symbols. Also use different symbols. For many of the line graphs, increasing the weight of lines in the legend and in plots would help. Also consider whether or not you need to show the spread. The overlapping shading obscures the lines showing the means. Using shading works for two, or possibly three, series, especially if they are separated, but it starts to detract from a plot and not convey the information you want it to with more series. For example, the North Atlantic plot in Figure 11: I can't see the lower limit of the JRA55 spread or the upper limit of ASR. In many cases the spread is not discussed in the text. If you still need or want to show spread, you could just show May 1 snow depth spread as vertical bars off to the right hand side of each panel. The issue of including plots in figures but not describing the plots in the text occurs in several figures (e.g. Fig 9). I would suggest that if it not discussed, then don't include it.

We have made several changes to the figures, especially based on the comments below, including clearer legends, thicker lines, better axes. Thanks for that.

The shading appears to be an issue in Figure 11 and 13. I am keen to include the shading as it helps show where the model results may significantly differ from each other over interannual variability. While it isn't always mentioned, the presence of clear differences beyond the spread (e.g. some of the JRA and ASR regional results) is clearer than showing just the means. Showing just some shading for some of the lines would be odd in my view and questionable to the reader. We have changed the axes to prevent the cropping, lightened the shading. and changed the colours, to make this clearer. These figures should also be more colorblind friendly (dropped the green).

The spread as of May 1st is summarized in the Table, which also allows for an easier comparison across the sensitivity studies if the reader struggles to see this in the figure.

Specific comments

L3, P5. "...to avoid complexity of snow melt processes". As I note in General Comments, Fig 3 shows what could be interpreted as melt. I would like to see more justification. Furthermore, a simple temperature index approach could have been used to account for melt.

We refer to our response to the general comment above and our inclusion of more discussion about this in the revised manuscript.

Equation 1. See General Comments.

See response to that comment

Equation 3. Should this be $-\nabla \cdot (hu)$ Equation 4. Should the divergence be $-h\nabla \cdot u$

Yes, thanks. The code was correct at least! I have changed these equations.

Equation 5. You have a wind speed threshold for wind packing but not for blowing snow. Why is this? Studies for prairie environments indicate blowing snow initiates above ~ 4 m/s, which is similar to your wind compaction threshold.

We agree it does appear odd to not apply this threshold to the blowing snow loss term, so have now added this in. This only had a small impact on the results, but has meant that we had to re-run the model and reproduce all the figures to highlight this small impact. We didn't change any of the coefficients as the changes were so negligible.

Section 2.5. Suggest this section is moved to 2.1 as an introduction to the modelling framework. This sets up the discussion of the parameterizations of the accumulation and sink terms.

We have made this change to the model description, as recommended. We hope this has improved the readability of the model formulation.

L12, P11. An advantage of reanalyses is that they produce consistent outputs. Mixing and matching fields from different reanalyses breaks this consistency. How similar are the ERA-Interim winds to MERRA and JRA55.

Yes true, but as we accumulate snow, the seasonal growth is more important than the daily variability. The study of Lindsay et al., (2014) compared several reanalyses in the Arctic (not including JRA) and showed that ERA-I winds were slightly higher (~0.5 m/s) than winds measured on drifting stations, and MERRA was slightly lower (~0.5 m/s). We decided not to add this to the investigation as it should be second order compared to the other variables and would likely just involve a recalibration of the wind packing and blowing snow loss coefficients.

L17, P11. "We linearly interpolate..." Do you mean bilinear interpolation? See General Comments. This needs more detail.

We use the Python SciPy interpolation package. The linear interpolation uses a triangular (barycentric) not bilinear interpolation approach. We have added a line: ' Gridding scripts written in Python are included in the GitHub code repository' to indicate that the reader can refer to our scripts to learn more about this and reproduce our gridding.

L13, P14. Are significant amounts of snow in summer likely to be present in recent years? The data in Warren 1999 is 30 years old at a minimum. Are there observations from N-ICE or other field campaigns to justify non-zero initial snow depths. Further- more, how do you initialize new sea ice? This needs to be explained.

The inclusion of more recent summer initial snow depths was also justified to correct a low bias in snow depths when we compared to OIB. We agree there is not much direct evidence to justify this, and the location of the N-ICE campaign will limit its utility here (highest initial snow depths north of Canada/Greenland coastlines).

The initial snow depth is only applied to regions where we have grid-cells with a concentration above 15%. As we track the snow in a given a grid-cell, new ice forms with no new snow, but can accumulate snow instantly. We have added this to the manuscript: ' New ice that forms in a grid-cell is assumed to be snow free, but these grid-cells can accumulate snow instantly.'

L24, P14. "The snow depth is distributed evenly over the old and new snow layers...". Is there a reason why initial snow depth was not just assumed to be dense old snow.

As discussed after this line, albeit briefly, we assumed based on the sparse old observational studies that this was a combination of snow that didn't melt/persisted through the melt season, and

some early summer/fresh snowfall. We admit this is somewhat unconstrained, but it based on the observations in Radionov et al. (1997)

L19, P16. "We carried out initial model calibration..." For this study, you only calibrated the model once, right? Suggest drop "initial" throughout this discussion unless multiple calibrations were made.

True, dropped initial from the text in a few places where it seemed inappropriate.

L22, P16. "...calibration involved manually tuning NESOSIM to improve the general fit..." Was this fit judged "by eye" or was some metric used? Also were all years 1980 to 1991 used, or did you leave a year out for validation during this period. While I recognize that you validated for 2000 to 2015 using OIB data, measurement accuracy and conditions might be different between the two periods.

Yeah this was by eye. We would have carried out a more optimized calibration effort if the in-situ data were more consistent, but as it was, we didn't want to over fit the model to this sparse dataset. We agree with your statement about old and new accuracies but feel this was the best compromise between producing a calibrated model we want to run primarily for a new arctic time period (because of the available altimetry missions).

L14, P17. It looks as if there is a larger difference in snow depth between January and March. Modelled snow depths gradually increase, while observed depths appear to increase in accumulation rate. Is this a shortcoming of the snowfall products. Also, you should mention the decrease in depths in April that could relate to melting and or compaction.

As discussed earlier, the limited coverage of the in-situ data prevents us from saying more about these comparisons (i.e. to be confident of the presence of a particular high or low bias in the model in our view.

L2, P18. It is difficult to believe a correlation of 0.6 for density given the spread of points in the plot.

We checked and stand by these values. The change in axes limits perhaps makes this clearer/more believable! Note that the density of points increases to the upper right which is the main reason for the moderate correlation strength.

L7, P19. "Including the blowing snow loss... but no significant change in snow density." My first thought here, is why expect any change in density? The only mechanism by which density can be influenced by the blowing snow parameterization is a reduction in the "new snow" depth. So how deep is this "new snow" layer and how quickly does it get redistributed to the "old snow" layer?

It could change the snow density by essentially removing fresh snow (only this layer can be blown into leads) before it gets transferred into the old snow layer. We have added: ' This parameterization can impact the bulk density implicitly by reducing the amount of fresh snow contributing to the total snow depth/density.'

L7 to 15, P19. Maybe add that blowing snow loss in the central Arctic are small because sea ice concentration is close to 100%.

Adapted that line to read: 'As the drifting station data are collected primarily within the Central Arctic where ice concentrations are near to 100%,'

Section 4.2 and Figure 8. I am struggling to make sense of this section. I think part of the problem is that evolution terms are shown as cumulative, which makes comparison difficult: a big snow storm could deposit several 10's of centimeters of snow, dominating the snow depth for the rest of the season. I think you can compare the magnitudes of the terms at the end of the season (May 1) (as you do in the text) but not during the season. To compare terms during the season, I think you need to compare the timestep change in each component. The comparison is not helped by the fact that it is very difficult to distinguish lines in Figure 8. The lines in the legend need to be thicker. I would suggest leaving snow volume out of the figure.

We have tried to make the lines and shading similar, through similar efforts to Figure 11 and 13. I generally like the idea of showing the daily timestep change, although they were pretty noisy and harder to distinguish than just the cumulative plots (a lot of lines crossing over).

Our aim was really to show the cumulative impact as this fits in with the fact we validate the model in spring, at the end of the accumulation season. We agree understanding the model budget terms in all seasons is also crucial but hope to explore this more in follow up work.

L12, P24. Prefer "advected" to "drifting". For snow, drifting implies blowing snow.

Agreed, changed.

Figure 9. I would suggest showing only the evolution terms that you discuss in section 4.2. Other plots can be put in supplementary figures. While I suggest you don't show the snow volume, note that the units are a depth. Moreover, I think Ah_s (sea ice concentration * snow depth) is better thought of as a gridcell mean thickness.

In response to another reviewer we have added better labels to this figure. We are keen to keep all these panels in as we think readers could be interested in seeing the spatial importance of the various fields. We have changed volume to snow depth, and changed the final panel to 'snow depth over ice'.

L13, P26. Prefer "Soviet Station" to "old" period.

We have actually changed this to the '1980s' time period here and throughout the revised manuscript. The Soviet Station era goes back to the 1950s so didn't want to use that label to avoid confusion.

L13, P26. Given the spread in snow depths in the "New Arctic" and "Soviet Station" periods, are they really that different?

The spread in snow depths based on the input data make such conclusions hard to state and we have tried to avoid this in the paper. We have added more discussion around this issue in response to the other reviewers, but have also tried to remove the focus on comparing time differences as we think this beyond the scope of this initial model evaluation paper in line with your comments of too much regional analyses. As such we have shifted the focus to the 2000s analysis and put the 1980s results in the SI.

L8, P29. Maybe use "difference" instead of "bias" as you have no "truth".

Agreed, changed.

L2, P30. See General Comments. If Median-SF is biased it might not be that useful.

We refer to our response to the general comment.

L21, P30. "regional variability" – suggest "regional scale". Regional scale is contrasted with Pan-Arctic Scale.

Agreed. Changed this to ' capturing the variability in snow depth at this regional scale'

Figure 17. Why does NESOSIM have zero snow depths but OIB has non-zero snow depths. It is difficult to interpret the panels with the OIB datasets overlayed. Maybe just show All-years but with separate panels for SRLD, JPL and GSFC OIB products. The individual years can be included as supplementary figures and/or discussed in the text. Also maybe use dots rather than x's to avoid symbols overlapping.

Good spot, we realized there was a small error in the gridding and zero values in the model along the coastline were being included in the binning/regression. We have now removed these values (adding in a zero snow depth mask) which has removed this issue. This is another thing to consider in this comparison as the various OIB products have different treatments for including low snow depths, and some may use cut-offs for higher snow depths. We decided to keep all the OIB snow depths in this comparison (especially as we bin the data up to 100 km), but is something to consider in future validation efforts.

We have also taken your advice and now just show the all year regressions in separate panels. The figures of the individual years are shown in the SI and the r/rmse values are still shown in the table. We also included Kernel Density Estimate contours to highlight more clearly the differences in the distributions.

Technical Comments
Abstract, L9, P1. "Several simple parameterizations to represent key sources and sinks". The number of processes is not large, so you might as well list them explicitly, rather than keeping the reader guessing :).

Agreed, we have added: '(accumulation, wind packing, advection/divergence, blowing snow lost to leads)' in parentheses.

L22, P3. Suggest "availability" rather than "presence".

Agreed, changed.

L12, P4. "(Show later)" Give a figure number.

Agreed, added.

L14, P4. "Ice drift". Suggest "Ice Motion" to avoid confusion with drifting snow.

Agreed we have changed drift to motion here and in several other places in the manuscript.

L8, P5. "...our reanalysis data..." Suggest "...reanalysis fields...".

Agreed, changed.

Table 1. Add symbols for snow densities.

Added, along with the model variables (suggested by other reviewers).

Figure 3. As snow depths and densities are binned, could the data in the upper panels be shown as "box and whiskers" or just boxes. That way readers can see the amount of overlap between depth and density estimates. I suggest you spell out Soviet Stations in the figures. Add 1:1 lines on the lower panels. It would be nice to have a single symbol in the legend.

We have added Soviet Station to the legend and axis labels, reduced number of markers to 1, and added the 1:1 line. We decided box and whisker plots were too much as we already show the mean and spread (+/- 1 SD) and have the raw values below.

Figure 4. Does No Initial Conditions (NO IC) mean the model was initialized with 0 cm snow depth?

Yes! We have added more information to the caption to help clarify the model runs.

L1, P26. Shouldn't this be Section 5?

Yes! changed.

L13-14, P29. Reference needed.

Added reference to Boisvert et al., (2018).

Figure 16. Why two symbols in the legend. Also the colors are difficult to distinguish. Maybe no black border on symbols. Also use different symbols.

We have changed this to just show one marker, made the markers bigger and made the OIB markers squares to more clearly differentiate them from the model markers.

---

## Author Comment (AC3) · 24 Aug 2018

**Author response to Anonymous Reviewer #2 on: "The NASA Eulerian Snow on Sea Ice Model (NESOSIM): Initial model development and analysis"** *by* **Alek A. Petty et al.**

Reviewer comments are in black, our responses are in blue.

We will also submit the revised manuscript and a word document highlighting the tracked changes we have made based on these comments.

This clearly written manuscripts provides a detailed description and exploration of the new NESOSIM snow model. NESOSIM produces gridded, daily snow thicknesses and densities for Arctic ocean sea ice during the accumulation season (defined as mid- August through April) given daily inputs of Arctic wide snowfall, sea ice concentration, ice drift, and near surface winds. Although the Arctic melt season may extend well into September, the model does not include thermodynamic or radiative processes, and this certainly limits its utility. Rather, the emphasis here is on the impacts of wind via wind packing and blowing snow loss to leads/open water. The parameterizations are fairly simple – winds exceeding a threshold can only decrease snow thickness and increase snow density. There are no snow drifts, for example, or sub-grid regions of bare ice which are present in other models. In addition, there are no snow-aging processes that may contribute to density changes. Still, the authors do a commendable job validating their model against observations and do a thorough evaluation of model sensitivity to the various snowfall reanalysis, ice drift and ice concentration products. This latter analysis highlights the true utility of the model – a simple framework for the inter-comparison of reanalysis-derived snow on sea ice data products.

We thank the reviewer for their time in producing this review and the useful comments provided.

Some specific scientific comments: The authors need to better place the work in scientific context and show how the work is unique. How is this an improvement over the simple models of snow depth forced from reanalyses? There are more complex snow on sea ice models (Lecomte-LIM, Liston-SnowModel, Hunke-CICE) which include some of the same processes (ice drift, dynamics, precipitation) yet rather than develop wind loss and compaction include some distinctly different processes (thermodynamics, radiative properties, snow ice formation, dune formation, ridge accumulation. . .). Are these models missing the "key sources and sinks"? There is also Dery and Tremblay (2004, JPO) that specifically looks at the effects of wind redistribution with an explicit mass flux into leads. Is your approach better? More useful? Consistent?

This is a fair comment, although we want to stress we don't seek in this paper to produce a model that is 'better' or 'more sophisticated' than the models currently available. Indeed this would be a real challenge considering the complexity of some models already available. But model sophistication isn't the only factor in producing reliable snow depths - the forcing is arguably more important, and there is still a high level of uncertainty surrounding the sensitivity of snow to various forcing data, especially snowfall. As such we sought o develop a model that will enable us to explore these sensitivities and increase model sophistication as needed based on these uncertainties.

We have moved up to the introduction and added more text regarding our motivation/philosophy (this was at the start of the modelling description) to make this clear from the outset and have also adapted the introduction to add the following before this statement:

" Due to these observational limitations, the sea ice community often relies on simple models of snow depth forced by reanalyses (primarily snowfall data) (e.g., Maksym and Markus 2008;

Kwok and Cunningham, 2008; Blanchard-Wrigglesworth et al., 2018). More sophisticated snow on sea ice models are available, such as SnowModel, a terrestrial snow model recently adapted for sea ice environments (Liston et al., 2018), as well as the snow parameterizations included in sea ice climate model components, such as CICE (Hunke & Lipscomb, 2010) and the Louvain-la-Neuve Sea Ice Model (LIM), which has recently undergone various improvements to its snow physics (Lecomte et al., 2015)."

What is the impact of excluding thermodynamic processes on your results? Does this change your conclusions about the impact of wind processes?

Unfortunately we really lack the data needed to better answer this question. It's likely that including thermodynamic processes (e.g. snow melt) will reduce the impact of the wind loss term, but that is already a very unconstrained process in this model.

We also refer the reviewer to our response on this subject of missing melt processes raised by Reviewer 1 and the newly added discussion of not including snow thermodynamics, the potential impact of this, and the future work section at the end of the paper.

Some misleading statements: First sentence of the abstract. . . . "produces daily esti- mates of depth and density of snow across the polar oceans". Not yet because of some important missing processes. Qualify with Arctic only and during the accumulation sea- son.

Agreed. Changed 'polar oceans' to 'Arctic Ocean through the accumulation season'.

Using old vs new snow in the text and figures. It's clear that there is intention to one day include snow aging, but for now there is only fresh vs compaction. The depth hoar densities of 150-250 is never used in the model even though paragraph 10 seems to suggest that it is. The old snow value is 350 kg/m3 which is not the average of the higher end of wind slab and depth hoar (325 kg/m3) but rather the average of the wind slab bounds.

The reviewer is correct that there is the intention to one day include snow aging in the model, but, for now, there is only "new" and "old" snow. We think there may be some confusion here as we take a weighted average based on the average ratio of depth hoar and wind slab from Radionov et al. (1997) and Sturm et al. (2002). The density values chosen for these layers are based on the literature on snow density observations (e.g., Radionov et al., 1997; Sturm et al., 1998; Sturm et al., 2002; Sturm and Massom, 2017) and implicitly include values for density layers that are not explicitly treated in the model. For example, the compacted layer uses a value of 350 kg/m3, which is a value calculated using the average ratios of depth hoar (40%, 250 kg/m3) and wind slab (60%, we found values of 410 kg/m3 in the referenced works above) within a snowpack. Admittedly, the ratio of depth hoar and wind slab seasonally changes and the model does not account for this seasonal change. However, we feel this is a better treatment than neglecting the influence of depth hoar altogether in the "old" snow layer. We have edited this description in the revised manuscript to make this clearer.

Perhaps future developments could be kept to a specific section to better clarify what the model does and doesn't do.

Yes, Reviewer 1 wanted this too, so we have added this to the final summary section. We refer the reviewer to this new subsection.

Snow density in NEOSIM is bounded by the two chosen snow density parameters (200 and 350 kg /m3) even though the observations referenced give values for dry snow of 150 and wind slab ~400 kg /m3 on average. Why exclude these possibilities at the outset? Instead of using an average value, doesn't it make more sense to use the upper and lower bounds given the nature of the parameterization? How sensitive is the model to these values?

We did experiment with these values, but found that this resulted in an over-estimation of the strength of the snow density cycle. See figure below. We have added this to the new manuscript:

"We did experiment with alternative snow densities (e.g. the wider spread of 150 and 400 kg m$^{-3}$) but found this provided worse correspondence with the seasonal snow density evolution compiled from in-situ Soviet Station data (introduced in Section 3.4)."

[Figure]

The late summer initial conditions integrate all the missing snow melt processes and for that reason, they are rather important. The paragraph on page 14 does a fairly good job motivating your approach, but it would be clearer if you showed the equations for hs(0) and hs(1) after summer melt. Also better explain how snowfall events factor into this parameterization and explain why keeping the same fraction for fresh/compacted snow is the right approach (or clarify if you do something different). It would also be informative to see the Aug 15 values in your figures 3 and 4. Are there Aug observations to help validate the IC parameterization and fig 2 in particular?

We think there may be some confusion here. We rerun the model every August, regardless of the prior spring snow conditions. The starting point for the model is a direct application of the initial condition snow depths.

We don't show these values in the figure because we start the model halfway through August and these figures are showing monthly means/spreads.

The equal split between fresh/old snow was due to (albeit fairly crude) reasoning that some of this initial snow may be from snow persisting through the melt season and some due to new snowfall through summer based on observations in Radionov et al. (1997). We explain and cite this in the manuscript.

We lack consistent observations of August snow depths to validate this component of the model, but we did look at the snow depth data from ice mass balance buoys – some show the presence of snow in August, while others do not, within the same summer seasons. While these observations suggest snow is present in August, these are point measurements and may not be wholly representative of basin-scale snow depth distributions.

Why absorb the timestep in the model equations? In (2) the parameter alpha has a timestep dependence that isn't explicitly called out and as a result, 0.05 is less meaningful. Better to define an alpha with units of per second.

Agreed, we have changed these equations to include a time step value, so these coefficients are now independent of the length of the time step. The units have been updated accordingly (Beta needs to be in units of  per meter as we multiply by wind speed).

Are the differences between simulations with different snowfall estimates larger than the differences between time periods? Are the time period differences significant?

The spread between reanalyses makes it challenging to ascertain confidence in a physical system change between periods. Understanding the factors contributing to the differences between periods and reanalyses (e.g. the contribution from precip, freeze-up, other processes) is beyond the scope of this paper, but is a topic of ongoing work.

The potential changes in snow depth and their physical cause are part of a hopeful follow-up study which can elaborate on this in the detail required. We also refer you to our response to Reviewer 3 where we provide a brief summary of our thoughts about this.

Note also that to shorten the analysis description (and to prevent the focus being about changes in snow depth) we have merged the 1980s/2000s reanalysis sensitivity study sections and moved the 1980s figures of model evaluation to the Supplementary Information. The 1980s reanalysis sensitivity studies did not add much to our discussion and interpretation of the model performance so we have dropped this from the revised manuscript.

Fig 14 seems to suggest that ice drift is actually quite important but masked by basin or large regional averaging. Magnitudes of the differences are similar to the snowfall sensitivity. Impacts are near the ice edge (increase ice retreat?) and add to smaller (but still > 100 km) scale variability (potentially impacting melt-pond formation).

Agreed! We discuss this in the paper already so haven't added more here.

Technical corrections:

Table1. add the model variable in the table.

We have added all model variables to the table.

Define U in Eq. (5).

It was defined after Eq. 2 earlier.

Missing ) in Eq. (6)

Added

Why is the bs term in Eq. (7) positive?

The wind loss term is now negative, so this should make more sense now!

Missing t dependence in some terms in Eq. (7) and the next equation Line 26 missing "of" in "one the better.."

Thanks, good spot. Added.

Add W99 to upper panels of fig3 Add W99 to fig 6 (a)

We decided not to add W99 to figure 3 as the legend makes clear we show data from the drifting Soviet Stations (hence SS in the legend) from 1981 to 1991. The W99 is a climatology of these data over the longer time period (back to the 1950s) so it is not exactly the same. Figure 6 shows no Warren or Soviet Station data so we didn't add this either.

In explanation for fig 8, NA region shows a small "decrease" due to convergence, not increase. CA shows a small increase but is not mentioned.

Changed this to: 'The NA region also shows a small (~2 cm) decrease (increase) in snow depth driven by snow/ice divergence (ice/snow advection), while the CA region shows a small (~2 cm) increase in snow depth driven by snow/ice convergence.'

Fig (9). What is (b) Ocean? Change (d) wind/leads to wind loss to leads. Is there an ice area cutoff for snow depth in (k)?

Ocean was the snowfall into the ocean. We have updated all these labels to make this a lot clearer, including the variable names and a link in the caption to the table which provides the model variables.

All snow depth results and budget fields shown in the analysis use a concentration cut-off of 15% and which we have now added to the revised manuscript at the start of page 9 after the model formulation description: 'Note that this bulk snow density is masked if the respective ice concentration is less than 15% or snow volume is less than 2 cm, while all snow budget terms presented here show data only when the concentration is above 15%, to prevent spurious results in regions of near open water conditions.'

Table 4 could be improved by adding the 1981-1991 time period for comparison, identifying the boxed regions as "reanalysis sensitivity", "ice drift sensitivity", "ice concentration sensitivity" and including in the description that the default configuration is MEDIAN-SF, NSIDCv3 ice drift and Bootstrap ice concentration

See comment above. The comparison between time periods is the focus of on-going work led by co-author of this paper Melinda Webster. As it requires careful evaluation of the uncertainty in forcing data and changes in these fields we feel we can't make a simple comment about that here. We have thus decided to drop the 1980s reanalysis figure (the 2000s figure serves this purpose just as well) and moved the 1980s budget figures to the SI.

End of p 30. Comment on the NA region in table 4 when ice drift is included. Seems to be important here too.

Yes, this was an omission and has been added to the text.

Fig 13, (d) NA does not appear to be consistent with table 4. The NODRIFT value on May 1 is around 34 cm in table 4 but seems to be much higher in this figure. Explain.

The table value was wrong and has been corrected. Although based on the tweak to the wind loss term (introduction of the wind action threshold for this term as well as wind packing based on reviewer recommendation), all values have been reproduced and are slightly different from their original values. Apologies!

Last comment before 5.3 does not seem correct. There does appear to be bias in the real-time products with respect to the peripheral seas .

Agreed, we have dropped this statement.

P 37. What are uncertainties in the OIB observations?

The recent STOSIWIG study (Kwok et al., 2017) stated that snow depth can be estimated from a Snow Radar echogram with an uncertainty of 'several centimeters' although this depends strongly on the ice conditions, the particular Snow Radar system being used, and various other factors (e.g. geolocation errors associated with the plane pitch and roll. We have added a sentence regarding this to the discussion at the end of Section 3, where the OIB data are introduced.

Mention the results from sensitivity of ice concentration in the summary. These were interesting and significant.

OK we added the following: 'We also briefly assessed the sensitivity of NESOSIM to the input concentration data, with our results suggesting that this choice of product (Bootstrap and NASA Team explored in this study) can have a significant impact and should not be overlooked.'

---

## Author Comment (AC4) · 24 Aug 2018

**Author response to Anonymous Reviewer #3 on: "The NASA Eulerian Snow on Sea Ice Model (NESOSIM): Initial model development and analysis"** *by* Alek A. Petty et al.

Reviewer comments are in black, our responses are in blue.

We will also submit the revised manuscript and a word document highlighting the tracked changes we have made based on these comments.

The authors present a new open source model, the NASA Eulerian Snow on Sea Ice Model, for estimating daily depth and density of snow on sea ice. The authors note at a few points in the paper that the model is being developed primarily with application to altimetry-based ice thickness determination in mind, though other applications are likely. The model is a simple representation of the snow that is largely an accounting of snowfall produced by reanalysis data, similar to prior efforts (e.g. Maksym and Markus 2008; Kwok and Cunningham, 2008), with terms for snow compaction, loss to leads, and transport on sea ice. It is Eulerian, but features pseudo transport by exchange between grid cells, features only 2 layers, and is forced with available spatially and temporally complete datasets that are known to be of limited accuracy (e.g. Reanalysis, passive microwave concentration). The model is calibrated/validated against limited available snow on sea ice data from Operation Ice Bridge and from 1980s era Soviet drifting stations. The description of the model is complete and in this regard the model is publishable with minor revisions – but reviewer doesn't feel the model is very good or useful in its current form for its intended purpose. Reviewer focuses most of this review on highlighting its shortcomings. In fact, a possible conclusion of this data presented would be that simple treatment of snow on sea ice will not meet the accuracy levels required for altimetry applications. The reviewer encourages the early career team to put the paper aside for awhile and take the time write a model that would actually be highly used.

We thank the reviewer for putting the time into providing this review. As also highlighted by the other reviewers, the model we developed has provided a useful means of thoroughly assessing reanalysis-derived snow depths and their sensitivity to the input forcing data used. It is hard to develop a highly sophisticated model and explore true sensitivity to input forcing data, and the latter has been extremely lacking in the literature to-date. We believe this study has provided the needed baseline from which we hope to increase model sophistication in future as needed.

The reviewer feels that the key issues are that the model is excessively simplistic, not representative of known physical process (even at the level of simplicity targeted), and that its results show it is inadequate for the intended purpose. There are errors in the equations presented, many compromises appear to have been made that make accuracy and/or realism lower in favor of rapid release, and as a result the work is unlikely to have much impact as presented. The presentation in the paper is quite long, and focuses on trying to convince the reader that the model is good, rather than taking a hard look and comparing against a reasonable standard.

We know of no other pan-Arctic snow depth & density product that provides significantly higher accuracies (e.g. the rmse when compared against OIB say).

Based on the comments of Reviewer 1 we have added similar comparisons of the OIB snow depths against Warren and modified Warren climatologies, highlighting similar/better agreement when using NESOSIM (in terms of correlation coefficient and/or root mean squared errors). We are somewhat limitied by the uncertaintiy in the OIB products (several centimeters) that prevent us from carrying out a more complete validation.

We have also added a lot more justification at the start and end of the revised manuscript regarding our approach and expectations for future model development/calibration efforts to improve these error.s

We have also added the following to the end of the abstract: ' Potential improvements to this initial NESOSIM formulation are discussed in the hopes of improving the accuracy and reliability in these simulated snow depth and density'

The development of a snow product for improving retrieval of sea ice thickness from altimetry is critical for ICESAT 2 to be useful and this team should have NASA's support to do just that. Such a snow model's accuracy goal must be based on a desired accuracy in thickness retrievals. (e.g. retrieval of ice thickness accurate to +-0.5m over a given domain demands snow depth accurate to O5cm over the same domain). The model presented here is not up to meeting these kinds of needs, and does not leverage the existing (more sophisticated) models of snow on sea ice (e.g. LIM, SnowModel, CICE).

We are not aware of more sophisticated approaches providing the requirements you list, and our belief is that uncertainties in the input forcing data needed to be explored further, along with efforts to improve model sophistication. We see this model as a contribution towards this end goal. Based on the comments of the other reviewers we have added more discussion of the other snow models available and the physics they include.

We agree that working towards a 5 cm error makes sense, and we believe this provides a useful framework to build towards that goal. This is also discussed in a new study by a co-author of this study: Webster et al. (accepted) which speaks to several issues of our current abilities in observing and treating snow in models. Existing basin-scale observations unfortunately do not have 5-cm accuracy; snow depths derived from merged satellite data do not have 5-cm accuracy. So it is difficult to produce model output with 5-cm accuracy given that no observations of 5-cm accuracy (or better) at the basin-scale exist for model validation and assessment, and especially on a seasonal time-scale.

Some major issues include: Model design relative to state of knowledge: 1. The two layers used in the two layer model (new snow and windslab) do not represent the two layers of the snowpack discussed in literature (wind slab and depth hoar). Authors cite and discuss the literature indicating that windslab and depth hoar dominate the mass of the pack and have quite different density – then ignore these decades of observation to invent a new scheme unsupported by observations. Respecting the effort to create a simple, 2 layer model, new snow should not be one of the two layers. The references cited clearly state that new snow rarely comprises much of the Arctic snowpack, because it is very rapidly converted to windslab. The preservation of a new snow layer appeared to be designed for modeling loss of snow into leads – but little is known about the magnitude of this flux, and it was minor in this model. 2. The model is operated on a 100x100km grid, which is very coarse relative to the variability in ice – which is shown to be important in impacting the accumulation of snow. The data sets used provide much higher ice concentration, and movement information – this data should be used at full resolution and atmospheric data can be downsampled. 3. Melt is neglected despite it being important during part of the timeframe and having significant impact on results.

1. We expanded on the description in the manuscript about the second layer representing depth hoar and wind slab and their respective densities. We treat wind slab and depth hoar by taking the average ratio of these two layers, based on Radionov et al. 1997 and Sturm et al. 2002, over the

accumulation season. This ratio is then used to take a weighted average of the wind slab and depth hoar densities. We agree that new snow provides a smaller fraction of the total snow, especially towards the end of the accumulation season, which occurs in both the model and in observations (Warren et al., 1999).
2. We respectfully disagree, especially as we are taking gradients in ice drift fields on daily time-scales that can be very noisy, so other recent studies, e.g. Holland et al., (2014) smooth the data before using them. We hope to explore specific high res configurations of this model in the future, but wanted to carry out more extensive sensitivity studies across various forcing data in this initial model development stage.
3. Yes we neglect it and have added more discussion on this point based on the input from reviewers 1 and 2 in the model development and future priority sections.

Quality of the Model Results and Characterization thereof 1. Validation shown indicates the model produces results that do not capture the variability in observed snow depth or density reliably. Authors focus on averages of model output over decadal timeframes, which can be made to match observations by tuning of the arbitrary, non- physical constants in the model. This focus fails to acknowledge the inability of the model to capture interannual or spatial variability. 2. Prediction intervals are not pro- vided, but scatter plots show little relationship between individual observations of snow depth and modeled snow depth. No discussion is provided of how these errors would propagate in the intended use (altimetry retrievals of ice thickness) but it appears errors are sufficient to radically alter retrievals of depth and appear to indicate the data would not be useful for altimetry retrievals of ice thickness from ICESAT2. Authors fail to acknowledge any of these shortcomings and go to great pains to make the results appear good. 3. Modeled variability in density appears to have very little relationship to observations. 4. Comparison with the southern ocean, are pushed to a future effort, but validation statements in the paper suggest the model applies to 'polar oceans'. 5. Results from the median of the three reanalysis products are declared 'better' repeatedly with no reasonable support. Taking the median of atmospheric reanalysis models would result in nonphysical jumps between atmospheric states and the removal of extreme events from the record, and is challenging to support physically.

1. We are confused by this comment as the assessment of how well the model captures spatial and interannual variability is shown in the OIB validation section.
2. Not sure what you mean here either. We show both the correlation coefficient (how well it agrees on the relationship between the two distributions) and the root mean squared error which provides a model error compared to OIB. We did not hide these numbers and have been very open with the performance of the model. The RMSE of ~9-10 cm is clearly not ideal but a value like this was broadly expected considering the challenges inherent in modelling snow accumulation and snow depth/density. The observational uncertainty on the order of several centimeters should also be considered here and we have made note of this in the discussion. It's thus hard to translate this into a snow depth uncertainty.
3. Correct, as we stated in the manuscript.
4. We have made clearer this is focused on the Arctic Ocean.
5. We have made clearer the justification for showing more of the median results in the New Arctic time period discussion was mainly for simplicity!

DETAILED COMMENTS

Page 1, line 16. "very strong agreement" Delete "very strong"

It does show very strong agreement with the seasonal cycles (very high correlations with the seasonal correlation plots) which is what we are referring to here.

Page 1 line 22 descriptions of agreement too subjective. The use here is altimetry. Tell the reader about the error in estimates implied.

We have changed this to: showing moderate/strong correlations and root mean squared errors of ~10 cm depending on the OIB snow depth product analyzed. These are similar to the comparisons of OIB-derived snow depths and the commonly used modified Warren snow depth climatology. Potential improvements to this initial NESOSIM formulation are discussed in the hopes of improving the accuracy and reliability of these simulated snow depths and densities.' We are wary of translating this into an error, due to the uncertainty and differences in the OIB snow depth products.

Page 2 line 5-8. Poorly worded sentence. Consider modifying. One suggestion is: The altimetry technique involves measurements of freeboard, the extension of sea ice or snow surface above a local sea level. Estimates of snow depth are required to derive sea ice thickness from either snow surface freeboard or ice freeboard, because snow depresses ice freeboard and adds to snow surface freeboard. Snow depth is one of the primary sources of uncertainty for both laser and radar altimetry (e.g. Giles et al., 2007).

We appreciate the suggestion but prefer the sentence as is.

Page 2 line 10. Replace 'lacking' with something more descriptive/accurate (they aren't lacking they are just not complete/good enough).

Changed this to 'very limited'. Also added 'direct' to observations at the start of the line.

Page 2 line 22-24. The sea ice community often relies on simple models of snow depth forced by reanalyses – please clarify how this is different. To the reader, it still looks like a simple model forced by reanalyses!

Here we are just making a comment here about what the community are using currently. Ours is also a simple model forced by reanalysis as you say and we don't think this suggests otherwise.

P 3 Line 16 "and two snow layers to broadly represent the evolution of both old/compacted snow and new/fresh snow." The assignment of the two layers in this two layer model is not consistent with the widespread understanding of the primary two layers on sea ice as depth hoar and windslab. New snow is occasionally present but usually rapidly transformed to windslab. It may be an acceptable third layer. See many of the snow on sea ice references cited here, such as Sturm et al., 2002 – generally the snow is treated in these two layers. The author's choice here to take the two layers to represent layers that the extensive literature reviewed does not discuss is perplexing.

We completely agree with the reviewer that the primary snow density layers are wind slab, depth hoar, and to a much smaller degree, new snow which, in reality, is more like a "transient" snow layer that gets redistributed by the wind. If reduced to two layers, the snowpack would be wind slab and depth hoar.

Given that the model does not have a temperature dependency, we were not able to parameterize a depth hoar layer since this is dependent on the temperature gradient within the snowpack.

Instead, we chose to include a "new snow" second layer, representing recent snowfall and blowing snow, which reduces the bulk density of the snowpack. Likewise, for the "old" snow layer, we explicitly chose density values that result from the mean ratio between wind slab and depth hoar from works by Matthew Sturm and historical data from Radionov. Although the model doesn't explicitly take into account the seasonal cycle of this ratio, we feel that it's a better treatment than applying the higher-end density value of wind slab and ignoring depth hoar altogether. Related, we chose not to apply a bulk climatological density because of its questionability in representing regions where observations are lacking.

P2 line 18 replace "detailed" with "iterative". The simplified scheme does not permit a 'detailed' assessment of connection between input data and snow depth given its lack of physical complexity – it permits an easier iteration of possibilities.

Removed detailed from this line.

P4 line 13 Input data from passive microwave higher resolution than 100x100km, even if atmospheric data is not. Since ice concentration is so important, reviewer questions if 100km resolution is adequate. Further - does observed snow depth vary over 100km resolution? Since this is the motivation, what resolution is needed for useful for altimetry based determination of sea ice thickness?

As both drift and snowfall products come on grids with resolutions above 60 km, we did not want to use the higher res 25 km ice concentration grid. We think in these budget models the grids should be at least as coarse as the coarsest input data or it could look misleading. This is the approach taken in the concentration budget studies referenced in the manuscript. In reference to the later part of the question, higher resolutions are clearly better, but accuracy is really more important. One can always downscale data to higher resolutions but doing so doesn't add additional information at that higher resolution.

Page 4 line 14 add "from reanalysis data" after the word 'drift'.

The ice drift is not from reanalysis data. We have modified this to 'The model is forced with daily data of snowfall and near-surface winds from reanalysis data, satellite passive microwave ice concentration, and satellite-derived ice drifts.'

Page 4 line 16 – (volume of snow per unit grid cell in units of meters) – doesn't make sense volume is meters cubed. Throughout the treatment of snow varies between depth and volume freely, but this free transition between volume and depth is challenged for some considerations of snow – particularly convergence/divergences. Since the goal here is to understand depth for altimetry retrieval, a convergence, which moves volume into a cell, is not the same as a change in depth.

This kind of terminology is common in models (e.g. CICE) to describe a quantity that is expressed over the entire grid-cell. Based on the recommendation of reviewer 1 we have tweaked the terminology used so that we make it clear we track the effective snow depth (over the entire grid cell) but can derive the snow depth over the ice fraction by dividing by ice area.

Page 5 table one – put formal references to data sources, e.g. "bootstrap" is not sufficient.

We provide references in the text and in the data section of the paper.

Page 5 delete "snow pit and density data. . . helped guide. . . parameterization . . . seasonal evolution." There is no prescribed seasonal evolution of density, use of snow pit data etc. in this model. Two constant snow densities are selected and declared. This sentence obfuscates the very simple, non-experimentally supported nature of the scheme.

The model produces a prognostic density from the ratio of old and new snow and the calibration of the model was guided by this data. The average ratio of wind slab and depth hoar within the second (old snow) layer is parameterized based on snow pit observations from Radionov et al. 1997 and Sturm et al. 2002.

Page 6 line 8 replace bulk density with mass.

Agreed, changed.

Page 6 – here authors note that the community of snow science experts and prior literature they have created generally group the snow into two layers (wind slab and depth hoar). They further note substantial differences observed in density of these two layers, and that these two layers comprise the majority of the snowpack. Not noted, but available in the literature is data showing that the contribution of the two layers to the overall snowpack varies from the approximately 50-50% contribution seen at SHEBA. So it seems windslab and depth hoar are the two layers to model. But. . . these two layers are different than the layers the authors have chosen (new snow/old snow). It seems a major departure from decades of snow research is being made here and it is not being well defended. Why?

We discuss this in more detail in response to some of your earlier comments, but briefly we were aiming to keep the model simple in this fist iteration. As we have no snow thermodynamics, explicitly capturing both snow layers and their different densities is less important for our given purpose.

Page 6 line 12 "for this reason we use the average of higher end values of ws and dh". Reviewer sees no reason provided supporting the use of the higher end of the range of values for each of the two common layers. The mean density of each layer, multiplied by the mean fraction of each layer should provide a more representative density for the combined wind slab and depth hoar. Further, the value selected is not the average of the higher end of the range of values for each of the two common layers, leaving it unclear how it was determined.

The density values chosen for the wind slab and depth hoar in the "old snow" layer are based on the average ratio of these properties within the snowpack from Radionov et al., 1997 and Sturm et al., 2002. The compacted layer uses a value of 350 kg/m3, which is a value calculated using the average ratios of depth hoar (40%, 250 kg/m3) and wind slab (60%, we found values of 410 kg/m3 in the referenced works above) within a snowpack. Admittedly, the ratio of depth hoar and wind slab seasonally changes and the model does not account for this seasonal change. However, we feel this is a better treatment than neglecting the influence of depth hoar altogether in the "old" snow layer. We have revised this description in the manuscript to make this clearer. In the conclusion, we expand the discussion to include future work in incorporating a temperature-dependency of the model, which will enable the separation of wind slab and depth hoar layers rather than applying a crude treatment for the old snow layer.

Page 6 Line 16 "Our simple parameterization is thus expected to be generally representative" No reasonable evidence provided supports this. Statements like this are found throughout this paper.

Delete or support with concrete evidence that quantifies what the range of uncertainty they will work within.

We have deleted this statement from the revised manuscript.

Page 6, Line 23 (default of 5m/s). Default or for the purposes of this work is it simply always set to this?

It's fixed in the model but can easily be changed in the open source code.

Page 6, Line 24 "determines the fraction... transferred..." Over what time? (seems that the coefficient is model timestep dependent. . . and perhaps shouldn't be)

Yes, we have changed this to not be timestep dependent (units of $s^{-1}$ now )

Page 6 line 26 'Wind threshold of 5m/s was determined based on. . .' studies. Please add a description of the range of wind thresholds indicated by these studies, and why 5m/s was selected from within that range.

We implemented a 5 m/s threshold based on Liston and Sturm (2002) and the dry snow transport in Li and Pomeroy (1997). In reality, this threshold depends on the snowpack's physical properties (grain size, water content, etc.) and atmospheric conditions (humidity, air temperature, etc.).

We conducted sensitivity tests on the wind speed threshold and found this to have a negligible effect on the modeled snow depth distributions. However, there are some regions where the wind threshold and wind loss term will play more important roles, such as the Antarctic environment where more leads are present, more snowfall events occur, and windier conditions occur more frequently relative to the Arctic (Massom et al., 2001; Toyota et al., 2016; Massom and Sturm, 2017; Massom, pers. comm., 2018; Webster et al., accepted).

Page 6 Line 8 Daily gridded ice drift is still required in this Eulerian scheme, eliminating it as a reason for choosing Eulerian over lagrangian, discussed above.

Our statement referred to the fact you do not need consistent ice drift - i.e. the model can be run for periods/regions with no ice drift.

Page 7, line 19. Reviewer is not aware of any evidence indicating that the loss of snow to leads in the North Atlantic sector of the Arctic is significant relative to the thick snowpack in that region. No evidence seems to be coming out of the N-ICE experiment to that effect. Some quantification of loss to leads in the Antarctic has been made by Leonard and Maksym as noted, but this was in the southern ocean. Please cite appropriate literature or delete speculation.

Correct, the snow lost to leads in the North Atlantic may not be significant relative to the thick snowpack in that region. However, we speculate that a greater proportion of snow is lost to leads there than in other Arctic regions given the lower ice concentrations, more open leads, more frequent snowfall events making more fresh snow available to redistribute, and windier conditions in the North Atlantic.

Page 8 line 4 – This parameterization doesn't make sense and is under supported for several reasons. 1. It appears that a constant coefficient beta is multiplied by 10m windspeed NOT by the

amount which the wind speed exceeds the threshold velocity! So snow is lost to leads even when winds are too slow to move snow. 2. The amount of the snow lost to leads increases linearly with windspeed, when the drifting snow volume is well known to vary more rapidly than linearly 3. The loss to leads varies linearly with open water area, again this is likely more rapid than linear, and a thought experiment with random lead spacing/size could arrive at a better approximation. 4. The parameterization removes a fraction (2.5%) of the new snow layer to leads on each windy timestep – timestep is then important due to compounding what timestep is this defined for? 5. Is this parameterization/ value supported by any field quantification of loss to leads or is it simply made up due to lack of available observation. Either is fine, but state which it is. Page 6 line 9 – missing parenthesis on equation

1. We have changed this such that the wind action threshold is applied to the blowing snow loss term.
2. Our approach is simple, agreed. we hope to explore each term in more detail in future developments.
3. Agreed.
4. This term is now time step independent.
5. Yes, this was not based on any studies. We have added clarification to the text: ' We have no observational constraints for this parameter and is a free-parameter in this model, chosen through our model calibration efforts.'
6. Added the parenthesis, thanks.

Page 6 Equation 7 – appears incorrect. Change due to blowing snow is added (last term), but this should be a loss term (loss into leads). It appears that the term calculated in Eq 5 is always positive, so adding here will result in addition of snow, not loss. Similarly, how signs are handled on dynamics, convergence and divergence as well as advection depends on how (+-) ui is defined in equation 3 and 4, and this is not (but should be) specified above. . . so the reviewer is unsure if the sign here is handled correctly.

Correct, this was a mistake. We have added a negative sign to Eq 5 and 6. This was correct in the model code. Dynamics are all correctly signed.

Page 8 line 21 August is mid- late summer. Change "early" to 'late' or delete.

Changed this to 'middle'.

Page 9 line 2-3 Do these melt events invalidate the results here? Is this model useful before these 'hoped for' additions occur? It sounds like this is being hurried along.

We have no data to test if this invalidates the results here but the extra snowfall did improve our initial model testing efforts so we included it despite the obvious concerns of missing melt events.

Page 9 line 8-13 This paragraph appears to handle a specific test case, not discussed here. Seems out of place possibly a draft fragment. Unclear what tests this new density applies to, or how this test relates to the model released for community use. (update after later reading, now understand what this refers to, but still feel it was out of place and not well enough contextualized here)

Changed calibration to testing to make this clearer. The 1-layer approach is not included in the model released to the community.

Page 9 line 16 Soviet - capitalize.

Changed.

P 9 line 26 one OF the

Changed.

P9 – would be appropriate to acknowledge the lack of validation sites or validation data over Arctic sea ice, and uncertain accuracy of the products in that region.

We have added some comments regarding the uncertainty in precip earlier in the revised manuscript.

P11- Taking the median of the reanalysis products is an interesting idea if one has no idea which of the different products is best, but don't authors have better information about which is doing best from the comparison studies in literature?

Not really over Arctic sea ice. A new study of precipitation comparisons over the Arctic Ocean has recently been published (in early form) in the Journal of Climate but co-authors of this study, but the lack of validation data makes it challenging to recommend a particular product that one should focus on for such studies. This exercise was useful to guide us to exclude some reanalyses from our study - e.g. MERRA-2 - as stated in the MERRA data description.

P14 L10 – Initial conditions the Warren climatology is quite outdated. It is good you are trying to update them somehow. Is there evidence, e.g. from current autonomous ice mass balance buoys, that snow still regularly survives summer? Can you 'calibrate' this adjustment scheme based on those observations? Would a degree-day model be better than number of melting days? Also, what category is this snow placed in? Does it have a density reflective of melting snow (i.e. 400-500 kg/m3)?

Good question, and one that we can't answer with certainty about large-scale snow depth distributions in summer given the lack of observations. However, we can piece together information to hint at an answer. Based on these "pieces" (observations), some snow persists and the amount of snow that persists in summer has decreased relative to the Warren climatology.

In the IMB data, for example in 2012 and 2013, four IMB buoys show snow is present while six buoys show snow-free conditions in mid-to-late August. Based on survey line data from SHEBA and HOTRAX, snow can persist in summer where the largest drifts exist (next to ridges, etc.). However, when in the field, it's extremely difficult to tell the difference between a melting, slushy snow layer and a melting slushy ice surface (see the SHEBA field notes for an interesting reference for this – some scientists called this slush snow while others called it ice). What we can say with more confidence is that there is less snow in summer than there used to be based on the decreasing trend in surface albedo from AVHRR and melt pond information from MODIS data.

If the reviewer means a model based on the number of freezing-degree days, then our opinion is that the persistence of above-freezing days would have a larger effect on the removal of snow than the amount that's present in May.

We don't explicitly use a density in the initial conditions since we scale the climatological snow depth based on the number of consecutive above-freezing days.

P14 L22 – explain how this is 'linearly scaled' a bit better. Provide an equation. Is the fraction by which duration of melt is different from mean simply multiplied by snow depth? Does it mean that at 2x duration no snow is left and at 0x duration 2x snow is left?

We have provided more detail in the revised manuscript.

scaling factor = (mean duration for climatology - duration for individual summer)/mean duration for climatology

initial snow depth = august climatology*(1+scaling_factor)

We could add this to the manuscript if the reviewer thinks necessary.

*We only have ERA-Interim data for 1979-1991 rather than the full 1954-1991 climatological period to calculate the mean duration of above-freezing days in summer.

P14 L 28 – were necessary. . . Could this be because the model doesn't handle melt processes?

We don't believe so since deeper snow depths were needed to improve the comparison between the model and Soviet station observations (hence the initial conditions). Melt would reduce the snow depths so we don't think that is why.

P15Fig2 – These substantial August snow depths in 2012 and 2013 should be compared against available buoy data to determine if they are reasonable. The reviewer believes they are not and that this is ultimately a nonphysical tuning mechanism that helps account for lack of melt processes and poor representation of precipitation phase at this time of year in reanalyses.

The IMB data for 2012 and 2013 show four IMB buoys with snow present in summer (and six buoys with snow-free conditions). For the cases where snow is present, three of the four buoys show depths of ~10 cm and larger. The buoy IDs for persistent snow are: 2012C, 2012D, 2012G, and 2012I. Note, some of these buoys show data for summer 2013 and are not just limited to the year 2012, when they were deployed.

P 16 L 17 – this section is missing a clear statement of how accurate OIB data is expected to be.

The recent STOSIWIG study (Kwok et al., 2017) stated that snow depth can be estimated from a Snow Radar echogram with an uncertainty of 'several centimeters' although this depends strongly on the ice conditions, the particular Snow Radar system being used, and various other factors (e.g. geolocation errors associated with the plane pitch and roll. We have added a sentence regarding this to the discussion at the end of Section 3, where the OIB data are introduced.

P16 L 19-27 – pretty hand wavey – not rigorous.
Show plots of how snow evolves in model – what fraction of the snowpack is new snow layer vs time (it would have to be small to be realistic.)

We feel the plots already included in the manuscript provide the seasonal evolution of the budget terms highlighting the small contribution from the new snow layer.

P 17 Fig 3 – modeled data appears systemically low by about 5 cm depth. Snow density has essentially no relationship between modeled and observed. Individual year data – which is how this data would be used to derive altimetry based estimates of ice thickness – appear poor.

We didn't want to over fit the model to the data. Obviously it would be easy to calibrate this to increase snow depth in this comparison but we wanted to be flexible to the fact these depend strongly on the chosen forcings and the time period of analysis.

An r = 0.58 does not translate to no relationship. Clearly the snow depth shows a better relationship, but the density relationship isn't awful.

The validation part of the model is more in reference to the modern OIB data record, so we refer the reviewer to this.

P17 L 13 delete "extremely" – an error of ~5cm on a snowpack of ~20cm is still a 25% error.

We are referring to the seasonal cycle here and have attempted to make this clearer that we are referring to the seasonal cycle. We dropped extremely.

P18 L 1 – r of 0.74 would not generally be characterized as 'strong' P18 L2 - delete 'more' . . . its just moderate. Also, what is actually suggested here is that the model is good at predicting the MEAN – because you can tune your constants to make the mean look very nice, but not very good at capturing the interannual variability that is key to getting snow depth right for altimetry.

We are unsure how you have decided it's not good at getting the interannual variability? That's not really shown in this figure due to the lack of coincident data across years.

P18,L4-5 "In General, the moderate/high correlations. . . provide confidence. . ." This statement is hand waving and cheerleader- y without content. Delete this statement and replace it with a statement that articulates the degree of certainty with which output of the model should be treated. Suggest authors calculate the +-95% prediction interval for a modeled density or depth relative to this dataset. Suggest authors do this for individual locations/months on individual years, as well as mean.

At this stage we are just calibrating the model. The validation and expression of model errors comes in the OIB section.

P18, Figure 4 – This comparison is really just showing how well tuned the models are on average. Since the model is not presented as a mean climatology, but rather is presented as a deterministic snow product for specific locations on specific years, this comparison is inadequate.

Again, the OIB comparisons provide this later.

P19 L 3-6 Here authors make an odd argument. The model does not reproduce the climatology observations as well as a single mean over the entire timeframe. They argue this is OK because the model will handle interannual variability better because of its 'more advanced' density parameterization. The density parameterization is not particularly physically realistic, however, and fails to meaningfully capture interannual variability of the climatology density data (figure 3d). Reviewer therefore finds this statement lacking.

We state that the model is able to respond to expected interannual variability in the forcing, which is different to how the reviewer expresses this. Again we test this later with OIB.

We agree our approach is very simple, so have changed the more advanced line to read 'include a simple bulk density parameterization;

P19 l10 – not all marginal ice zones have low concentration – clarify that low concentration areas are where greater impact is expected

We have changed this to 'low ice concentration regimes'

P 19 L 19 – Reviewer disagrees that wind threshold velocity for blowing snow is un- constrained. Resources reviewed can establish that under all but extreme conditions (e.g. recent rain on snow) a threshold of 10 m/s is pretty high, maybe unreasonable. It would be better to range this within the values observed in the references cited earlier.

Changed this to poorly constrained. As you say this is on the high side, but we wanted o test the sensitivity so wanted to provide a big change, and doubling was consistent with the other changes. It does appear from this that increasing this value only slightly might lower the low bias in snow depths compared to the Soviet station data, but would also lower the density to create a low bias. This is obviously a tough balancing act and further calibration efforts will be explored in future.

P 20 L 15-18 – Speculative. Reviewer finds no reason to believe the median should be superior in regions of heavy snowfall. Defend or delete.

We have changed this to 'for simplicity'. As you say we provide no evidence that the results are better, but it is a useful synthesis forcing data to use - preventing the creation of figures across all the different forcing sets.

P21 L 5 – again this represents the mean over the decade being presented/compared. The model performance over this timeframe is highly tunable and not the performance metric of interest to an end user taking this data as an input to altimetry – that user would want to know the prediction interval for individual or moderate size groups of snow data points, and probably also whether there is any change in mean bias over time.

We have provided more information about the comparisons in terms of the root mean squared errors, which we feel are the best and most fair way of comparing these snow depth distributions.

P 22 Fig 6 – are standard deviations in depths this low comparable to any observations? If so they suggest a single climatology would be adequate for most end uses.

It's challenging to compare the SD from a pan-Arctic model with the SD from an in-situ dataset. We also dont have enough confidence in the snow density to make such a statement.

P22 L 15 plurals

Not sure what this refers to.

P23 L4 The fact that there is scatter among the reanalyses is not necessarily an argument for taking the median of them. Delete.

The point is that the snowfall forcings show strong differences so also having some kind of 'consensus' (e.g. median) snowfall dataset is useful. We have changed this line in the text.

P23 L19 providing significant VOLUME REDUCTIONS and sinks of snow (wind packing is not a sink)

Agreed, we have changed this to reductions of snow volume.

P24 L1 – Convergence really causes an increase in snow volume, not an increase in depth. The use of depth vs volume for a cell needs to be sorted out and treated consistently throughout this paper. Reviewer recalls a section way up at the top saying snow would be handled in volume throughout the paper, but has seen treatment vary.

We have made it clearer that we are tracking an effective snow depth (snow volume distributed across the entire grid-cell) and replaced volume with depth throughout the revised manuscript.

P25 Fig 9b – unclear what "Ocean" refers to. Snowfall directly into water? Caption needs to be more descriptive and the figure subcomponents should be linked back to which equation # they represent.

We have completed re-labeled this figure to improve clarity.

Fig 9f – this underscores the issue with treating convergence as a change in depth- in reality convergence/divergence of the ice at scale does not change depth in the sense that such would be used to interpret remote sensing. It changes snow volume, further, it appears the impact of convergence/divergence is noisy at best.

We treat this as a change in effective snow depth over the grid-cell. This is the most consistent way of treating this and avoids issues with the changing ice concentration.

Fig 9 I – density map warrants discussion. For example, density appears highest in central arctic (far from melt) This is likely untrue and an issue with not including melt/rain on snow processes. Very low density is indicated in marginal areas around N Greenland and in Baffin bay/Canadian islands. Can these be supported at all?

We agree that melt processes, especially in the marginal seas, should increase the density in these locations. This has been added as a future priority for the model.

Fig 9 h-k colors appear to fade toward lower value indication near land in general. Is this valid or a plotting artifact? Again volume and depth are used interchangeably and plotted on a depth scale. This needs to be resolved consistently.

Not sure I really see what the reviewer is referring to. Maybe some lower values in the older snow but likely due to wind packing as not seen in the new snow results..

Page 24 L 5-7– Authors state these measurements are explicitly for altimetry retrievals, so they must have characteristics useful for such, (more than matching seasonal evolution on aver- age) including: 1. Capture interannual variability 2. Capture spatial variability 3. No long term bias or trends in error (that could be mistaken as trends in ice thickness). If these cannot be shown, perhaps a discussion about whether this approach is viable is needed.

This all comes in the OIB validation section.

Page 26 Figure 10 – "As in figure 6" This is a nice addition to convey consistency, but pls also provide full description in caption, don't make reader hunt back several pages.

OK added.

Page 26 L15 Without melt processes included, what explains the loss in depth?

I believe in this case it is no significant accumulation and wind packing reducing the density. Can be shown in Figure S4.

Page 26 L 5 Why the depths are different during the later time period IN THIS MODEL is within the scope of demonstrating a model, even if understanding why they are changing in reality is not. Please answer: Are depths less due to less ice for snow to fall on? Or due to less precip? The reader must know if the model is representing the changing Arctic – since it is calibrated on old data. This cannot reasonably be scoped out of the study.

The snow depths are different between periods primarily due to the difference in the timing of sea ice freeze-up. Maximum snowfall rates occur in autumn (Warren et al., 1999; Webster et al., 2014; Lique et al., 2016; Boisvert et al., 2018), so if sea ice forms later, more snow falls into the open water relative to when sea ice formed earlier. This has been modeled in Blanchard et al. (2018) and Webster et al. (accepted), studies implementing a similar modeling approach to NESOSIM, and another study using a more sophisticated model (CCSM) (Hezel et al. 2012) as well as in ongoing work using CESM 1. This relationship has also been shown in Webster et al. (2014) based on ice mass balance buoy, *in situ*, airborne, and satellite data. The *in situ* and buoy data suggest no significant change in snowfall rates, while the combination of *in situ*, buoy, and airborne data show a decrease in snow depth that corresponds to later sea ice freeze-up (derived from passive microwave data).

We deliberately do not include a more in-depth discussion on this topic because 1) the manuscript is already at 13,000 words, and 2) we're preparing a manuscript on this topic and feel that including this discussion would detract from the purpose of the other manuscript. A comparison of the reanalysis precipitation between the 1980s and 2000s show little difference, which suggests that less precipitation (magnitude, phase) is not the primary driver of these differences. This is still open to debate, however. As the reanalysis description was very similar across the time periods we have now dropped this from the manuscript (following the recommendation of Reviewer 1). The 1980s budget figures have been moved to the SI too. Our aim is to focus on the model performance in the 2000s period, along with the OIB validation analysis. If the reviewer is interested in seeing more results on this topic, we would encourage him or her to contact Melinda Webster at melinda.a.webster@nasa.gov

P 27 Fig 11 – see comment on Figure 10

OK, changed.

P 29 Fig 12 – please clarify what positive and negative deviation mean (is the product higher or lower snow than the median product)?

Red (blue) colours indicate the individual reanalysis-forced simulations have higher (lower) snow depth.

P 30 line 3 – it is not clear that the median provides a result any more useful than the others. One should note which product compared best to coastal stations data and any other indications from literature which might be best.

The comparisons with coastal stations (e.g. Lindsay et al., 2014) are not wholly representative of precipitation biases over the Arctic Ocean. This is discussed more in Boisvert et al., (2018) and we reference this in the revised manuscript.

P 30 line 20 – this is not surprising and should be noted as such.
Advection/convergence/divergence was much less important than snowfall in the plots above.

This is showing the spread based on the product spread, rather than its importance to the total mass balance. We agree this isn't surprising, but it is worth documenting and including.

P30 L 23-24 – here is where the idea of snow depth vs snow volume is really important. Dynamics are perhaps not important in depth over a 100km cell average, but they are important to the DEPTH on the actual subgrid ice, since divergence creates new ice with no snow, rather than rearranging all the snow into a gridcell average. The averaging over the 100km cell at each timestep may be particularly important in ice generating areas, where snow is continually averaged back into source regions, rather than being advected out entirely. Tracking ice classes within the cells, as is done in CICE may be critically important.

We agree this is worth thinking more about in future.

P 31 figure 13 – the drift scheme matters little over huge areas because convergence and divergence cancel. This plot is just not the right way to consider this, particularly in the context of use fore spatially distributed altimetry observations. Figure 14 suggests that the drift products don't differ that much between them in the central basin, but that having drift represented at all is very important, altering snowpack by O50% in large areas of the Arctic.

The point here is that if one cares about regional mean snow depths, the choice of product isn't that important really. Agree it matters more on the grid-scale hence the reason to show maps in the following figure.

P32 L6 – There are actually substantial biases in the peripheral seas – which may not be important overall, but cannot be ignored in the statement about biases.

Added ' and issues around the ice edge'

P33 – given the importance of concentration product, better understanding the role of changing concentration in the changing modeled snow depth above is important.

Agreed.

P34 – Tough to compare to observational data this noisy. Reviewer agrees they can be considered 'in agreement' within the bounds of the error of either. . . both of which are large. Are any of the OIB algorithms emerging as superior? Must all three be treated as equally likely?

Not sure noisy is the right phrase to use here, but there are clear differences in the products (an observational uncertainty). They seem to all have different pros and cons depending on the

region/year/scale being analyzed. The STOSIWIG paper (Kwok et al., 2017) made no clear recommendations in this regard unfortunately.

P35 – comparison of 100km grid cells still includes substantial averaging, but already shows poor agreement. Agreement should be presented in terms of a 95 % prediction interval so user knows the capability of the method in useful terms – if the model says snow was xx OIB will say snow was xx +- yy 95% of the time.

Figure 14 shows the spread in snow depth across all the OIB campaigns, with the spread/uncertainty based on the product spread. If we wanted to add the individual product uncertainty we would have to guess at this as, they don't necessarily provide this. It is also thought to be highly variable and a function of the ice type and snow depth profiled. The spread is expected to be very large for a regional mean. The RMSE comparisons in Figure 15 etc. are a useful way of showing how the products and NESOSIM compare, but we acknowledge throughout the high uncertainty in both NESOSIM and OIB estimates making such comparisons challenging to interpret.

P36 – this discussion of the comparison of the scatter plots goes to great lengths to avoid describing the obvious. The model isn't very good at reproducing variability on OIB data, and if you believe OIB snow data is in any way representative of the variability in snow depth on ice, the modeled snow depth isn't very good at capturing spatial or interannual variability. The conclusion should then be that more sophisticated model representations are needed or that OIB data is trash. Since the model didn't agree with the Soviet drift station data scatter plot very well either, I don't think you can conclude that the model is adequate but OIB is trash.

We are confused by this statement. We very clearly present the correlations and rmse values for the model and OIB comparisons, then also show the mean interannual variability comparisons. The reader can clearly make their own opinion regarding the comparison but I don't think that would be that either the model or OIB are trash. The correlations are moderate/strong in general, there are no obvious skews/biases and the rmse is ~10 cm, with some values lower than this. Both the model and OIB have uncertainties associated so saying stronger statements than we have should be done with caution. Hopefully our inclusion of the mW99 comparisons help put thse in context.

P37 – "in general, however, the moderate to strong correlations. . . gives us confidence" Reviewer cussed in exasperation when reading this. This is a science paper not an opinion piece. These are not moderate to strong correlations! They clearly show NE- SOSIM cannot capture the variability observed well. Get this subjective language out of the paper and replace it with quantifications of how well the model does at both rep- resenting means (where performance is good because of tuning) and variability (where the model is not working so well). Talk about whether the model is good enough to be used in altimetry honestly and present some paths forward to getting there if it isn't.

See discussion above. We did indeed include an interpretation of the comparison metrics we presented in this study based on the chosen metrics, which is pretty standard practice. The choice of moderate/good/strong was based on standard definitions for the interpretation of correlation coefficient values.

P 37 L 19 – data is yet to be released in parenthesis? Thin its out now. . .

We changed this to 'the data was not available for this study but was made available during the review phase of this paper'

P 37 L 20 – There must be some field data available that you could at least spot check it against!

Unsure what the benefit of a spot-check would be in this instance as it wouldn't be a particularly robust comparison.

P 37, L 26 delete very strong, delete good

We would like to keep these in. The agreement with the mean seasonal cycle was very good (very high correlation).

P 37 L 28 contributing to the MODELED seasonal evolution in snow depth

Added.

P38 L5 uncapitalize New, consider replacing with 'more recent.'

New Arctic was cited earlier so we wish to keep this statement.

P38 L7 There is no evidence presented that this median product is better, and good reason to believe it just averages in erroneous values and non-physically jumps be- tween atmospheric states toward limited representation of extreme events. Defend the use on scientific merit or consider deleting the median product.

We dropped the last part of this sentence.

P38 L10 use consistent language. . . it is 2nd order on mean, but first order in some regions.

Added 'in our regional mean analysis'

P 38 L 14 "moderate/strong correlations" This statement is flatly unsupported by the results shown, and authors 'confidence' in line 16 is unfounded. The product does not represent the OIB data well in terms of the intended use – in retrieving thickness from freeboard.

We are discussing correlation coefficients above 0.5, with some above 0.6/0.7. Moderate/strong are the appropriate way of describing these values based on the statistical guidelines we referred to.

Please provide a variable list

Thank you. We have added a variable list to the paper.

---

## Author Response (AR2)

Reviewer comments in black, our responses in blue.

Comments to the Author from the Topical Editor (Jeremy Fyke):

To the authors,

I'd appreciate a final set of comments from you - ideally that relate to final manuscript edits - on the last reviewer's final reviews, prior to publishing. My main resistance to publishing immediately is that the last reviewer lean towards 'publish' but also is frankly still uncomfortable with what they perceive as 'releasing' a model that still contains deficiencies that more or less preclude it from it's intended usage (as I interpret their comments).

Perhaps your final comments to myself and within the manuscript could clearly address the scope in which your model is applicable, and which scope in which it is not? For example, the reviewer suggests model is not suitable for usage in determining snow thickness during late spring/summer. In general, a GMD paper should very honestly list the model caveats and related usage limitations. These are ultimately intended to provide 'cover' to the model developers, so that blame for user over-use/over-interpretation doesn't fall back to the developers.

We sincerely appreciate your continued efforts to support this peer review process. See below for our responses to these remaining issues.

With regard to the comment on intended usage - the sea ice community still relies heavily on the Warren snow depth climatology (a quadratic fit to in-situ snow depth measurements collected through the 1950s-1980s) which are now often adjusted (halved) based on the given sea ice type. This is the snow depth data currently used in satellite ice thickness retrievals (e.g. for all the current CS-2 thickness products). The ice type adjustment improves errors when compared with Operation IceBridge snow depth data, but clearly fails to capture the full Arctic snow depth distribution - the distributions show a clear bimodal distribution due to the crude ice type modification used. While our NESOSIM derived RMSE's are not really lower than the modified Warren snow depths, the model is able to capture interannual/decadal variability in forcings and produces a more realistic, continuous, snow depth distribution, as discussed by the reviewer below. Perhaps more importantly, it also provides a framework to address improvements in a physically consistent way (i.e. new model) instead of purely calibration approaches (e.g. tweaking the ice type modification).

To make clear where we are with these snow depth errors and their impact on ice thickness retrievals, we have added the following to the end of the summary section.

*' Errors in snow depths of around 10 cm are thought to contribute to errors in ice thickness estimates derived from laser (radar) satellite altimetry of ~70 cm (50 cm) assuming typical freeboards of ~30 cm (Giles et al., 2007). We expect that further*

*model development, calibration, and validation is needed to improve accuracy and reliability in the NESOSIM snow depths/densities, to improve their utility in ice thickness retrieval analyses. '*

With regard to the comment on late/spring summer, which we take to be June through August - we make clear that we are only using this model for the accumulation season (end of August to May 1st) throughout the manuscript, at the start of the final summary section and now even at the start of the abstract, following the suggestion by the reviewers. We stand by our data being appropriate to use compared to the existing snow depth products available for this time period. We agree the model is not currently suitable for the late spring/summer time period and have added in more discussion of this idea in the revised manuscript (e.g. how we likely need to add some kind of parameterization of surface melt for the model to be useable in those months).

We have added the following line to the start of the model description section to make this clearer: *'NESOSIM is not currently configured to produce snow depth estimates through the melt season (late spring through summer) due to the lack of surface melt processes included in this initial model formulation.'*

We have done a lot of the caveat listing already with regards to our model formulation/sophistication, especially in the revised manuscript, to make clear the shortcomings of the model. The new section on future model improvements we hope made this even clearer and provided some thoughts on how we could improve this, especially as we hope this to be a community-driven effort over the coming months/years.

I also wanted to add that the majority of existing snow accumulation modelling efforts have not made their code available, hence our need and desire to produce a new, open source, snow accumulation model. GMD's strong focus on this important part of the publication process was a key reason for submitting to this journal.

**Second Round Comments**

***Suggestions for revision or reasons for rejection (will be published if the paper is accepted for final publication)***

One note that I think the authors should include is that figures 15 and 16 indicate a considerably better performance of their model than the W99 climatology than the metrics they use suggest. For the OIB estimates, W99 essentially indicate snow depths of ~20 and ~35 cm for first year and multi-year ice. Their model results appear to capture much more of the variability of the OIB depths. They should also note that OIB depths are themselves retrievals not direct measurements. The higher correlations between OIB and adjusted W99 seem to result entirely from including FYI and MYI snow depth distributions in the analysis. If FYI and MYI were treated separately, correlations would be lower (figure S12 suggests about 0.2). This does

not change the RMSE but does at least show some improvement of the model over W99.

Thanks for your comments and continued efforts to improve this manuscript, this additional comment regarding the Warren comparisons is an interesting one.

We have added 'retrievals' as a modifier to the OIB snow depth products throughout the discussion in Section 4.6, thanks.

We have also added a bit more to the discussion of the mW99-OIB comparisons, highlighting the significance of the ice type weighting.

With regard to the idea that our model would give better comparisons than modified Warren if we delineated by ice type: we did carry out this extra analysis and the results were interesting, but very mixed! Note that Figure S12 was not modified based on the ice type which explains those worse comparisons with OIB, but here we want to mainly focus on the modified Warren snow depths and just pick the modified Warren values in the bins delineated as FYI/MYI (remembering that only the FYI snow depths are modified).

The results when delineated by FYI/MYI depend on the OIB product analyzed - e.g. the JPL comparison showed lower RMSEs over the MYI, but also lower correlations, the SRLD product comparison showed the opposite in terms of the RMSE, while the GSFC product shows virtually similar RMSEs regardless of ice type! See Figures 1 and 2 below. For the NESOSIM comparisons there was generally a small reduction (increase) in the strength of the comparison when delineating by FYI (MYI) across all OIB products (Figure 3 and 4). As there is much more MYI than FYI in the OIB data, it is also tough to ascertain the degree to which this increase in coverage explains some of the improvement in comparison statistics.

As such, we have decided not to fully include this discussion in the revised manuscript, as all this really does is highlight the challenge of interpreting comparisons considering the strong differences in snow depth retrievals. We have added the following to the discussion, however:

*'We also carried out OIB comparisons by delineating by ice type (first-year ice and multi-year ice) using the same OSI-SAF product discussed above. However, the results were mixed, and were also strongly dependent on the OIB product analyzed. Such delineations are also hindered by the lower coverage of first-year ice in the OIB data, despite this becoming an increasingly dominant component of the Arctic sea ice pack. We thus choose to exclude this analysis from our discussion for simplicity.'*

[Figure]

**Figure 1:** As in Figure 16 but only using the bins delineated as MYI

[Figure]

**Figure 2:** As in Figure 16 but only using the bins delineated as FYI

[Figure]

**Figure 3:** As in Figure 15 but only using the bins delineated as FYI

[Figure]

**Figure 4:** As in Figure 15 but only using the bins delineated as MYI

[revised manuscript text omitted]

| Year | | | | | | | | | | | | |
|---|---|---|---|---|---|---|---|---|---|---|---|---|
| 2011 | 0.3811 cm | 0.28 8 cm | 0.46 9 cm | 0.5610 cm | 0.47 7 cm | 0.619 8 cm | 0.29165 cm | 0.207 9 cm | 0.3812 cm | 0.149 10 cm | 0.04 9 cm | 0.2510 cm |
| 2012 | 0.7389 cm | 0.70689 cm | 0.73910 cm | 0.75 8 cm | 0.72 8 cm | 0.75 9 cm | 0.7212 cm | 0.6711 cm | 0.7210 cm | 0.67 8 cm | 0.63910 cm | 0.6611 cm |
| 2013 | 0.697 cm | 0.68134 cm | 0.6515 cm | 0.737 cm | 0.74123 cm | 0.70614 cm | 0.679 cm | 0.65101 cm | 0.63123 cm | 0.6778 cm | 0.66134 cm | 0.6415 cm |
| 2014 | 0.68101 cm | 0.635911 cm | 0.63110 cm | 0.746910 cm | 0.706910 cm | 0.68 10 cm | 0.64123 cm | 0.58123 cm | 0.6015 cm | 0.53112 cm | 0.48102 cm | 0.5210 cm |
| 2015 | 0.659 50 910 cm | 0.52 101 cm | 0.485 0 10 cm | 0.658 810 cm | 0.625 2 911 cm | 0.545 5 9 cm | 0.584 9 133 cm | 0.504 1 112 cm | 0.495 0 15 cm | 0.370 101 cm | 0.292 1 112 cm | 0.325 109 cm |
| All years | 0.58101 cm | 0.54101 cm | 0.478 11 cm | 0.64910 cm | 0.625910 cm | 0.5310 cm | 0.57144 cm | 0.5311 cm | 0.4713 cm | 0.4311 cm | 0.413112 
[revised manuscript text omitted]